Registered report

# Quantitative evaluation of methods to analyze motion changes in single-particle experiments

Gorka Muñoz-Gil [1] ✉, Harshith Bachimanchi [2], Jesús Pineda [2], Benjamin Midtvedt[2], Gabriel Fernández-Fernández [3], Borja Requena [3], Yusef Ahsini [4], Solomon Asghar[5], Jaeyong Bae[6], Francisco J. Barrantes [7], Steen W. B. Bender [8,9], Clément Cabriel [10], J. Alberto Conejero [4], Marc Escoto[11], Xiaochen Feng [12], Rasched Haidari[13,14], Nikos S. Hatzakis [8,9], Zihan Huang [15], Ignacio Izeddin [10], Hawoong Jeong [6,16], Yuan Jiang[12], Jacob Kæstel-Hansen[8,9], Judith Miné-Hattab[17], Ran Ni [18], Junwoo Park [17], Xiang Qu [15], Lucas A. Saavedra[7], Hao Sha [12], Nataliya Sokolovska[17], Yongbing Zhang [12], Giorgio Volpe [5], Maciej Lewenstein [3,19], Ralf Metzler [20,21], Diego Krapf [22], Giovanni Volpe [2,23] ✉ & Carlo Manzo [24,25] ✉

The analysis of live-cell single-molecule imaging experiments can reveal valuable information about the heterogeneity of transport processes and interactions between cell components. These characteristics are seen as motion changes in the particle trajectories. Despite the existence of multiple approaches to carry out this type of analysis, no objective assessment of these methods has been performed so far. Here, we report the results of a competition to characterize and rank the performance of these methods when analyzing the dynamic behavior of single molecules. To run this competition, we implemented a software library that simulates realistic data corresponding to widespread diffusion and interaction models, both in the form of trajectories and videos obtained in typical experimental conditions. The competition constitutes the first assessment of these methods, providing insights into the current limitations of the field, fostering the development of new approaches, and guiding researchers to identify optimal tools for analyzing their experiments.

Physiological processes occurring in living cells rely on encounters and interactions between molecules. Archetypal examples include gene regulation, transduction of biological signals, and protein delivery to specific locations. All these processes involve the active or passive transport of biomolecules in highly complex, time-varying, and far-from-equilibrium environments, such as the cell membrane (Fig. 1a). One of the most powerful tools to study these transport phenomena is the combination of live-cell single-molecule imaging with single-particle tracking[1,2] because it can provide the time when and location where single events take place (Fig. 1b, c). Alternative ensemble methods (e.g., fluorescence correlation spectroscopy or fluorescence recovery after photobleaching[3]) usually provide limited information because they lose track of crucial details when averaging out spatial and temporal fluctuations.

Methods for single-molecule imaging and single-particle tracking have seen tremendous progress in the last decade, in terms of both

**Fig. 1 | Rationale for the challenge organization. a** The interactions of biomolecules in complex environments, such as dimerization, ligand binding, or trapping at the cell membrane, regulate physiological processes in living systems. These interactions produce changes in molecular motion that can be used as a proxy to measure interaction parameters. **b, c** Time-lapse single-molecule imaging allows us to visualize these processes with high spatiotemporal resolution (**b**) and, in combination with single-particle tracking methods, provide trajectories of individual molecules (**c**). **d, e** Analytical methods can be applied to imaging data, either raw (**b**) or processed in the form of trajectories (**c**), to infer interaction kinetics and quantify their dynamic properties at the ensemble (e.g., probability distributions, **d**) or single-trajectory level (e.g., changepoints, **e**).

experimental acquisition and data analysis[1,2,4,5]. The abundance of experimental single-particle trajectories, encompassing molecules, protein complexes, vesicles, and organelles, has led to the development of numerous methods dedicated to the reliable detection of changes in their motion patterns (as summarized in Supplementary Table 1). These changes serve as valuable indicators for the occurrence of interactions within the system. For instance, diffusing particles may exhibit variations in diffusion coefficients (due to processes like dimerization, ligand binding, or conformational changes) or shifts in their mode of motion (attributed to transient immobilization or confinement at specific scaffolding sites) (Fig. 1a)[6]. These interactions can also result in deviations from standard Brownian motion, as characterized by Einstein's free diffusion model, which includes a linear mean-squared displacement (MSD) and a Gaussian distribution of displacements[7]. This is the case, e.g., of spatiotemporal heterogeneities producing transient subdiffusion at specific timescales[8–19]. Other mechanisms can instead produce asymptotic anomalous diffusion[2,20–22]. Anomalous diffusion compatible with models such as fractional Brownian motion[23–28], continuous-time random walk[29,30], scaled Brownian motion[31], and Lévy walk[32] has been observed for telomers, macromolecular complexes, proteins, and organelles in living cells. Several approaches have been recently proposed to detect and quantify these behaviors[33,34], also involving machine-learning techniques[35–41].

To gain insights into the performance of methods to detect anomalous diffusion from individual trajectories, in 2021, we successfully ran the 1st AnDi Challenge[42]. The discussion that developed between members of diverse research communities working on biology, microscopy, single-particle tracking, and anomalous diffusion (including experimentalists, theoreticians, data analysts, and computer scientists) emphasized the necessity for deeper insights into biologically relevant phenomena. First, it identified a need to evaluate methods to determine the switch between different diffusive behaviors, as often observed in experiments. Second, it highlighted the necessity to assess the methods' crosstalk in detecting inherent anomalous diffusion from nonlinearity in the MSD due to motion

constraints or heterogeneity. Third, it emphasized the importance of determining whether the bottleneck of the analysis process was at the level of the analysis of the single trajectory or associated with their extraction from experimental videos. These needs shaped the design of the 2nd AnDi Challenge, defining its scope with a focus on characterizing and ranking the performance of methods that analyze changes of dynamic behavior. While we retained the name of the 1st AnDi Challenge to build upon its already-established community, the 2nd AnDi Challenge focused mainly on revealing heterogeneity rather than anomalous diffusion. In the simulated datasets, anomalous diffusion emerged from heterogeneity itself or was intentionally introduced for evaluation purposes.

A multitude of methods have been designed to identify and characterize heterogeneous diffusion (Supplementary Table 1). They can be classified based on the heterogeneity they aim to identify or the kind of analysis they perform. We considered three heterogeneity classes that these methods aim to identify: (i) changes in the diffusion coefficient $D$; (ii) changes in the anomalous diffusion exponent $\alpha$ (often classified as subdiffusion, diffusion, or superdiffusion); and (iii) changes in the phenomenological behavior associated with interactions with the environment (often classified as immobilization, confinement, (free) diffusion, and directed motion). While changes in the diffusion coefficient and in the phenomenological behavior have been widely reported, the exploration of changes in the anomalous diffusion exponent is a more recent development[43–46], which is attracting increasing interest also from the theoretical point of view[47–50]. The introduction of new methods for data analysis, as promoted by the Challenge, had the objective to push the performance for detecting subtle changes in these diffusion properties in systems where they could have been overlooked. Along this line, it must be pointed out that the traditional analysis based on the calculation of the scaling exponent of the mean-squared displacement (MSD) can create some ambiguity between the last two classes. Just to provide an example, a particle performing Brownian diffusion in a confined region has an exponent $\alpha = 1$ in terms of the generating motion, but its MSD features a horizontal asymptote at long times, corresponding to $\alpha = 0$. In the

following, we will refer to the exponent $\alpha$ as the characteristic feature of the generating motion.

From the analysis point of view, we identified two classes of methods: (i) ensemble methods, meant to determine characteristic features out of an ensemble of trajectories (Fig. 1d) and (ii) single-trajectory methods, meant to identify changepoint (CP) locations through trajectory segmentation (Fig. 1e). While most available methods rely on the analysis of trajectories obtained from video processing[51], recent advances in computer vision have led to methods capable of directly extracting information from raw movies without requiring the explicit extraction of trajectories[52,53]. Each method has its own set of advantages and disadvantages, and its performance may depend on the specific problem under consideration. However, there is no universally accepted gold standard for determining which method to use to address each specific problem.

To cater to these more advanced needs, we ran an open competition as the 2nd Anomalous Diffusion (AnDi) Challenge. The rationale described above shaped the scope of the challenge, defining the choice of the datasets and the design of the tasks. To rely on an objective ground truth, we assessed the methods' performance on simulated datasets inspired by models of diffusion and interactions documented in biological systems. These datasets describe particles undergoing fractional Brownian motion (FBM,[54]) with piecewise-constant parameters. FBM-type motion has been widely observed in biological systems by means of microrheology, a technique that uses large tracer particles as probes to study the properties of the environment[55]. Anomalous diffusion compatible with FBM has also been reported for telomers and macromolecular complexes in living cells[20,23–28,56]. Beyond this evidence, in the context of the Challenge, FBM served as a tool to enable the tuning of diffusion parameters. The combination of parameter values and interaction models might produce situations that do not correspond to previously documented biological scenarios but will be valuable to test the methods' performance in a wide range of conditions. In biological experiments, other kinds of motion and even non-Gaussian behavior have been reported[21]. However, the choice of FBM did not limit the generality of the Challenge since other models of diffusion and non-Gaussian behavior can be obtained by properly tuning the parameters of the simulations. Datasets provided for the last phase of the competition included motion with parameters inspired by actual experiments for their comparative analysis with the Challenge methods.

The standard and straightforward approach in live-cell single-molecule imaging primarily captures information related to lateral motion. In cases involving flat membranes or isotropic systems, employing 2D imaging and tracking techniques suffices for obtaining accurate motion-related parameters. However, when dealing with motion on non-flat surfaces or within anisotropic 3D environments, relying solely on 2D projections can result in critical information being overlooked, potentially leading to the misinterpretation of diffusion coefficients or the appearance of apparent anomalous diffusion effects[57,58]. Consequently, drawing definitive conclusions under such circumstances should be avoided or approached with caution. To study motion occurring in 3D space, it is advisable to employ 3D tracking methods, such as off-focus imaging (i.e., the analysis of ring patterns in the defocused point spread function)[59], interference/holographic approaches[60], multifocus imaging[61], or point spread function engineering[62]. Although more challenging, these methods can also measure the motion along the axial dimension, facilitating a more thorough characterization. For the purposes of the Challenge, we choose to concentrate on studying changes in diffusion behavior occurring within a 2D context, driven by particle interactions of various types.

While this challenge focused on data inspired by biological systems, the use of regime-switching detection and trajectory segmentation extends well beyond the domain of living cells. Particularly interesting applications also include, e.g., the analysis of biomedical signals[63], speech[64], traffic flows[65], seismic signals[66], econometrics[67,68], ecology[69], and river flows[70].

## Results

### Datasets and ground truth

In order to benchmark the different methods on data with a known ground truth, we relied on numerical simulations. We developed the `andi-datasets` Python package[71] to generate the required datasets to train and evaluate the various methods. Details about available functions can be found in the hosting repository[71].

Particle motion was simulated according to fractional Brownian motion (FBM,[54]), a model that reproduces Brownian and anomalous diffusion processes by tuning the correlation of the increments through the Hurst exponent $H$. FBM is a Gaussian process with a covariance function

$$E[B_H(t)B_H(s)] = K(t^{2H} + s^{2H} - |t - s|^{2H}), \qquad (1)$$

where $E[\cdot]$ denotes the expected value and $K$ is a constant with units length$^2 \cdot$ time$^{-2H}$. In order to generalize FBM in two dimensions (2D), a trajectory $\mathbf{R}(t)$ is represented as $\mathbf{R}(t) = \{X(t), Y(t)\}$, where $X(t)$ and $Y(t)$ are independent FBM processes along the $x$ and $y$ axes, respectively[33]. The anomalous diffusion exponent is related to the Hurst exponent as $\alpha = 2H$[54], and the MSD for an unconstrained FBM in 2D scales with time $t$ as

$$\text{MSD}(t) = 4Kt^\alpha. \qquad (2)$$

When $\alpha = 1$, FBM reverts to Brownian motion and $K$ corresponds to the diffusion coefficient $D$. FBM describes subdiffusion for $0 < H < 1/2$ ($0 < \alpha < 1$), Brownian diffusion for $H = 1/2$ ($\alpha = 1$), and superdiffusion for $1/2 < H < 1$ ($1 < \alpha < 2$).

We considered the following physical models of motion and interactions (Fig. 2a):

- Single-state model (SSM)—Particles diffusing according to a single diffusion state, as observed for some lipids in the plasma membrane[14,15,72]. This model also serves as a negative control to assess the false positive rate of detecting diffusion changes.
- Multi-state model (MSM)—Particles diffusing according to a time-dependent multi-state (2 or more) model of diffusion undergoing transient changes of $K$ and/or $\alpha$. Examples of changes of $K$ have been observed in proteins as induced by, e.g., allosteric changes or ligand binding[73–76].
- Dimerization model (DIM)—Particles diffusing according to a 2-state model of diffusion, with transient changes of $K$ and/or $\alpha$ induced by encounters with other diffusing particles. Examples of changes of $K$ have been observed in protein dimerization and protein-protein interactions[77–81].
- Transient-confinement model (TCM)—Particles diffusing according to a space-dependent 2-state model of diffusion, observed for example in proteins being transiently confined in regions where diffusion properties might change, e.g., the confinement induced by clathrin-coated pits on the cell membrane[82]. In the limit of a high density of trapping regions, this model reproduces the picket-and-fence model used to describe the effect of the actin cytoskeleton on transmembrane proteins[9,83].
- Quenched-trap model (QTM)—Particles diffusing according to a space-dependent 2-state model of diffusion, representing proteins being transiently immobilized at specific locations as induced by binding to immobile structures, such as cytoskeleton-induced molecular pinning[17,84].

While the interaction mechanisms producing the heterogeneous diffusion are inspired by biological scenarios, some of the combinations

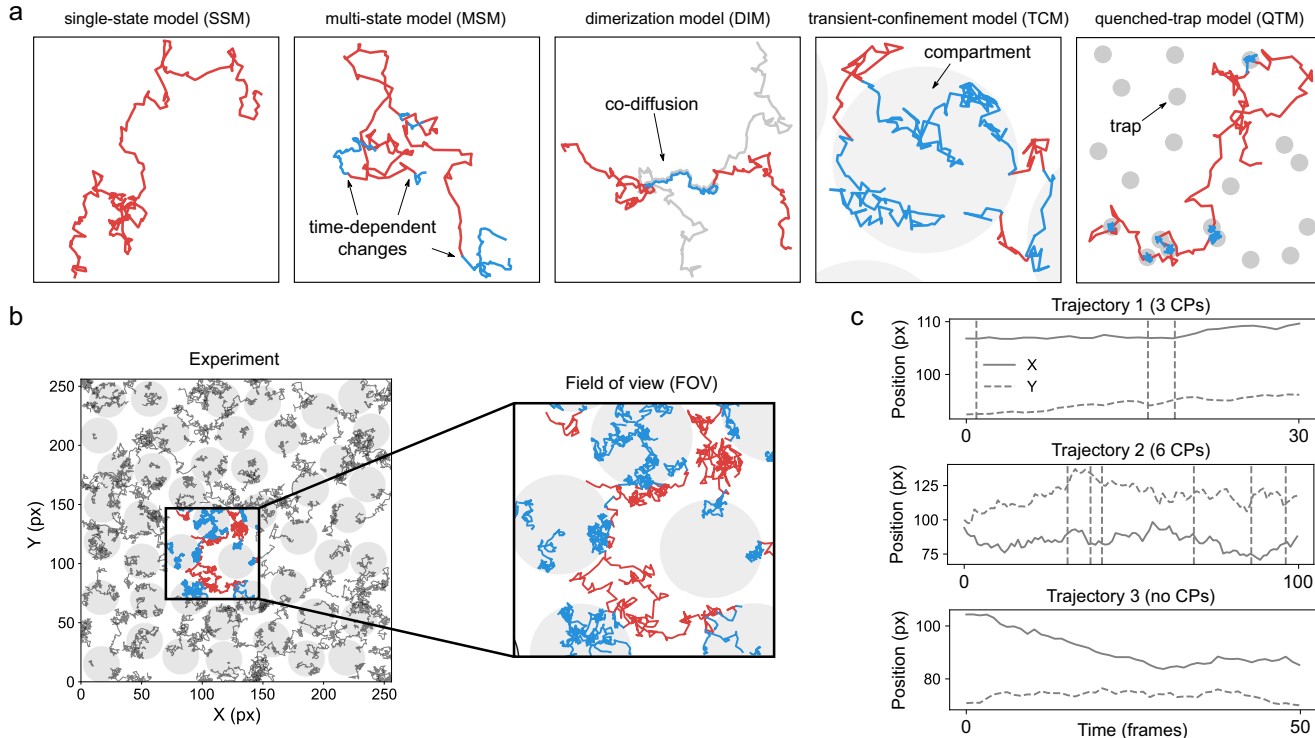

**Fig. 2 | Physical models of interaction and structure of the simulated datasets.**
**a** Examples of 2-dimensional trajectories undergoing interactions inducing changes in their motion. From left to right: single-state model (SSM) without changes of diffusion; multi-state model (MSM) with time-dependent changes between different diffusive states (red and blue); dimerization model (DIM) where a particle (red) selectively interacts with another particle (gray) and the two transiently co-diffuse with a different motion (blue trajectory); transient-confinement model (TCM) where a particle diffuses inside (blue) and outside (red) compartments with osmotic boundaries (gray area); quenched-trap model (QTM) where a particle is transiently immobilized (blue) at specific loci through interactions with static features of the environment (gray areas). **b** An experiment (left panel) consists of simulations performed according to one of the models of interactions described in (**a**) (here showing a TCM experiment), with a set of parameters describing the dynamic interplay of the particles and/or the environment. From the same experiment, several fields of view (FOVs) are selected. Particles within the same FOV (right panel) diffuse and undergo interactions among themselves and/or with the environment (gray areas) that affect their trajectories. **c** Time traces of the coordinates of exemplary trajectories from the experiment depicted in (**b**), displaying changes of diffusion properties at specific times (changepoints CPs, dashed vertical lines). For the Challenge, the motion analysis can be either performed directly from the video recording of the FOV (Video Track) or from detected trajectories linking the coordinates of individual particles at different times (Trajectory Track).

of diffusion parameters and models lead to situations that may not correspond to previously documented biological contexts. Nevertheless, this approach holds substantial value as it enables the comprehensive assessment of method performance across a broad spectrum of conditions.

In the simulations, each dynamic state is characterized by a distribution of values for the parameters $K$ and $\alpha$. For each trajectory, the values of $K$ and $\alpha$ for each state are randomly drawn from Gaussian distributions with bounds $\alpha \in (0, 2)$ and $K \in [10^{-12}, 10^6]$ pixel$^2$/frame$^\alpha$. The interaction distance and the radius of confinement or trapping have constant values across each experiment. Simulations are provided in generalized units (i.e., pixels and frames) that can be rescaled to meaningful temporal and spatial scales.

A detailed description of the simulation procedure is presented in Extended Methods.

**Competition design**
To enable the assessment of the performance of previously established methods while fostering the development of new approaches and the participation from diverse disciplines, the challenge was organized along two tracks:
- Video Track—based on the analysis of raw videos.
- Trajectory Track—based on the analysis of trajectories.

For each track, datasets were provided according to a hierarchical structure (Fig. 2b, c) that includes:

- Experiment—A given biological scenario defined by a model of interactions and a set of parameters describing the dynamic interplay of the particles and the environment.
- FOV—A region of the sample where the recording takes place. Particles within the same field of view (FOV) can undergo interactions among themselves and/or with the environment.
- Video (Video Track only)—Videos corresponding to each FOV.
- Trajectory (Trajectory Track only)—Trajectory corresponding to the motion of an individual particle.

For both tracks, all particles used in the simulations and located in the FOV are provided/visualized (i.e., full labeling conditions). The effect of blinking or photobleaching was not taken into account.

In each track, participants could compete in two different tasks, as typically done in the analysis of experimental data:
- Ensemble Task—Ensemble-level predictions providing, for each experimental condition, the model used to simulate the experiment, the number of states, and the fraction of time spent in each state. For each identified state, participants had to determine the mean and standard deviation of the distribution of the generalized diffusion coefficients $K$, and the mean and standard deviation of the distribution of the anomalous diffusion exponent $\alpha$ corresponding to the underlying motion.
- Single-trajectory Task—Trajectory-level predictions providing for each trajectory a list of $M$ inner CPs delimiting $M + 1$ segments with different dynamic behavior. For each segment, participants had to

identify the generalized diffusion coefficient $K$, the anomalous diffusion exponent $\alpha$ corresponding to the underlying motion, and an identifier of the kind of constraint imposed by the environment ($0 =$ immobile, $1 =$ confined, $2 =$ free (unconstrained, $0.05 \leq \alpha < 1.9$), $3 =$ directed ($1.9 \leq \alpha < 2.0$). For the Video Track, predictions had to be provided for a subset of particles (in the following, we will refer to them as VIP, very important particles) identified through a label map of the first frame of the movie. For the Trajectory Track, predictions had to be provided for all trajectories in the FOV.

For each task, several metrics were evaluated (see Scoring and evaluation). Participants were allowed to provide partial submissions, e.g., including predictions for a limited subset of experiments or for specific parameters. For ranking purposes of the Challenge, missing predictions were scored with the worst possible value of the corresponding metric.

## Competition overview

The 2nd AnDi Challenge was held between December 1, 2023, and July 15, 2024, on the Codalab platform. It was divided into three phases, namely *Development*, *Validation*, and *Challenge*. The Development Phase (5 months) was intended for the participants to set up their methods, test them, and familiarize themselves with the datasets and the scoring platform. An unlabeled dataset was available, and the public leaderboard showed scores obtained on this dataset. An online workshop was held on February 22, 2024, to instruct the participants about the details of the challenge. The Validation Phase (1 month) was a test of the actual final challenge. A new dataset (described in Challenge Dataset) was provided, and the leaderboard was again public. The Challenge Phase (15 days) was the final stage of the competition. A new dataset was provided, and the number of submissions per team was limited to 1 per day. The results were not publicly disclosed, and the leaderboard was made public only after the end of the competition. In total, we received 1343 submissions during the three phases. Participants registered in teams of 1 to 5 people. In the final stage, out of 80 registered participants, 53 individuals, divided into 18 teams, were included in the leaderboard (see Supplementary Table 2 for the list of participating teams). The teams' affiliations spanned Europe (12 teams), Asia (6 teams), and America (1 team). From the final leaderboard, members of the top 5 teams in each task were invited to co-author this article. An overview of these teams and the methods is provided in Supplementary Information− Overview of Teams and Methods.

The results of the Challenge were discussed with the participants and other experts from the field during the 2nd Anomalous Diffusion Workshop that was held in June 2025.

## Challenge dataset

The Challenge dataset was composed of 12 experiments corresponding to different diffusion models and parameter values. Details about the numeric values of parameters of the experiments are given in Supplementary Table 3. In addition, Supplementary Fig. 1 summarizes the distribution of specific features within the dataset. EXP 1 aimed at mimicking multistate diffusion in membrane proteins. Average diffusion coefficients and the transition matrix of the MSM were chosen to reproduce, with the appropriate scaling, the three fastest states reported for the diffusion of the $\alpha$2A-adrenergic receptor[80]. EXP 2 reproduced changes in diffusion coefficient due to protein dimerization, inspired by the behavior reported for the epidermal growth factor receptor ErbB1[77]. EXP 3, EXP 4, and EXP 5 were designed to compare the methods' ability to detect changes from the same free diffusive state to a slow diffusing state characterized either by traps (QTM, EXP 3), small confinement regions (TCM, EXP 4), or a subdiffusive dimeric state (DIM, EXP 5). EXP 6 and

EXP 7 were meant to assess the methods' ability to take advantage of the knowledge of the physical model itself and additional information present in the experiment to improve predictions. The experiments corresponded to different theoretical models (DIM and MSM) with the same diffusive parameters. EXP 8 served as a negative control and contained only SSM trajectories with very broad distributions of $K$ and $\alpha$. EXP 9 was generated from QTM with very short trapping times and superdiffusion in the free state to assess how the methods deal with such extreme conditions. The other three experiments contained data with extreme and unrealistic parameters meant to assess potential biases of the methods, and will not be discussed further.

## Scoring and evaluation

The performance of the methods was evaluated using specific metrics for each task. For ranking purposes in the Challenge, composite metrics were used, as described below.

**Ensemble task.** Participation in the Ensemble Task required predictions of the type of model used for simulating each experiment, the number of states $S$ of the model, and the parameters of each state. The type of model was simply evaluated as correct or wrong. The prediction of the number of states was assessed by measuring the difference with the ground truth. For both the generalized diffusion coefficient and the anomalous diffusion exponent, predictions had to include the mean, the standard deviation, and the relative weight of each state. From these values, we computed the associated multi-modal distributions $P_\alpha$ and $P_D$. The similarity of these distributions to the ground-truth distributions $Q_\alpha$ and $Q_D$ was assessed by means of the first Wasserstein distance ($W_1$),

$$W_1(P, Q) = \int_{\text{supp}(Q)} |\text{CDF}_P(x) - \text{CDF}_Q(x)| dx \qquad (3)$$

where $\text{CDF}_Q$ is the cumulative distribution function of the distribution $Q$ and $\text{supp}(Q)$ is the support ($\alpha \in (0, 2)$ and $K \in [10^{-12}, 10^6]$ pixel$^2$/frame$^\alpha$).

**Single-trajectory task.** Participation in the Single-trajectory Task required predictions of the $M$ CPs and the dynamic properties, i.e., the generalized diffusion coefficient $K$, the anomalous exponent $\alpha$, and diffusive-type identifiers of the resulting $M + 1$ segments. Different metrics were used to evaluate the methods' performance.

**CP detection metrics.** Following Ref. 51, given a ground-truth CP at locations $t_{(\text{GT}),i}$ and a predicted CP at locations $t_{(\text{P}),j}$, we defined the gated absolute distance:

$$d_{i,j} = \min(|t_{(\text{GT}),i} - t_{(\text{P}),j}|, \varepsilon_{\text{CP}}), \qquad (4)$$

where $\varepsilon_{\text{CP}}$ was used as a fixed maximum penalty for CPs located more than $\varepsilon_{\text{CP}}$ apart. For a set of $M_{\text{GT}}$ ground-truth CPs and $M_{\text{P}}$ predicted CPs, we solved a rectangular assignment problem using the Hungarian algorithm[85] by minimizing the sum of distances between paired CPs:

$$d_{\text{CP}} = \min_{\text{paired CP}} \left( \sum d_{i,j} \right). \qquad (5)$$

The distance $d_{\text{CP}}$ allows to define a pairing metric:

$$\alpha_{\text{CP}} = 1 - \frac{d_{\text{CP}}}{d_{\text{CP}}^{\max}}, \qquad (6)$$

where $d_{\text{CP}}^{\max} = M_{\text{GT}} \varepsilon_{\text{CP}}$ is the distance associated with having all predicted CPs unpaired or at a distance larger than $\varepsilon_{\text{CP}}$ from all ground-

truth CPs. The metric $\alpha_{CP}$ is bound in [0, 1], taking a value of 1 if all ground-truth and predicted CPs are matching exactly. Similarly, we define a CP localization metric:

$$\beta_{CP} = \frac{d_{CP}^{max} - d_{CP}}{d_{CP}^{max} + \overline{d_{CP}}}, \tag{7}$$

where $\overline{d_{CP}}$ is the distance associated with having all unassigned predicted CPs at a distance larger than $\varepsilon_{CP}$ from all ground-truth CPs. This metric measures the presence of spurious CPs and is bound in [0, $\alpha_{CP}$], taking value $\alpha_{CP}$ if no spurious CPs are present. We also calculate the number of true positives (TP), i.e., the paired true and predicted CPs with a distance smaller than $\varepsilon_{CP}$. Spurious predictions, i.e., not associated with any ground truth or having a distance larger than $\varepsilon_{CP}$ were counted as false positives (FP). Ground truth CPs not having an associated prediction at a distance shorter than $\varepsilon_{CP}$ were considered false negatives (FN). Given an experiment containing $N$ trajectories, we computed the overall number of TP, FP, and FN. We then used these values to calculate the JSC over the whole experiment as:

$$JSC = \frac{TP}{TP + FN + FP}. \tag{8}$$

For the predicted CPs classified as TP, we also computed the root mean square error (RMSE), defined as:

$$RMSE = \sqrt{\frac{1}{N} \sum_{\substack{\text{paired CP} \\ d_{i,j} < \varepsilon_{CP}}} \left( t_{(GT),i} - t_{(P),j} \right)^2}. \tag{9}$$

**Metrics for the estimation of dynamic properties.** For the evaluation of the methods' performances on the estimation of the dynamic properties, we first followed a procedure similar to the one described above for the pairing of the CPs. Predicted CPs were used to define the predicted trajectory segments. We defined a distance between predicted and ground-truth segments based on the JSC calculated with respect to their temporal support, where time points at which predicted and ground-truth segments overlap were considered as TP, predicted time points not corresponding to the ground truth as FP, and ground-truth time points not predicted as FN. The Hungarian algorithm was used to pair segments by maximizing the sum of the JSC. Only paired segments were used to calculate metrics assessing methods' performance for the estimation of dynamic properties. For the generalized diffusion coefficient $K$, we used the mean squared logarithmic error (MSLE) defined as:

$$MSLE = \frac{1}{N} \sum_{\substack{\text{paired} \\ \text{segments}}} \left( \log(K_{(GT),i} + 1) - \log(K_{(P),j} + 1) \right)^2. \tag{10}$$

For the anomalous diffusion exponents $\alpha$, we used the mean absolute error (MAE):

$$MAE_\alpha = \frac{1}{N} \sum_{\substack{\text{paired} \\ \text{segments}}} |\alpha_{(GT),i} - \alpha_{(P),j}|, \tag{11}$$

where $N$ is the total number of paired segments in the experiment, $\alpha_{(GT),i}$ and $\alpha_{(P),j}$ represent the ground-truth and predicted values of the anomalous exponent of paired segments, respectively. For the

classification of the type of diffusion, we used the $F_1$-score:

$$F_1 = \frac{2TP_c}{2TP_c + FP_c + FN_c}, \tag{12}$$

where $TP_c$, $FP_c$, and $FN_c$ represent true positives, false positives, and false negatives with respect to segment classification. The metric was calculated as a micro-average, which aggregates the contributions of all classes to compute the average metric and is generally preferable when class imbalance is present.

### Metrics for challenge ranking
For ranking purposes, we used the mean reciprocal rank (MRR) as a summary statistic for the overall evaluation of software performance[42]:

$$MRR = \frac{1}{N} \cdot \sum_{i=1}^{N} \frac{1}{rank_{M_i}}, \tag{13}$$

where $rank_{M_i}$ corresponds to the position in an ordered list based on the value of the corresponding metrics $M_i$.

For the Ensemble Task, the metrics involved in the calculation were the $F_1$-score of the model and the MAE of the distributions of $K$ and $\alpha$. For the Single-trajectory Task, we used the JSC and the RMSE of CPs, the MSLE of $K$, and the MAE of $\alpha$.

### Overview of the challenge results
The Challenge dataset was comprehensively designed to test the submitted methods under distinct scenarios, using ad hoc metrics to evaluate their specific capabilities. For ranking, we employed composite metrics that aggregate the scores from different experiments and subtasks. The results are summarized in Fig. 3. Here, we present an overview of the Challenge results, highlighting the general trends observed. The complete rankings are provided in Supplementary Fig. 2.

In the Single-trajectory Task (Fig. 3a), one method based on UNet3+[86,87] (team I) clearly outperformed the others, whereas the Ensemble Task (Fig. 3b) showed a more balanced competition. From the MRR breakdown, we observed that the top team in the Single-trajectory Task performed consistently well across all metrics. In contrast, for the Ensemble Task, the top teams improved their final ranks by specializing in one of the two subtasks.

We also show the correlation between pairs of metrics associated with CP detection (Fig. 3c) and the prediction of diffusive properties (Fig. 3d, e). The predictions for the Video Track (represented by filled squares) are also included alongside those of the Trajectory Track (represented by empty circles). Across methods, enhanced CP detection, reflected by higher JSC and lower RMSE, yields a tight correlation between these metrics (Fig. 3c). A similar but weaker trend appears for $K$ and $\alpha$ errors (Fig. 3d, e), because their estimation often relies on distinct algorithms, decoupling improvements in one from the other.

In the plots, the dashed lines connect the predictions of teams participating in both tracks. All teams in the Video Track (teams E and Q for the Single-trajectory Task, teams E and F for the Ensemble Task), except for team K, improved their predictions in the Trajectory Track compared to the Video Track. Notably, all four teams first extracted the trajectories using a previously established tracking method[5,40,88–91] and then performed the ensuing analysis using the same method developed for the Trajectory Track. While this highlights the influence of error associated with the tracking process[51], none of the methods explored the possibility of obtaining results directly from the video, which was one of the exploratory goals of this competition.

Finally, Fig. 4 shows the score obtained for subtask metrics by all teams for each experiment (filled symbols). The consistently lower performance of the Video Track compared to the Trajectory Track

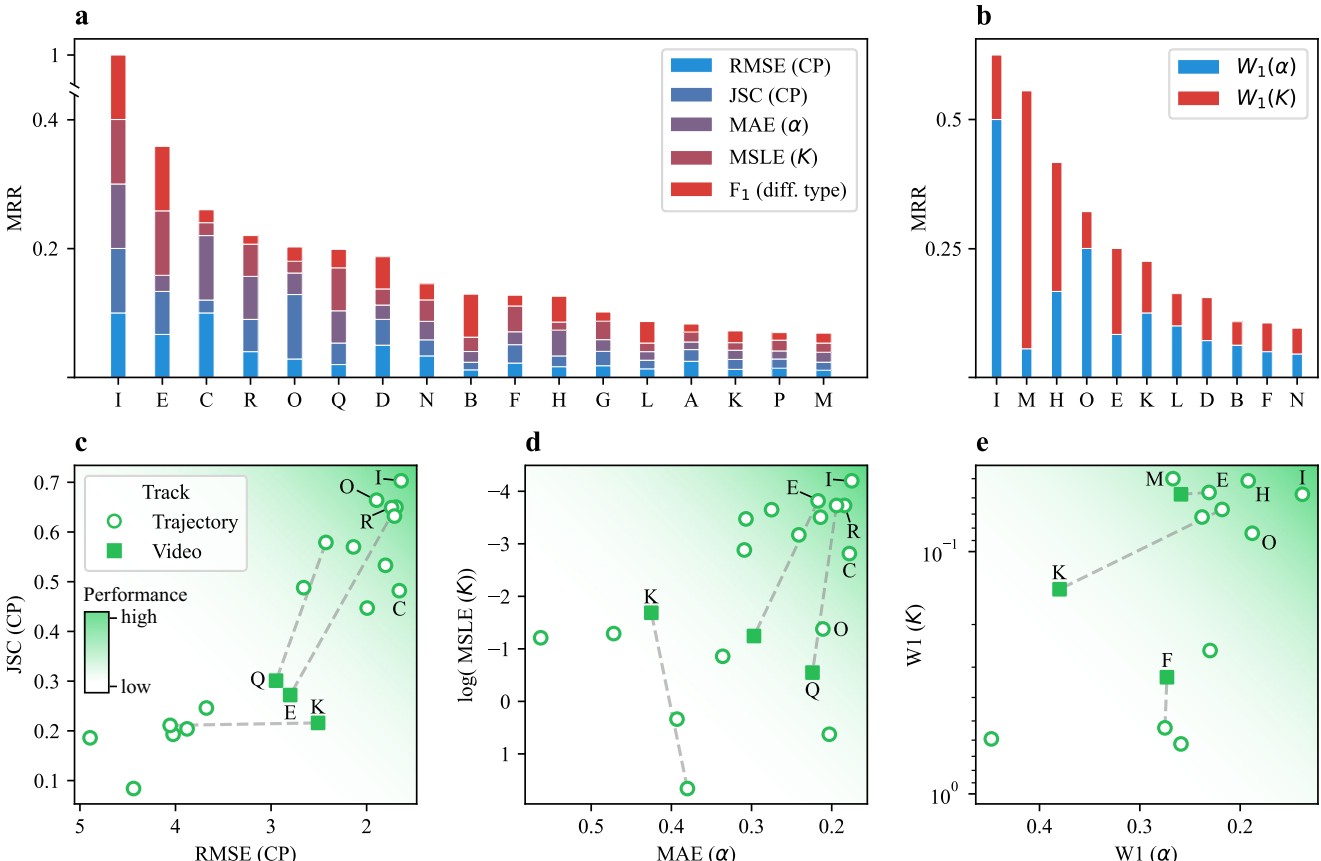

**Fig. 3 | Challenge rankings.** Mean reciprocal rank (MRR) of all methods participating in the Single-trajectory Task (**a**) and Ensemble Task (**b**) in the Trajectory Track. The colors represent the relative contributions of the metrics of each subtask to the overall MRR. **c**–**e** Correlation between subtask metrics associated with changepoint (CP) detection (**c**), the prediction of segment properties (**d**) in the Single-trajectory Task, and the prediction of diffusive properties in the Ensemble Task (**e**). Empty circles and filled squares represent the metrics obtained for the Video Track and the Trajectory Track, respectively. Dashed lines join results obtained by the same team in the two tracks. The darker background color indicates the area of the plot corresponding to better performances. Source data are provided as a Source Data file.

lends support to the third rationale: it suggests that challenges in accurately extracting trajectories from experimental videos represent a more significant bottleneck than the downstream analysis of pre-extracted tracks.

These plots provide further insight into which experimental conditions were more challenging for each subtask. For example, CP detection in EXP 1 (MSM with 3 states) was particularly difficult, as indicated by the low JSC in Fig. 4a. As shown in Fig. 4e, classification of the type of diffusion for EXP 4 (TCM) was more challenging than EXP 3 (QTM), despite having similar parameters for the unrestrained motion. For the Ensemble Task, we observe poorer predictions for $K$ in EXP 8 (SSM, Fig. 4f) and for $\alpha$ in EXP 9 (QTM, Fig. 4g). In the following, we will comparatively discuss results obtained for groups of experiments aimed at detecting specific method capabilities. For most of these analyzes, we will mainly consider the methods of the top 5 teams in each Track and Task.

**CP detection and segment diffusion properties**

A main aspect of the Challenge was the evaluation of CP detection capability and the ensuing assessment of diffusion properties for the identified segments. In particular, we tested the methods' ability to distinguish true anomalous diffusion from subdiffusive behavior that emerges solely from physical constraints, directly addressing the second rationale. These insights were provided by the Single-trajectory Task.

As shown in Fig. 4a–e, the methods generally performed well when tested on time-varying processes. We sought to characterize the

false positive rate of the methods by evaluating their behavior over the trajectories of EXP 8 having no CPs (Fig. 5a, b). EXP 8 also served to assess the methods' ability to estimate parameters $K$ and $\alpha$ independently of errors induced by incorrect segmentations. Submitted predictions were benchmarked with the estimations of $K$ and $\alpha$ obtained by linear and logarithmic fits of the MSD, respectively (dashed lines). Most methods predicted very few CPs for these trajectories, producing a low false positive rate and outperformed the MSD fit for both $K$ and $\alpha$ (Fig. 5a, b).

A relevant aspect associated with CP detection accuracy is its dependence on the number of CPs per trajectory, shown in Fig. 5c–e, which is inversely related to the average segment duration. As expected, the JSC shows worse performance as the number of CPs increases (Fig. 5c). Regarding the diffusion parameter estimation, we observe that the methods allow a robust estimation of $K$ independently of the number of CPs (Fig. 5d), whereas for $\alpha$ we observe a drop in performance as the number of CPs increases (Fig. 5e). This confirms the difficulty of estimating $\alpha$ from short segments, due to its asymptotic nature, already observed in the 1st AnDi Challenge[42].

**Classification of types of diffusion**

One of the goals of this competition was to assess the methods' ability to classify different diffusion types and distinguish among distinct physical models. Results for all experiments of the Video and Trajectory Tracks are shown in Supplementary Figs. 3 and 4, respectively. The results of the two tracks were qualitatively similar but the Video Track had overall lower scores since all teams except team Q missed

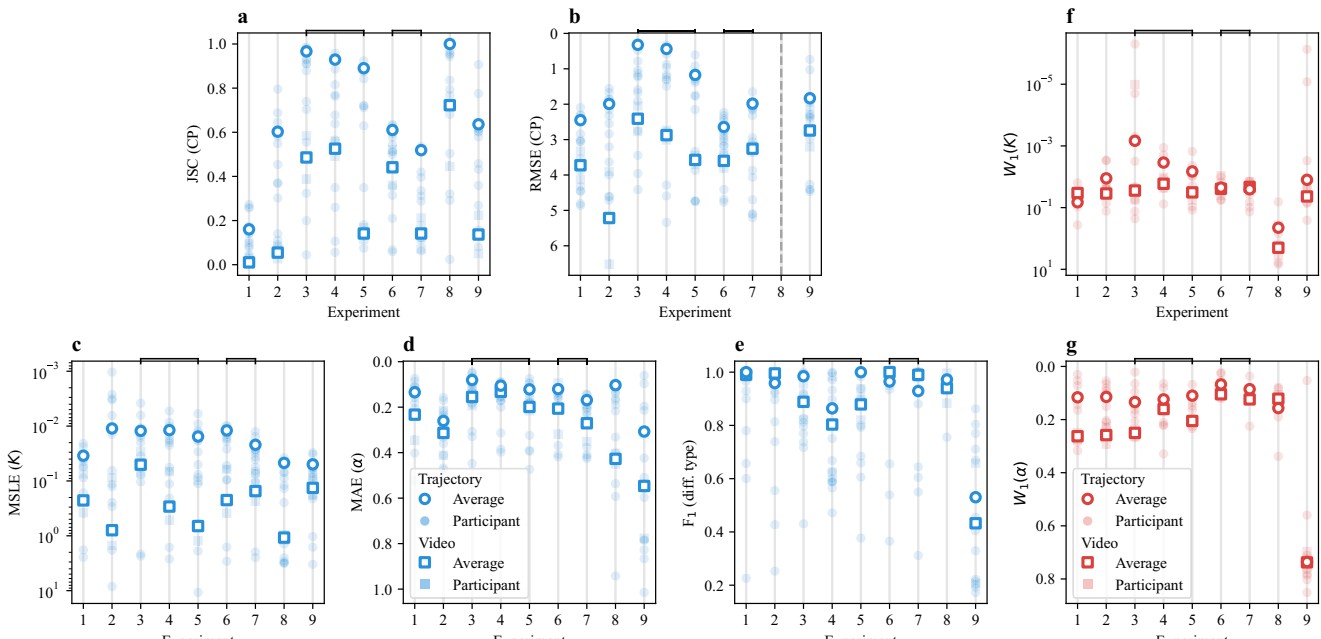

**Fig. 4 | Overview of the results. a–e** Scores obtained for each subtask metric by all teams for each experiment of the Single-trajectory Task (filled symbols). Squares and circles represent the metrics obtained for the Video Track and the Trajectory Track, respectively. **f, g** Scores obtained for each subtask metric by all teams for each experiment of the Ensemble Task (filled symbols). Squares and circles represent the metrics obtained for the Video Track and the Trajectory Track, respectively. In all panels, open symbols represent average scores. The vertical axes are arranged so that the best performance is always shown at the top. Horizontal square brackets indicate groups of experiments that are discussed comparatively. Source data are provided as a Source Data file.

the immobile state (Supplementary Fig. 3). To summarize the methods' ability to assign segments to diffusion types, in Fig. 6 we show the distribution of each diffusive state compared to the ground truth (horizontal segments) for representative experiments of the Trajectory Track. In Fig. 6a we exemplarily show the results obtained for EXP 9, a QTM with an unconstrained state having a narrow distribution of $K$ but with $\alpha$ values that could produce either superdiffusive or directed motion. In this case, only the top method (team I, light blue) was able to produce a reliable classification of the diffusion type of the segments. The difficulty in inferring the correct type of mechanism producing interaction underscores the challenges in accurately analyzing this kind of data, which can have significant implications for the biological interpretation of the results. Although perfect classification of diffusive states remains challenging, the algorithms nonetheless provide precise estimates of critical biophysical parameters, namely, the average dwell times in both trapped and unconstrained states (inset of Fig. 6a). The measure of these parameters is essential for quantifying binding kinetics, confinement lifetimes, and transition rates that directly inform biological interpretation.

The second rationale for the Challenge was to probe the methods' ability to disentangle genuine anomalous diffusion from subdiffusive behaviors arising purely from motion constraints. To test the methods in challenging conditions, we designed a group of experiments (EXP 3, EXP 4, and EXP 5) with different underlying models but with diffusive parameters that produce similar trajectories. The three experiments share an unconstrained state with normal diffusion, and $K \approx 1$: EXP 3 is simulated as QTM, whereas EXP 4 is from a TCM with a small confinement radius and $\alpha \approx 0.2$, and EXP 5 is DIM with a dimeric state with $\alpha \approx 0.2$. Other parameters were set to obtain similar residence times in the different states. Figure 6b–d highlights the performance of the top five methods across EXP 3–5. Teams I, C, and R each correctly classify over 95% of segments, closely matching the true distribution of diffusive states. Team E tends to over-label segments as diffusive, while Team O occasionally confuses confined segments for diffusive ones

and vice versa. Team R, despite its high overall accuracy, also makes occasional misclassifications of diffusive segments as immobile or confined. Importantly, for EXP 4 (small-radius confinement) and EXP 5 (dimerization-induced subdiffusion), misclassification as immobile is negligible for Teams I, C, and R. Detecting confinement in EXP 4 is particularly challenging since short dwell times in confined areas yield few boundary reflections, inducing confusion with unconstrained anti-persistent subdiffusion of EXP 5. The ability of Teams I, C, and R to resolve these subtle cases underscores the high sensitivity and robustness of their methods.

**Using physical models to enhance method performance**

The information contained in an individual trajectory is typically sufficient to estimate CPs and diffusive properties. However, for some physical models, the knowledge of the model itself offers additional information that could be used to improve further CP detection and parameter estimation. This is the case for QTM and TCM, where changes in diffusion correspond to spatial constraints. For DIM, diffusion changes are associated with particle proximity; in addition, since particles in a dimer co-diffuse, one could, in principle, use twice as much information to estimate $K$ and $\alpha$, although in typical experimental conditions it may be very challenging to track two co-diffusing particles.

Along these lines, for the Single-Trajectory Task, the lowest JSC values were obtained for EXP 1 and EXP 7 (circles in Fig. 4a). Both experiments correspond to simulations of MSM, a model where the diffusion changes are produced in a purely time-dependent fashion and the dataset itself does not provide additional hints to determine them. This suggests that the methods can directly or indirectly take advantage of the presence of a physical event (e.g., trapping, confinement, or dimerization) to enhance CP detection accuracy. To assess this effect quantitatively, we used EXP 5 and EXP 6, which correspond to different physical models (DIM and MSM, respectively) generated with an identical set of diffusive parameters. To quantify

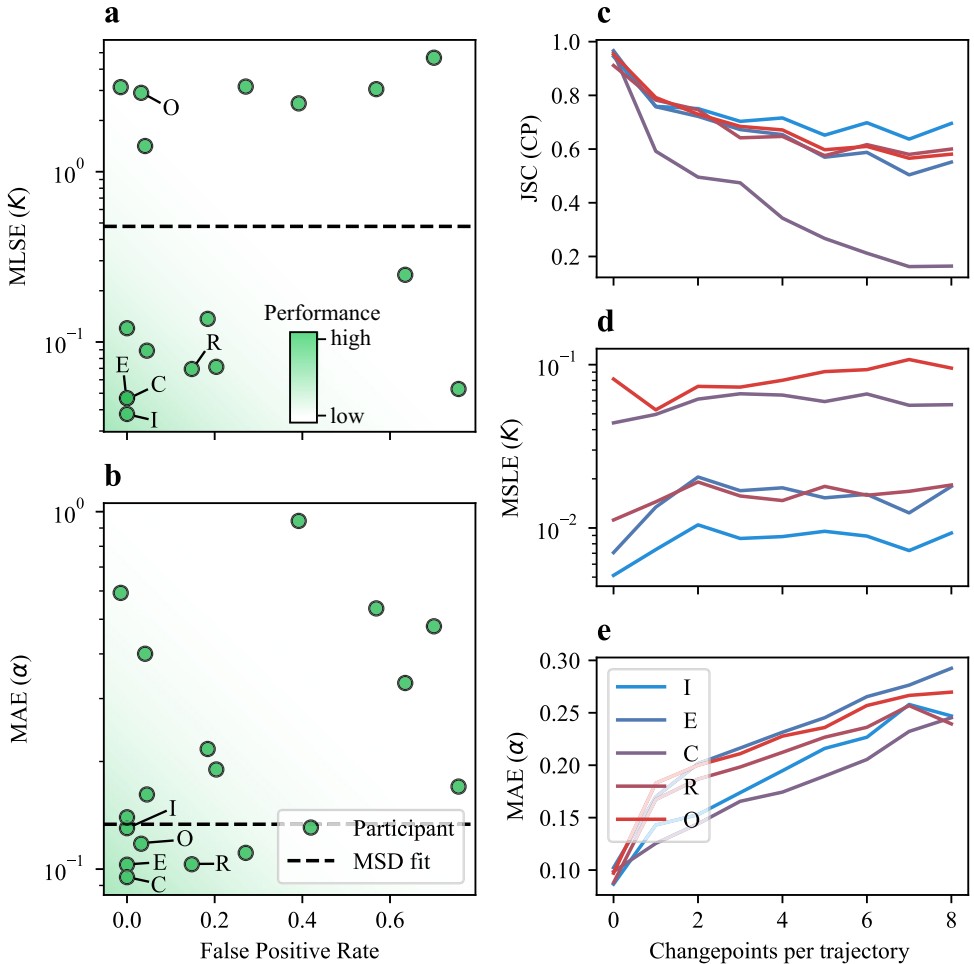

**Fig. 5 | CP detection and segment diffusion properties. a, b** Scatter plots of the metrics associated with segment diffusion properties in the Single-trajectory Task of the Trajectory Track as a function of the false positive rate calculated for EXP 8, composed of trajectories without CPs. Results obtained by all participants are shown. Dashed lines correspond to the scores obtained by using the fit of the mean-squared displacement, as a benchmark. The darker background color indicates the area of the plot corresponding to better performances. **c–e** Dependence of the JSC, MSLE, and MAE on the number of changepoints for the Single-trajectory Task of the Trajectory Track. Only the results of the top 5 teams are shown. The color code represents their position in the ranking (blue is the highest, red the lowest). Source data are provided as a Source Data file.

model-based gains, we computed the relative improvement

$$\Delta m(\%) = \frac{m_{\text{DIM}} - m_{\text{MSM}}}{m_{\text{MSM}}} \times 100\% \qquad (14)$$

for each subtask metric (JSC, MSLE, and MAE). Figure 7 reports these improvements for all methods, with the overall average shown as a dashed line.

Surprisingly, while most of the methods showed improved performance for the CP prediction in DIM (Fig. 7a), there were minor differences in the prediction of diffusive properties (Fig. 7b, c). We believe this is because the methods predict each trajectory's properties without considering it in the ensemble of the FOV or of the experiment, an observation that may improve the next generation of methods.

**Ensemble predictions**
The Ensemble Task was designed to test whether the methods could take advantage of the increased statistics obtained from common parameters shared by all trajectories within the same experiment to better identify the type of motion and estimate its parameters. As discussed earlier, several approaches of this type have been devised and used in the past to extract biophysical information from single-

particle tracking data (Supplementary Table 1). However, no pure ensemble-level method, i.e., one that disregards the individual trajectory identity, was employed for the Challenge. Instead, all teams that provided submissions for the Ensemble Task used predictions obtained at the single-trajectory level, which were then pooled together to estimate the moments of the distributions of the diffusive parameters. Results for all experiments of the Video and Trajectory Tracks are shown in Supplementary Figs. 5–8. The resulting distributions are summarized in Fig. 8 for 4 exemplary experiments (EXP 4, EXP 7, EXP 8, and EXP 9) of the Trajectory Track. The pooling operation was performed using two general approaches: teams either applied a Gaussian mixture model (GMM) or a clustering algorithm on the predicted segments to extract subpopulation parameters, with four of the top 5 teams opting for the former approach (teams E, I, M, and O). Interestingly, as it can be inferred from Fig. 3a, b, the scores obtained by the teams participating in both tasks showed a low correlation. Therefore, accurate predictions at the single-trajectory level do not necessarily translate into reliable ensemble-level predictions, pointing to a critical role of the clustering approach.

Figure 8a, b shows an experiment where all teams provided consistent and reasonable predictions. This is particularly evident for the $K$ distribution in EXP 7 and EXP 8 (Fig. 8b, c). Since the methods

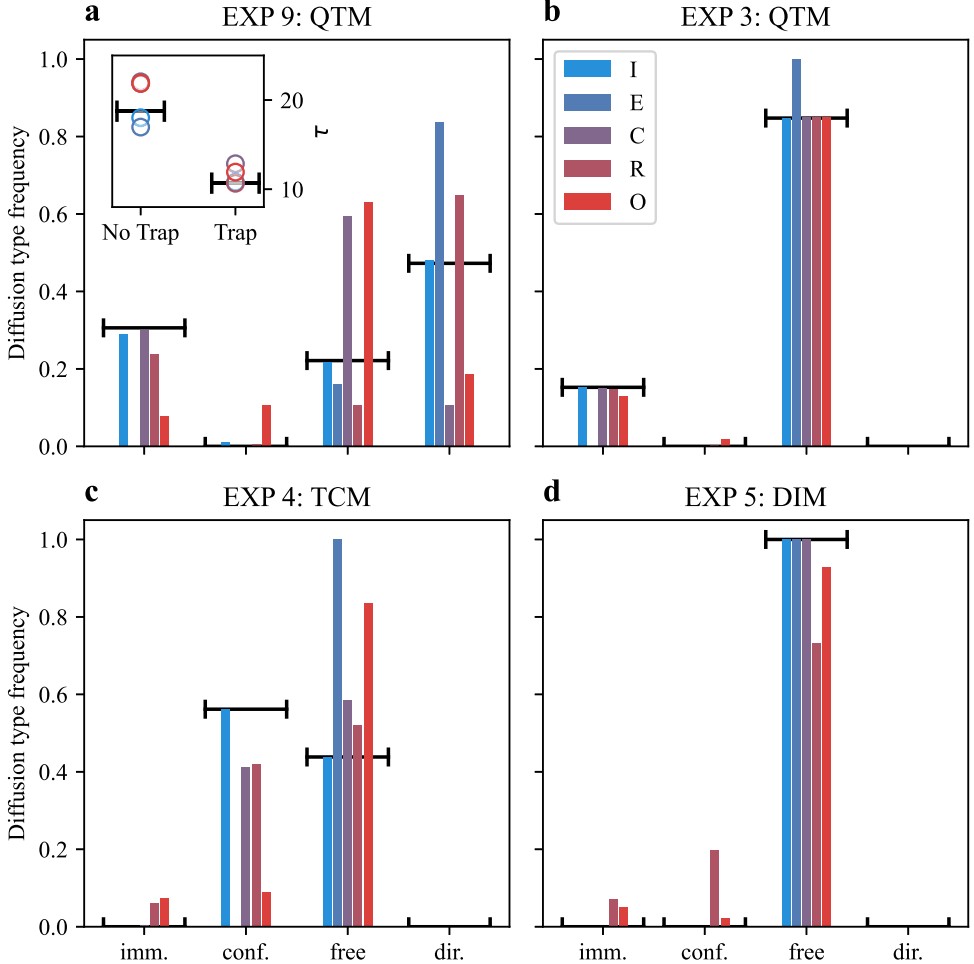

**Fig. 6 | Classification of types of diffusion.** Only the results of the top 5 teams are shown; the color code represents their position in the ranking (blue is the highest, red the lowest). Predictions for the frequency of time spent with a given diffusion type for EXP 9 (**a**), EXP 3 (**b**), EXP 4 (**c**), and EXP 5 (**d**) for the Single-trajectory Task of the Trajectory Track. Horizontal segments represent the ground truth. Inset of (**a**) predicted average residence time for the trapped and non-trapped states. The non-trapped state includes segments corresponding to both unconstrained (free/anomalous) diffusion and directed motion. Source data are provided as a Source Data file.

rely on estimates of $K$ per segment and then apply GMM or k-means, they generally tend to over-fragment wide $K$ ranges, misrepresenting the overall distribution. The corresponding predictions for the distributions of $\alpha$ for these experiments are shown in Fig. 8f, g. For EXP 8, characterized by the absence of CPs and nearly flat distributions of $K$ and $\alpha$, most methods successfully captured the broad distribution of $\alpha$ (Fig. 8g). However, their predictions for $K$ (Fig. 8c) were often biased toward different ranges within the allowed support. In contrast, EXP 9 presented a population of short dwell times in the trapped state. Most methods successfully detected the occurrence of these events, as reflected in the $K$ distribution (Fig. 8d), but, with the exception of team I, failed to associate these events with the correct $\alpha = 0$ Fig. 8h.

We further point out that optimizing methods to provide high scores for the metrics of the competition did not always translate into more meaningful insights about the underlying physical processes. For instance, teams M, H, and O showed significant biases across all experiments when predicting the $K$ distribution but still achieved high rankings according to the metric in Eq. (3) (Supplementary Fig. 6). Moreover, accurately predicting the number of true states did not provide a clear advantage with this metric, as most top teams overestimated the number of states but carefully adjusted their relative weights to minimize differences with the ground-truth distribution.

## Results summary and take-home messages

**Robust changepoint detection.** Top single-trajectory methods (e.g., based on UNet3+[86]) consistently achieve over 95% accuracy in identifying segment boundaries, with only minor false-positive rates across all scenarios.

**Distinguishing confinement, immobilization, and anomalous diffusion.** Leading algorithms accurately classify segments arising from geometric constraints or anomalous dynamics. Only very short segments and exponents close to zero remain challenging, indicating minimal crosstalk between distinct diffusion mechanisms.

**Trajectory extraction is a bottleneck.** Video-Track performance lags the Trajectory Track by 10–30%, highlighting that linking and localization errors-not downstream analysis–drive most of the accuracy loss.

**Parameter estimation benefits from physical priors.** Incorporating known physical models may yield significant gains in changepoint detection, but separate estimation pipelines for $K$ and $\alpha$ result in only modest improvements in parameter accuracy.

**Dedicated ensemble approaches are needed.** Ensemble Task submissions rely on GMM or k-means clustering of per-trajectory outputs,

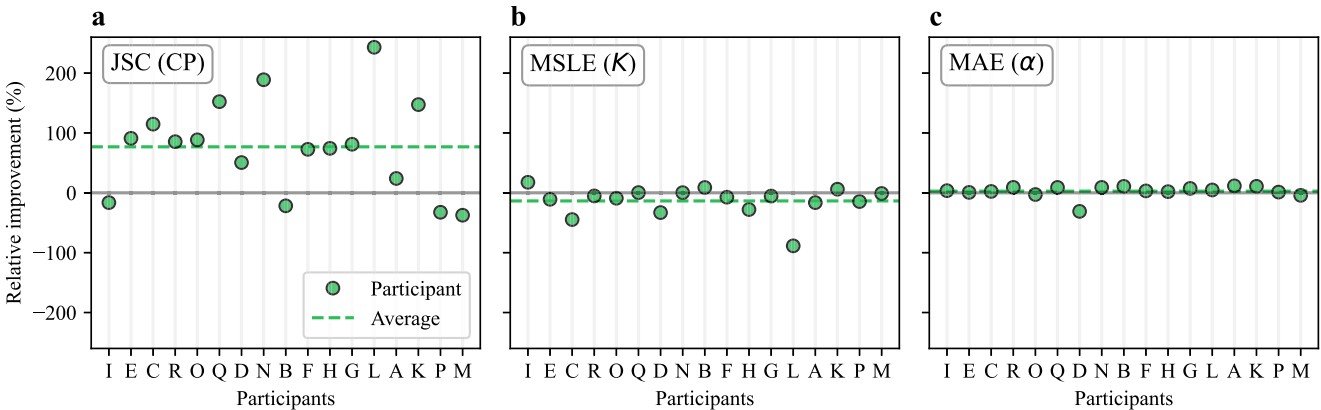

**Fig. 7 | Effect of the physical model.** Relative improvement for all participants when predicting trajectory properties from EXP 5 (DIM) with respect to EXP 6 (MSM) for the Single-trajectory Task of the Trajectory Track. Each panel reports the metrics associated with a different subtask: JSC (**a**), MSLE (**b**), and MAE (**c**). Source data are provided as a Source Data file.

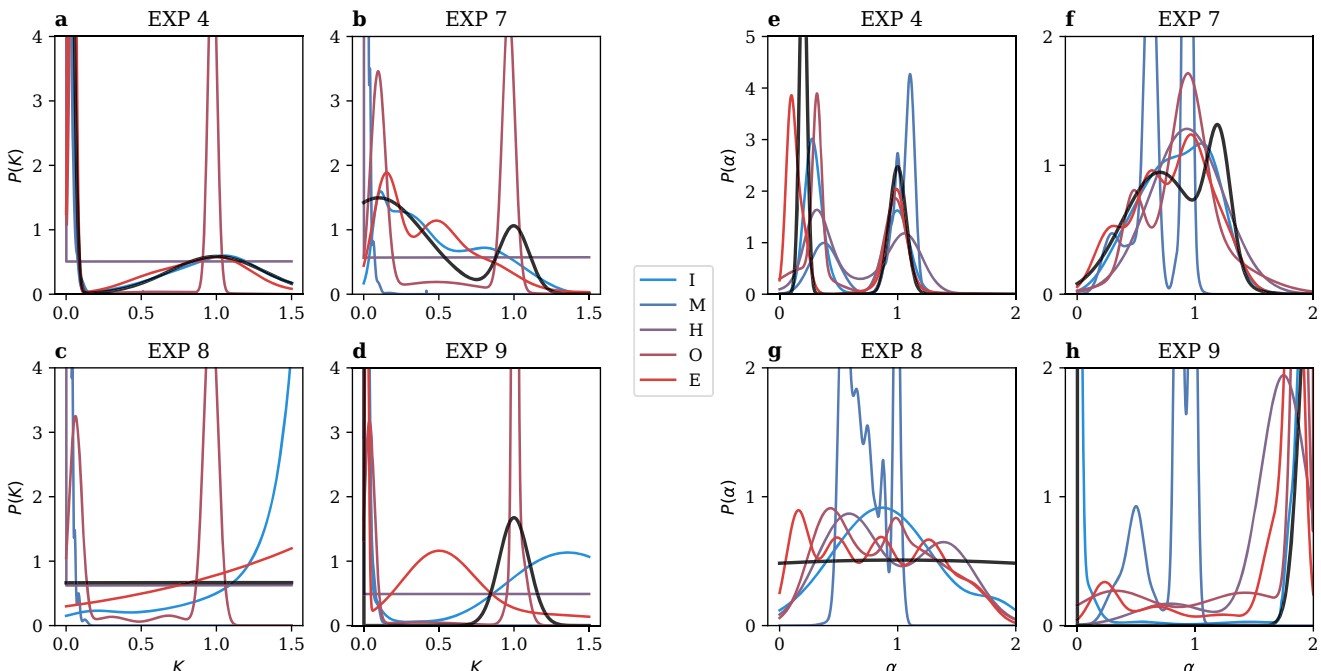

**Fig. 8 | Ensemble task predictions for the trajectory track. a–d** show the predicted distributions of the diffusion coefficient $K$, and **e–h** the anomalous exponent $\alpha$ for EXP 4, EXP 7, EXP 8, and EXP 9, respectively. Distributions were computed from the estimated means and variances (see Scoring and evaluation−Ensemble Task). Only the results of the top five teams are displayed, with color indicating rank (blue is the highest, red the lowest). Black curves denote the ground-truth distributions. Source data are provided as a Source Data file.

which fragments broad parameter distributions (e.g., EXP 7–8). Ensemble approaches, either bypassing single-trajectory clustering or using more sophisticated grouping techniques, hold potential for uncovering population-scale insights.

## Discussion

The 2nd AnDi Challenge provided a platform for advancing methods to characterize diffusion trajectories, with a special focus on those exhibiting transitions between distinct diffusive regimes. Through this Challenge, participants developed approaches that, when applied to standardized benchmarks, demonstrate robust capabilities in analyzing processes akin to those found in complex biophysical environments.

The high participation from teams spanning different fields vividly demonstrated the first rationale for the Challenge: the urgent need for standardized, rigorously evaluated methods to analyze dynamic changes in particle motion.

The Challenge highlighted several key insights. The methods for changepoint analysis have reached a good level of maturity. Participants demonstrated strong capabilities in detecting changepoints, which is crucial for understanding transitions between different diffusive regimes. However, the characterization of the resulting segments can still be improved. Accurate estimation of diffusion parameters within these segments remains challenging, particularly for short segments where the asymptotic nature of certain parameters, such as the anomalous diffusion exponent $\alpha$, complicates analysis. Sequence-to-sequence machine learning methods, mostly based on architectures combining convolutional[92] and transformer[93] layers, have shown great flexibility and effectiveness. The top-performing methods often utilized these architectures, highlighting their potential

for further advancements in the field. Notably, the methods did not take into account information coming from common parameters shared among trajectories or the underlying physical processes. Incorporating this knowledge could enhance the accuracy and robustness of the analyzes.

Nevertheless, significant challenges remain, and we hope the Challenge will help pave the way toward their resolution. In particular, we highlight two promising new avenues, which we believe may have a great impact on our understanding of the physics underlying biophysical processes.

First, the precision with which we can extract diffusion parameters remains fundamentally limited by current tracking algorithms, directly highlighting the third rationale for the Challenge. Notably, all participants in the Video Track relied on existing tracking techniques, subsequently applying the methods developed for the Trajectory Track to analyze the resulting trajectories. Despite the rapid advances in deep learning, none of the participants have yet leveraged these cutting-edge technologies to directly extract diffusive properties from video data. This missed opportunity could be attributed to several factors: the analysis technology may not yet be fully mature, the training processes might be too lengthy and complex, or the computational resources and time required could be prohibitively high. We foresee that as these bottlenecks are addressed, a new generation of methods will emerge, capable of bypassing the tracking step altogether and setting new standards of accuracy.

Second, in the Ensemble Task, all participants relied on post-processing of single-trajectory outputs. Features were first extracted from individual trajectories, and then a separate step was used to infer the parameters of the diffusive populations. No team developed dedicated ensemble-level algorithms or used established ensemble frameworks.

Although this single-trajectory-based approach produced high Challenge rankings, it offered limited biophysical insight due to the proliferation of predicted states and the instability of each mode's variance. Minimizing the Wasserstein-1 ($W_1$) distance aligns predicted and ground-truth distributions, but $W_1$ offers no penalty for over-splitting into numerous states or for unstable variance estimates, nor does it encourage physically interpretable solutions (e.g., filtering overlapping modes or very low-population segments). This warns us that outputs should not be blindly trusted when applied to real experiments. Care should always be taken not to overfit the data with too many states that cannot be assigned to a biophysical process. Whenever an analysis yields a large number of states, their identities should be validated through control experiments. In practice, a priori biological knowledge often narrows the expected state count, providing essential context for interpreting algorithmic results.

Looking ahead, methods capable of inferring population distributions directly from the raw ensemble of trajectories, thereby bypassing single-trajectory feature extraction and clustering, may deliver deeper physical insights. Moreover, approaches that treat the full set of trajectories contextually, rather than in isolation, are likely to enhance both performance and interpretability.

To encourage further development of methods addressing these issues, as well as those aligned with approaches used throughout the challenge, we have made the labeled dataset discussed in this work publicly available on Zenodo[94]. This resource allows researchers to benchmark new methods in a standardized manner, while also providing the experimental biophysics community with a tool to better identify the methods best suited to their specific experimental scenarios.

## Methods
### Simulations of diffusion and interaction models
Trajectories are simulated according to a 2-dimensional fractional Brownian motion (FBM)[54]. FBM is a continuous-time Gaussian process $B_H(t)$ with stationary increments and a covariance function $E[B_H(t)B_H(s)] = \frac{1}{2}(|t|^{2H} + |s|^{2H} - |t - s|^{2H})$, where $H$ represents the Hurst exponent and is related to the anomalous diffusion exponent $\alpha$ as $H = \alpha/2$[54]. FBM features three regimes: one in which the increments are positively correlated ($1/2 < H < 1$, i.e., $1 < \alpha < 2$, superdiffusive); one in which the increments are negatively correlated ($0 < H < 1/2$, i.e., $0 < \alpha < 1$, subdiffusive); and one in which the increments are uncorrelated ($H = 1/2$, i.e., $\alpha = 1$, diffusive Brownian motion).

The models included in the Challenge describe trajectories where diffusion properties are piecewise constant along segments of varying duration $T_s$ and undergo sudden changes. To obtain a trajectory segment of length $T_s$ with given anomalous diffusion exponent $\alpha$ and generalized diffusion coefficient $K$, a set of $T_s - 1$ displacements for each dimension is sampled from a fractional Gaussian noise generator[95]. The displacements are then standardized to have variance $\sigma^2 = 2K\Delta t$, where $\Delta t$ is the sampling time.

Simulations are performed considering particles diffusing in a square box of size $L$ with reflecting boundary conditions. However, to avoid boundary effects, the fields of view used for the Challenge datasets correspond to a square region of size $L_{FOV} \ll L$ within the central part of the original box (Fig. 2b).

For Track 1, trajectory coordinates are used as sub-pixel localizations of individual particles to simulate movie frames as in single-molecule fluorescence experiments[5]. Each particle has a random intensity $I_i$ that corresponds to the total number of photons collected by the detector. $I_i$ is drawn from a uniform distribution in the interval $[I_{min}, I_{max}]$ and fluctuates over time according to a normal distribution with mean $I_i$ and standard deviation $\sigma_I$. Each particle is rendered as a diffraction-limited spot using an Airy disk as a point-spread function (PSF) with full width at half maximum $FWHM_{PSF} = 2.1$ px. A constant background of $I_{bg} = 100$ counts is added to each frame. Images are corrupted with Poisson noise.

For Track 2, trajectory coordinates are corrupted with noise from a Gaussian distribution with zero mean and standard deviation $\sigma_N$ to take into account the finite localization precision obtained in tracking experiments. All simulated trajectories were generated without missing frames: no gaps were introduced, yielding continuous tracks to isolate segmentation performance from linking or gap-filling complexities.

All the models share a set of parameters required for the simulations that are described here. Model-specific parameters are defined when describing the details of the models in the following sections.

- $[K_1, K_2, ..., K_n]$: average values of the (Gaussian) distribution of the generalized diffusion coefficient for each of the $n$ diffusive states considered in a given experiment, with support $[10^{-12}, 10^6]$ pixel$^2$/frame$^\alpha$.

- $[\sigma_{K_1}, \sigma_{K_2}, ..., \sigma_{K_n}]$: standard deviations of the (Gaussian) distribution of the generalized diffusion coefficient for each of the $n$ diffusive states considered in a given experiment. If not provided, the standard deviation is considered to be equal to 0 (i.e., the distribution is $\delta(K - K_i)$).

- $[\alpha_1, \alpha_2, ..., \alpha_n]$: average values of the (Gaussian) distribution of the anomalous diffusion exponent for each of the $n$ diffusive states considered in a given experiment, with support $(0, 2)$.

- $[\sigma_{\alpha_1}, \sigma_{\alpha_2}, ..., \sigma_{\alpha_n}]$: standard deviations of the (Gaussian) distribution of the anomalous diffusion exponent for each of the $n$ diffusive states considered in a given experiment. If not provided, the standard deviation is considered to be equal to 0 (i.e., the distribution is $\delta(\alpha - \alpha_i)$).

- $L$: size of the box in which trajectories are simulated with reflecting boundary conditions.

- $L_{FOV}$: size of the box defining the FOV used for the Challenge datasets. The same particles can enter and exit the FOV over time

but, for evaluation purposes, they will be considered as generating different trajectories.

- $\Delta t$: sampling time at which the original motion of the particle is tracked. For the Challenge datasets, we consider $\Delta t = 1$.
- $T$: duration of the recording over each FOV, given as the number of time steps $\Delta t$. It also corresponds to the maximum trajectory duration. For the Challenge, we set $T = 200$;
- $T_{min}$: minimum duration of a trajectory to be included in the dataset. For the Challenge, we use $T = 20$;
- $I_{bg}$ (Track 1): background level of noise (counts) used in the simulation of videos.
- $\text{FWHM}_{PSF}$ (Track 1): full width at half maximum in pixels of the point-spread function used to render fluorescent particles.
- $I_{tot}$ (Track 1): mean value in counts of the total fluorescence collected for the detected particles.
- $\sigma_{tot}$ (Track 1): standard deviation in counts of the distribution of total fluorescence collected for the detected particles.
- $I_{peak}$ (Track 1): mean value in counts of the peak fluorescence collected for the detected particles. Can be calculated as $I_{peak} = I_{tot} \frac{4 \ln 2}{\pi \text{FWHM}_{PSF}^2}$
- SNR (Track 1): typical signal-to-noise ratio of the movies, calculated as the average peak intensity over the standard deviation of the noise[51] and thus equal to

$$\text{SNR} = \frac{I_{peak}}{\sqrt{I_{peak} + I_{bg}}}. \tag{15}$$

- $\sigma_N$ (Track 2): standard deviation of the Gaussian localization noise used to corrupt trajectory coordinates.
- $t_{min}$: minimum distance between changepoints, corresponding to the minimum amount of time that a particle spends in a state. Shorter segments are eliminated by smoothing the time trace of the state label using a majority filter with a window of 5 steps. For the Challenge, we set $t_{min} = 3$ frames to test the sensitivity and robustness of the segmentation methods under minimal data conditions.

A schematic representation of each of the models presented below is shown in Fig. 2a.

**Model 1 - Single-state model (SSM).** This model simply corresponds to particles diffusing according to FBM with constant generalized diffusion coefficient $K$ and anomalous diffusion exponent $\alpha$. For each trajectory, a value of $K$ and a value of $\alpha$ are sampled from the corresponding distribution. Data corresponding to these models are necessary to establish the false positive rate of the methods toward the detection of changes of diffusion properties.

**Model 2 - Multi-state model (MSM).** The multi-state model is a Markov model describing particles undergoing FBM whose diffusion properties can change at random times. The number of states $S$ is fixed for a given experiment, as are the parameters defining the distributions of $K$ and $\alpha$ for each state. For each trajectory, $S$ values of $\alpha$ and $S$ values of $K$ are sampled from the distribution of the corresponding states, i.e., one per state. At every time step, a diffusing particle has a given probability to undergo a change in one of its diffusive parameters (either $\alpha$ or $K$). The probability of switching is given by a transition matrix $M$. Namely, $M_{ij}$ is the probability of switching from state $i$ to state $j$ at each time step. In the same sense, $M_{ii}$ is the probability of remaining in state $i$. The residence time in a given state $i$ can be directly

calculated from the previous probability as

$$\tau_i = \frac{1}{\sum_{j \neq i} M_{ij}} = \frac{1}{1 - M_{ii}}. \tag{16}$$

**Model 2 (MSM) parameters.**
- $M$: transition matrix between diffusive states.

**Model 3 - Dimerization (DIM).** This model considers the case in which dimerization, i.e., the transient binding of two particles, may occur and produce changes in the diffusion properties of both particles. In particular, we consider the case of $N$ circular particles of radius $r$. For each trajectory, a value of $\alpha$ and a value of $K$ are sampled from the corresponding distributions associated with the monomeric state. If two particles are at a distance $d < 2r$, then they have a probability $P_b$ of binding. The two particles forming a dimer move with equal displacements, according to a generalized diffusion coefficient $K$ and an anomalous diffusion exponent $\alpha$ drawn from the distributions associated with the dimeric state. At each time step, the dimer has a probability $P_b$ of breaking its bond, freeing the two particles to go back to their original motion parameters. The particles cannot form any new dimers until taking a new step. Only dimers are allowed, and subsequent hits with other particles will not affect either the particles or the dimers.

**Model 3 (DIM) parameters.**
- $N$: number of diffusing particles in the box of size $L$.
- $r$: interaction radius, corresponding to the radius of the diffusing particles.
- $P_b$: probability that two particles bind to form a dimer in each time step. For this to happen, the particles must be at a distance $d < 2r$.
- $P_u$: probability that a dimer breaks up at each time step so that the two particles go back to diffusing independently.

**Model 4 - Transient-confinement model (TCM).** This model considers an environment with $N_c$ circular compartments of radius $r_c$. The compartments are distributed randomly throughout the environment such that they do not overlap. We consider that the compartments are *osmotic*, i.e., a particle reaching their boundary from the exterior has a probability 1 of entering them, but a particle reaching the boundary from the interior of a compartment has a probability $T$ of exiting it (and $1 - T$ of being reflected back to the interior of the compartment). The diffusion inside and outside the compartment is different, hence defining two diffusive states. For each trajectory, two values of $\alpha$ and two values of $K$ are sampled from the corresponding distributions, representing the motion outside and inside the compartments.

**Model 4 (TCM) parameters.**
- $N_c$: number of compartments in the box of size $L$.
- $r_c$: radius of the compartments.
- $T$: transmittance of the boundary. Probability that a particle reaching the boundary from inside the compartment exits the compartment.

**Model 5 - Quenched-trap model (QTM).** This model considers the diffusion of particles in an environment with $N_t$ immobile traps of radius $r_t$. The values of $\alpha$ and $K$ are sampled for each trajectory from the corresponding distributions and define its unrestrained motion. A particle that enters the domain defined by a trap has a probability $P_b$ of binding to the trap and, hence, getting temporarily immobilized ($K = 0$, $\alpha = 0$). At each time step, a trapped particle has a probability $P_u$ of unbinding and being released from the trap, going back to its unrestrained motion. A particle cannot be trapped again until taking a new step.

**Model 5 (QTM) parameters.**
- $N_t$: number of traps in the box of size $L$.
- $r_t$: radius of the traps.
- $P_b$: probability that a particle binds to a trap and gets immobilized. For that to happen, a particle must be at a distance $d < r_t$ from the trap.
- $P_u$: probability that a trapped particle unbinds from a trap and starts diffusing independently at each time $\delta t$.

## Dataset structure

The datasets used in the Challenge (Supplementary Fig. 9) include different experiments, each contained in a folder labeled with a sequential number (EXP_[exp number]) and corresponding to a specific model and a fixed set of parameters. The information about the model and the parameters is unknown to Challenge participants. Each experiment folder contains a list of files labeled with a sequential number (FOV_[fov number]) associated with 30 FOVs. Each FOV reports data from a variable number of particles diffusing on a $128 \times 128$ pixel$^2$ area.

For the Video Track, the coordinates of the particles in the same FOV are used to generate 200-frame videos as a series of 8-bit images in the multi-tiff format using Deeptrack 2.1[5]. Noise is added to the synthetic images to account for background fluorescence and shot noise. A map corresponding to the segmentation of VIP particles at the first frame for which CPs and diffusion parameters must be detected is also provided as a TIFF file. Connected components of the map are labeled with unique integer values that correspond to the particle index.

For the Trajectory Track, we provide a CSV file for each FOV with a table whose columns contain trajectory index, time step, $x$-coordinate, and $y$-coordinate. Coordinates of simulated trajectories are corrupted with Gaussian noise corresponding to finite (subpixel) localization precision. The trajectories have a maximum length of 200 frames.

Besides localization precision, motion blur can introduce a significant contribution to noise, in particular if the camera frame rate is slow compared to particle motion[96]. However, this aspect will not be included in the Challenge datasets since it would introduce complexities in the definition of the ground truth that could detract from the focus of the work. Nevertheless, the simulation software incorporates the capability to introduce the effect of motion blur both in videos and trajectories.

Exemplary data for all the models are shown in Supplementary Fig. 10. Files in different Tracks labeled with the same experiment and the FOV index (e.g., Track_1/EXP_4/FOV_3.tiff and Track_2/EXP_4/FOV_3.csv) include simulations obtained with the same set of dynamics parameters but do not correspond to the motion of the same set of particles.

## Protocol registration

The Stage 1 protocol for this Registered Report was accepted in principle on 31st October 2023. The protocol, as accepted by the journal, can be found at https://doi.org/10.6084/m9.figshare.24771687.v1.

### Reporting summary

Further information on research design is available in the Nature Portfolio Reporting Summary linked to this article.

## Data availability

The labeled benchmark dataset used in this study is available on Zenodo[94]. All datasets generated for the Challenge can be accessed on the Codalab platform (registration required). Source data for all figures are provided with this paper. Source data are provided with this paper.

## Code availability

All code used to generate the Challenge datasets is publicly available via the andi_datasets repository on GitHub: https://github.com/AnDiChallenge/andi_datasets[71].

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

## Acknowledgements

The authors would like to thank the other participants of the 2nd AnDi Challenge: Thomas Martynec, Sarah A.M. Loos; Maxime Lavaud, Juliette Lacherez, Yosef Shokeeb, Yacine Amarouchene, Thomas Salez; Roman Lavrynenko, Lyudmyla Kirichenko, Sophia Lavrynenko; Taegeun Song, Seunghee Han, Jaehyun Jeong, Jihye Kim; Farzaneh Nazari, Mohammad Mehdi Nazari; Janusz Szwabiński, Jakub Malinowski, Marcin Kostrzewa, Michał Balcerek, Weronika Tomczuk; Alvaro Lanza, Stefano Bo; Raffaele Pastore, Francesco Rusciano, Maurizio De Micco, Pier Luca Maffettone, Francesco Greco. The organizers of the 2nd AnDi Challenge acknowledge the CHAIR (Chalmers AI Research Center) Research Area AISDA (AI for Scientific Data Analysis) and the EUTOPIA Connected Community on BioImaging for sponsoring the final workshop, and thank Agnese Callegari for her invaluable help with its organization. G.M-G. acknowledges support from the European Union. S.A. and Gior.V. are grateful for the studentship funded by the A*STAR-UCL Research Attachment Program through the EPSRC M3S CDT (EP/L015862/1). Gior.V. acknowledges support for this work by The Chan Zuckerberg Initiative, "Multi-color single molecule tracking with lifetime imaging" (2023-321188). R.N. acknowledges support from the Academic Research Fund from the Singapore Ministry of Education (RG151/23 and MOE2019-T2-2-010) and the National Research Foundation, Singapore, under its 29th Competitive Research Program (CRP) Call (NRF-CRP29-2022-0002). Z.H. acknowledges the support from the National Natural Science Foundation of China (Grant No. 12104147) and the Fundamental Research Funds for the Central Universities. X.F. and Y.Z. Acknowledge grants from the National Natural Science Foundation of China (grant nos. 62031023 and 62331011). J.A.C. is supported by the European Union—NextGenerationEU, ANDHI project CPP2021-008994 and PID2021-124618NB-C21, by MCIN/AEI/10.13039/501100011033, and by "ERDF A way of making Europe", from the European Union. J.K.H., S.W.B.B., and N.S.H. acknowledge the Novo Nordisk Foundation Challenge Center for Optimized Oligo escape (NNF23OC0081287). H.J. and J.B. acknowledge support from the Basic Science Research Program through the National Research Foundation of Korea (RS-2025-00514776). R.H. acknowledges the Medical Sciences Doctoral Training Centre, University of Oxford for financial support. M.L., G.F-F. and B.R. acknowledge support from: European Research Council AdG NOQIA; MCIN/AEI (PGC2018-0910.13039/501100011033, CEX2019-000910-S/10.13039/501100011033, Plan National FIDEUA PID2019-106901GB-I00, Plan National STAMEENA PID2022-139099NB, I00, project funded by MCIN/AEI/10.13039/501100011033 and by the "European Union NextGenerationEU/PRTR" (PRTR-C17.I1), FPI); QUANTERA DYNAMITE PCI2022-132919, QuantERA II Program co-funded by European Union's Horizon 2020 program under Grant Agreement No. 101017733; Ministry for Digital Transformation and of Civil Service of the Spanish Government through the QUANTUM ENIA project call - Quantum Spain project, and by the European Union through the Recovery, Transformation and Resilience Plan - NextGenerationEU within the framework of the Digital

Spain 2026 Agenda; Fundació Cellex; Fundació Mir-Puig; Generalitat de Catalunya (European Social Fund FEDER and CERCA program); Barcelona Supercomputing Center MareNostrum (FI-2023-3-0024); (HORIZON-CL4-2022-QUANTUM-02-SGA PASQuanS2.1, 101113690, EU Horizon 2020 FET-OPEN OPTOlogic, Grant No. 899794, QU-ATTO, 101168628), EU Horizon Europe Program (This project has received funding from the European Union's Horizon Europe research and innovation program under grant agreement No. 101080086 NeQST); ICFO Internal "QuantumGaudi" project; Funded by the European Union. Views and opinions expressed are, however, those of the author(s) only and do not necessarily reflect those of the European Union, European Commission, European Climate, Infrastructure and Environment Executive Agency (CINEA), or any other granting authority. Neither the European Union nor any granting authority can be held responsible for them. R.M. acknowledges DFG grants ME 1535/16-1 and 1535/22-1. D.K. acknowledges funding from the National Science Foundation grant 2102832. Giov.V. acknowledges support from the Horizon Europe ERC Consolidator Grant MAPEI (grant number 101001267) and the Knut and Alice Wallenberg Foundation (grant number 2019.0079). C.M. acknowledges support through grant RYC-2015-17896 funded by MCIN/AEI/10.13039/501100011033 and "ESF Investing in your future", grants BFU2017-85693-R and PID2021-125386NB-I00 funded by MCIN/AEI/10.13039/501100011033/ and "ERDF A way of making Europe".

## Author contributions

G.M.-G., M.L., R.M., D.K., Giov.V. and C.M. conceived the study. G.M.-G., Giov.V. and C.M. organized the challenge and the corresponding workshop. G.M.-G., H.B., J.Pi. and B.M. designed and implemented the software for data generation. G.M.-G., J.Pi. and C.M. implemented the platform for scoring. G.M.-G. and C.M. analyzed the results. The methods discussed in the paper were designed, implemented, run, and described by the Challenge participants: G.F.-F., B.R., Y.A. S.A., J.B., F.J.B. S.W.B.B., C.C., J.A.C., M.E., X.F., R.H., N.S.H., Z.H., I.I., H.J., Y.J., J.K-H., J.M.-H., R.H., J.Pa., X.Q., L.A.S., H.S., N.S., Y.Z., Gior.V. The article was written by G.M.-G., M.L., R.M., D.K., Giov.V., and C.M. with input from all authors.

## Funding

## Competing interests

The authors declare no competing interests.

## Additional information

[1]Institute for Theoretical Physics, University of Innsbruck, Innsbruck, Austria. [2]Department of Physics, University of Gothenburg, Origovägen 6B, SE-41296 Gothenburg, Sweden. [3]ICFO – Institut de Ciències Fotòniques, The Barcelona Institute of Science and Technology, Av. Carl Friedrich Gauss 3, 08860 Castelldefels (Barcelona), Spain. [4]Instituto Universitario de Matemática Pura y Aplicada, Universitat Politècnica de València, València, Spain. [5]Department of Chemistry, University College London, 20 Gordon Street, London WC1H 0AJ, UK. [6]Department of Physics, Korea Advanced Institute of Science and Technology, Daejeon, Korea. [7]Molecular Neurobiology Division, BIOMED UCA-CONICET, Buenos Aires, Argentina. [8]Department of Chemistry, University of Copenhagen, Copenhagen, Denmark. [9]Novo Nordisk Center for Optimised Oligo Escape and Control of Disease, University of Copenhagen, Copenhagen, Denmark. [10]Institut Langevin, ESPCI Paris, Université PSL, CNRS, Paris, France. [11]Centro de Investigación en Gestión e Ingeniería de Producción, Universitat Politècnica de València, València, Spain. [12]School of Computer Science and Technology, Harbin Institute of Technology (Shenzhen), Shenzhen, China. [13]Gene Machines Group, Clarendon Laboratory, Department of Physics, University of Oxford, Oxford, UK. [14]Kavli Institute of Nanoscience Discovery, University of Oxford, Oxford, UK. [15]School of Physics and Electronics, Hunan University, Changsha, China. [16]Center of Complex Systems, Korea Advanced Institute of Science and Technology, Daejeon, Korea. [17]Laboratory of Computational Quantitative and Synthetic Biology (CQSB), Sorbonne Université, CNRS, Paris, France. [18]School of Chemistry, Chemical Engineering and Biotechnology, Nanyang Technological University, Singapore, Singapore. [19]ICREA, Pg. Lluís Companys 23, 08010 Barcelona, Spain. [20]Institute for Physics & Astronomy, University of Potsdam, Potsdam-Golm, Germany. [21]Asia Pacific Centre for Theoretical Physics, Pohang, Republic of Korea. [22]Department of Electrical and Computer Engineering and School of Biomedical Engineering, Colorado State University, Fort Collins, CO, USA. [23]Science for Life Laboratory, Physics Department, University of Gothenburg, Origovägen 6B, SE-41296 Gothenburg, Sweden. [24]Facultat de Ciències, Tecnologia i Enginyeries, Universitat de Vic—Universitat Central de Catalunya (UVic-UCC), C. de la Laura, 13, 08500 Vic, Spain. [25]Bioinformatics and Bioimaging, Institut de Recerca i Innovació en Ciències de la Vida i de la Salut a la Catalunya Central (IRIS-CC), 08500 Vic, Spain. ✉e-mail: gorka.munoz-gil@uibk.ac.at; giovanni.volpe@physics.gu.se; carlo.manzo@uvic.cat

