## [Transparent Peer Review file · Nature Communications]

Quantitative evaluation of methods to analyze motion changes in single-particle experiments

Corresponding Author: Professor Giovanni Volpe

Version 0:

Reviewer comments:

Reviewer #1

(Remarks to the Author)

This manuscript describes a competition to quantitatively compare methods that analyze motion changes in single-molecule tracking experiments. This competition is a modified repeat of a previous competition that the authors organized in 2021, called the first AnDi Challenge. Such challenges are important in that they bring together members of diverse research communities (experimentalists, theoreticians, data analysts, and computer scientists). Three reasons were given to rationalize the running a second AnDi challenge: First, there is a need to evaluate methods that determine the switch between different diffusive behaviors within single-molecule trajectories. Second, it is necessary to assess the methods' crosstalk in detecting inherent anomalous diffusion due to motion constraints. Third, there is a need to determine whether the bottleneck of the analysis process is at the level of the analysis of the single trajectories or associated with their extraction from the experimental videos.

The proposed second AnDi challenge will yield useful information for the field, if the following two criteria can be met: First, anomalous diffusion behaviors need to be clearly attributed to biological effects, such as molecular interactions or molecular confinement to certain subcellular regions. The simulated challenge data should aim to match the time and length scales of these biological effects and accurately represent experimental data (in terms of localization error or factors that contribute to localization errors, in particular). Second, it should be ruled out that observed anomalous diffusion behaviors can instead be attributed to how the data was acquired. This is particularly important when 3D diffusive motion is measured as a 2D (projected) trajectory. Consider, for example, a membrane protein diffusing freely in the membrane of a rod-shaped bacterial cell. If that membrane is pressed flat against the coverslip, then diffusion might appear normal ($\alpha=1$). On the other hand, if the same trajectory occurs on the side wall of the bacterial cell, then diffusion perpendicular to the long axis of the cell will appear highly confined (resulting in $\alpha<1$ in that dimension). A similar example could be made for directed motion along a cytoskeletal filament that is oriented along the z-axis of the microscope. In such a case, the 2D-projected motion would appear highly confined in both the x- and y- dimension ($\alpha<1$), while the actual motion is in fact superdiffusive ($\alpha>1$). The manuscript, in its current form, does not clearly disentangle these two effects. A clear connection between how the challenge data is simulated and biological effects resulting in anomalous diffusion is missing and the challenge data is only in the form of 2D data, which seems to restrict the scope of the proposed work to diffusion within flat membranes. The expected challenge results would be of higher significance and broader in scope, if 3D diffusion were included and the different contributions to experimentally observed anomalous diffusion could be clearly distinguished.

The following specific comments will hopefully prove useful for improving the manuscript.

Specific Comments:

1. Page 2: "There are three heterogeneity classes that these methods aim to identify: (i) changes in the value of the diffusion coefficient D ; (ii) changes in the anomalous diffusion exponent (often classified as subdiffusion, diffusion, or superdiffusion); and (iii) changes in the phenomenological behavior associated with interactions with the environment (often classified as immobilization, confinement, (free) diffusion, and directed motion)."

o Points 1 and 3 make immediate sense from a biological perspective.

Point 2 will be of limited interest to cell biologists, unless clear (biological) rationale for this type of motion can be given. That

rationale should also include the relevant time and length scales for a given motion type.

2. Page 2: "In the following, we will refer to the anomalous exponent α as the characteristic feature of the generating motion."

o It is clear that mathematically the anomalous exponent can be changed, ...BUT what are the biological processes that would change α other than immobilization, confinement, and directed motion? Immobilization and directed motion are readily detectable in single-molecule trajectories without the employment of sophisticated analysis algorithms. Please provide some specific biological examples (with relevant time and length scales) where transient confinement could influence experimentally measured single-molecule trajectories in living cells. (Experimentally measured single-molecule trajectories can cover the 200 us – 2s timescales; trajectories typically contain less than 100 displacements (the majority contain 10-20), and localization precisions range from ~20 to 200 nm depending on how fast emitters diffuse and the frame time with which images were acquired.)

3. Page 2,3: "we will assess the methods' performance on simulated datasets designed according to biologically inspired models of diffusion and interactions in typical experimental conditions. The participating methods will be further compared on real biological datasets obtained from particles moving in the cell nucleus, cytoplasm, and plasma membrane."

o The conceptual connection between the simulated datasets and what would/will be observed in live-cell single-molecule tracking experiments is not clear

4. Page 3: "Motion of the FBM-type has been widely observed in biological experiments [7–9] and is often used to describe dynamics in viscoelastic environments [29]."

o Same critique as in Comment #2. I don't think the reference given here provide a sufficient rationale of why it is worthwhile, from a biological perspective, to distinguish different FBM motion types.

o The cytoplasm and artificial dextran solutions have been shown to exhibit viscoelastic properties, when very large particles are used for tracking. What are biological mechanisms that could lead to anomalous diffusion for small proteins and protein complexes (<200 kDa)?

5. Page 3: "Multi-state model (MSM) — Particles diffusing according to a time-dependent multi-state (2 or more) model of diffusion, as observed for example in proteins undergoing transient changes of K and/or α , as induced by, e.g., allosteric changes or ligand binding [32, 33]."

o It makes sense that allosteric changes and ligand binding would change K (slightly), but why would it change α ? Please provide a biological example.

6. Page 3: "Dimerization model (DIM) — Particles diffusing according to a 2-state model of diffusion, with transient changes of K and/or α induced by encounters with other particles, observed for example in protein dimerization [34, 35]."

o It makes sense that dimerization would change K , but why would it change α ? Please provide a biological example.

7. Page 3: "Transient-confinement model (TCM) — Particles diffusing according to a space-dependent 2-state model of diffusion, observed for example in proteins being transiently confined in regions where diffusion properties might change, e.g., the confinement induced by clathrin-coated pits on the cell membrane [36]. In the limit of a high density of trapping regions, this model reproduces the picket-and-fence model used to describe the effect of the actin cytoskeleton on transmembrane proteins [37, 38]."

o The justification of the TCM is stronger overall, compared to the previous two models. Please also provide the relevant length and time scales of confinement in clathrin-coated pits. Are these readily measurable with single-molecule tracking methods?

o Same question for the picket-and-fence model: What are the relevant length and time scales of confinement and are these experimentally accessible using single-molecule tracking?

8. Page 3: "Quenched-trap model (QTM) — Particles diffusing according to a space-dependent 2-state model of diffusion, representing proteins being transiently immobilized at specific locations as induced by binding to immobile structures, such as cytoskeleton-induced molecular pinning [39]."

o What are the relevant length and time scales of cytoskeleton-induced molecular pinning and are these experimentally accessible using single-molecule tracking?

9. Page 3: "Datasets provided for the last phase of the competition will also include results from actual experiments corresponding to the motion of particles in living cells. These experiments have reported the occurrence of heterogeneous diffusion as a consequence of specific interactions, conformational changes, or underlying structures, but the ground truth is not established beyond any doubt. Therefore, these data will not contribute to the challenge scoring. Nevertheless, the predictions provided by different methods will be comparatively analyzed and discussed also with respect to the conclusions reached in the original publications."

o Are these datasets intentionally withheld from the participants?

o If these are already known/published, then they could be used to address my previous comments.

10. Page 5: "Each FOV reports data from a variable number of particles diffusing on a 128 x 128 pixel² area."

o Does that mean that only 2D diffusion is modeled?

o This would exclude many processes mentioned earlier, such as diffusion in the cytoplasm or within the nucleus.

o This raises a bigger question for me: Is anomalous diffusion observed, simply because 3D diffusive motion is measured as a 2D projected trajectory? In other words, is anomalous diffusion observed because of how the data were acquired and not

because of biological reasons?

11. Page 5: "For Track 1 (Fig. 3(a)), the coordinates of particles in the same FOV are used to generate 200-frame videos as a series of 8-bit images in the multi-tiff format using Deeptrack 2.1 [5]."

- o Are motion-blurring effects included in Deeptrack? The description in Appendix A suggests that it is not.
- o Localization errors in single molecule tracking are due to finite fluorescent photon numbers (static localization errors) and motion-blur effects (dynamic localization errors). Therefore, localization errors for fast-moving emitters are worse than those of slowly moving emitters. These factors should be modeled as realistically as possible.

12. Page 5: "Coordinates of simulated trajectories are corrupted with Gaussian noise corresponding to finite (subpixel) localization precision."

- o Same comment as Comment #11: Are motion-blurring effects included here?

Reviewer #2

(Remarks to the Author)

Attached

Version 1:

Reviewer comments:

Reviewer #1

(Remarks to the Author)

Summary

While the authors have attempted to address my comments in their rebuttal, I do not believe the edits in the manuscript are sufficient given my concerns. My main concern is as follows: As the authors state themselves, "the Challenge primarily centers around data from biological systems", and they make references to ligand binding and dimerization. For such molecular systems, anomalous diffusion is, to my knowledge, not established. This is in contrast to the field of microrheology, where researchers track tracer particles that are orders of magnitude larger than single proteins and protein complexes. In microrheology, anomalous diffusion is well documented and has been attributed to, e.g. the viscoelasticity of the cytoplasm, which becomes more pronounced at larger lengths scales. Unfortunately, there is MUCH confusion in the field about the differences between single (tracer) particle tracking (microrheology) and bona fide single-molecule tracking of small proteins and protein complexes. It would undoubtedly be beneficial for readers if the authors clearly distinguished between single tracer particle tracking (microrheology) and actual single-molecule tracking and make it clear that the methods should not be applied, without clear biological justification, to the diffusion of molecules in cells. I would also like to see a complete discussion of why anomalous effects may be appearing in tracking data. As highlighted in my initial peer-review comments and acknowledged by the authors in the rebuttal letter, anomalous diffusion may be an artifact due to how the data were acquired (e.g. 3D diffusive motion measured as a 2D (projected) trajectory). These critiques are also in line with Reviewer 2, who remarked that "It is not evident that your methods "guide" researchers to identify optimal tools for analyzing their experiments. ... How do experimentalists know that their data are related to FBM?"

If the necessary modifications are not introduced into the manuscript, my worry is that a myriad of inexperienced researchers will, upon reading it, simply apply the winning algorithm on their 2D single-molecule trajectories without sufficient considerations of why this algorithm may or may not be appropriate. I can appreciate that the authors want the competitors to address a general problem (formulated in generalized and later scalable length and time scales). I can also appreciate that that anomalous diffusion may play an important role in other fields (finance, bird movements, etc.) – although I am not an expert in those fields. If the authors want this manuscript to focus primarily on molecular and cellular biological systems (which it currently is), then they have the responsibility to clearly highlight the limitations of the challenge data and clearly delineate for which systems (i.e. microrheological motion, NOT single-molecule motion) the algorithms are appropriate.

Below are my responses to the authors' rebuttal comments and changes made in the manuscript.

Specific Comments:

1. Reply to R1.3 "... our discussions led us to conclude that the inclusion of 3D examples should be avoided and might be even misleading for the following reasons:

a. The number of studies on 2D diffusion in nearly flat membranes is significantly larger compared to 3D diffusion, primarily due to the simplicity of the technical approach and data interpretation.

Response: It is true 2D diffusion measurements are more numerous, perhaps due it being a technically simpler experimental approach. However, data interpretation is NOT simpler. In fact, it can be substantially complicated due to the potential of artifacts that lead to the appearance of anomalous diffusion. This needs to be made clear in the manuscript.

b. Live-cell single-molecule imaging provides indicators, such as changes in the diffraction pattern of fluorophores, that suggest the necessity of using 3D techniques to prevent misinterpretation of results, as exemplified by the reviewer.

Response: It needs to be made clear in the manuscript when researcher should not measure (projected) 2D trajectories. I am not quite sure what the authors mean by "changes in the diffraction pattern of fluorophores". Please elaborate.

c. Numerous experimental efforts have been made to track biological objects moving in 3D using alternative techniques to live-cell single-molecule imaging. However, these techniques yield different outputs and provide additional information to understand these processes. For cases like the one described by the reviewer, these alternative techniques should be preferred.

Response: I am unclear which methods the authors refer to here. Please clarify. These methods are not mentioned in the manuscript either. If they are to be preferred to tracking approaches, they should be mentioned.

d. The number of possible 3D geometries that can generate apparent deviations from Fickian diffusion is very large, making it unfeasible to consider all of them for the challenge. Selecting only a subset would be arbitrary and provide limited informative value.

Response: True, but researchers still need to be made aware of this issue. Otherwise, any sophisticated data analysis could be meaningless, because of unconsidered artifacts in the data.

To address Comment R1.3, the authors added the following text to the introduction: "In several biological settings, motion takes place on non-flat surfaces or 3D space but only a 2D projection is recorded when using techniques such as live-cell single-molecule imaging. Unless the isotropy of space and independence of motion along the three dimensions are known, the inference of 3D diffusion properties from 2D projections can generate artifacts. Therefore, we opted to focus the challenge on well-documented cases of diffusion changes in 2D. To track biological objects moving in 3D, alternative experimental techniques that provide additional information should be preferred to properly characterize these processes."

Response: This added text is VERY brief and should be expanded on to address my concerns listed above. Please clarify what is meant by "well-documented cases of diffusion changes in 2D" and "alternative experimental techniques".

2. In Response to Comment R1.4, the authors added the following text to the introduction: "While changes in the diffusion coefficient and in the phenomenological behavior have been widely reported, the exploration of changes in the anomalous diffusion exponent is a more recent development [17]. The introduction of new methods, along with a robust evaluation of their performance, could facilitate a more comprehensive assessment of this behavior."

Response: ...which would enable what exactly? Please add one more sentence to complete the thought for the reader – perhaps by giving a tangible, concrete example based on the publications listed by the authors.

The authors state in their rebuttal that "...we believe that the quantitative assessment of method performance in detecting changes in the anomalous diffusion exponent, as well as analyzing data with timescales that replicate dynamics in non-biological systems, holds significant value and relevance." The revised text should convince the reader of this rather than just listing potential applications without much context.

3. In Response to Comment R1.7, the authors gave a list of references meant to establish Motion of the FBM-type in real biological systems. These references are however not focused on motion of small proteins and protein complexes (which is what my comment referred to), but they instead focus on the motion of endocytic organelles (Han et al, eLife 9, e52224 (2020)), multicellular spheroids (Revery et al, Sci Rep 5, 11690 (2015)), dynamics of large tracer particles in an actin network (Levin et al, J Chem Phys 154, 144901 (2021)), and rhodamine 6G molecules in water nanofilms of a few (one to eight) molecular layers on glass coverslips as well as large tracer particles in mammalian cell, and macroscopic objects (Vilk et al, Phys Rev Res 4, 033055 (2022)).

Response: None of the above examples cover the motion of small proteins and protein complexes.

If no clear experimental evidence for anomalous diffusion of molecules in living cells can be referred to by the authors, then the manuscript should be revised to remove any overly broad and thus misleading statements and references claiming that anomalous diffusion behavior is "well documented" in "real biological systems" or in "biological environments" in general. Specifically, the language needs to be more precise overall and a distinction needs to be made, for the benefit of inexperienced readers, between single tracer particle tracking (microrheology) and actual single-molecule tracking of proteins and protein complexes. The dynamics of these diffusers are NOT the same based on the experimental evidence available to date.

In accordance with that fact, I appreciate that the authors changed the text in response to my comment R1.8. I do believe however that such edits need to be made where applicable, throughout in the manuscript.

4. In Response to Comment R1.12, the authors state that "As discussed above, we decided to focus the Challenge on 2D diffusion. This excludes diffusion in the cytoplasm or in the nucleus, which we believe should be better addressed using 3D tracking approach to prevent artifacts in data interpretation, as in the example described by the reviewer."

Response: I do believe that remains a weakness of the manuscript/proposed work. If the authors decide to stay with 2D diffusion only (due to it being the more abundant type of measurement), then a more complete discussion is needed to address whether anomalous diffusion is observed because of how the data were acquired and not because of biological reasons. Making a single fleeting mention to the potential of artifacts is not sufficient, in my opinion, as it would be easily overlooked by many readers. Specific use cases that are not appropriate for the tested methods should be specifically mentioned with sufficient context and rationale of why they are beyond the scope of this study. It should also be made clear that it is the future researcher's responsibility to make sure that the assumption made in applying the methods that are included in the competition are satisfied for their respective application. This is necessary to guarantee the quality of research that will build on the proposed competition results.

(Remarks to the Author)

The results of AnDi Challenge 1, which aimed to develop state-of-the-art methods for analysing diffusion and comparing them with each other, were published in the journal Nature Commun, 12, 6253 (<https://doi.org/10.1038/s41467-021-26320-w>). This submission is devoted to AnDi Challenge 2. The analysis of methods performance is also successful as a previous one. The datasets used in the challenge include different experiments, but it is fixed to a specific model and a given set of parameters. The performance of existing and novel methods is estimated using special metrics. I believe that the software library created in this work will be useful for a wide circle of readers.

This work has been carried out thoroughly. It can identify two classes of methods: (i) ensemble methods and (ii) single-trajectory methods. Since the method for each specific problem is chosen from researcher's preference, a variety of simulations facilitate this choice. The combination of experimental data analysis with simulations makes this work meaningful.

I am satisfied with the answers of the authors to my referee report. The authors have significantly improved their presentation. I recommend this manuscript to publication in Nature Communications.

Version 2:

Reviewer comments:

Reviewer #1

(Remarks to the Author)

The authors have sufficiently addressed my concerns and I recommend acceptance of their manuscript as a registered report. I appreciate the clarifying changes the authors made in the manuscript and their commitment to providing a balanced and nuanced discussion of the results in the context of these changes at the conclusion of the study.

Version 3:

Reviewer comments:

Reviewer #1

(Remarks to the Author)

Summary

This manuscript has been accepted as a registered report. It describes a competition to quantitatively compare methods that analyze motion changes in single-molecule and single-particle tracking experiments. The competition is a modified repeat of a previous competition that the authors organized in 2021, called the first AnDi Challenge. Three reasons are given for running a second AnDi challenge: First, there is a need to evaluate methods that determine the switch between different diffusive behaviors within single-molecule trajectories. Second, it is necessary to assess the methods' crosstalk in detecting inherent anomalous diffusion due to motion constraints. Third, there is a need to determine whether the bottleneck of the analysis process is at the level of the analysis of the single trajectories or associated with their extraction from the experimental videos. The authors are now providing the analysis and discussion of the Challenge results.

The results presented are interesting and intriguing, but I am afraid I am not grasping the overall conclusion that could be drawn based on these results. My main critique is that the results are shown in terms of error metrics, which (while they are clearly defined in the manuscript) do not provide an intuitive understanding of how good (or bad) the performance of individual tracking methods were. I get the vague sense that none of them were perfect in all tasks. What methods should experimentalists then turn to (or not) to analyze single-molecule/single-cell trajectory data (of a specific type, which may be known a priori). Perhaps the authors could summarize the results into a few, very clear take-home messages. Second, please discuss the results more clearly and in more detail in reference to the three points mentioned above (that provided the rationale for this second AnDi challenge). I am interested in learning what key insights were gained from the results of this challenge.

The critiques below focus mostly on the results and discussion section, as the other part of the manuscript has already been reviewed.

Specific Comments:

1. Line 469: "In this case, only the top method (team I, light blue) was able to produce a reliable classification of the diffusion type of the segments. The difficulty in inferring the correct type of mechanism producing interaction underscores the challenges in accurately analyzing these kind of data, which can have significant implications for the biological interpretation of the results. However, despite the misclassification, all predictions provided a reliable estimation of the residence time within both trapped and unconstrained states (inset of Fig. 6a), thus allowing precise quantification of the relevant biophysical parameters of the experiments."

a. The first conclusion seems to indicate the substantial care should be taken when drawing biological conclusions because tracking analysis results may be incorrect. By contrast, the conclusion just two sentences later states that biophysical

parameters can be precisely quantified. I am therefore not sure what I should take away from these seemingly differing conclusion. Please define more clearly what is meant by “biological interpretation” and “relevant biophysical parameters” to communicate a more cohesive message.

2. Line 482: “Four of the top 5 methods classify the segment rather precisely, which allows them to accurately recover the distribution of each diffusive state, as shown in Fig. 6b–d. The only exception is represented by team E (second bar from the left), whose predictions are biased to always classify segments as diffusive.”

a. I do not know how to interpret “rather precisely”. Such language is too vague.

Determining a segments length/duration is often of great biological significance. Can the methods do this well or not?

b. Perhaps not surprisingly, Fig. 6b shows that it is rather straightforward to distinguish a moving particle from an immobile one and assign trajectory segments accordingly. I believe it should be pointed out that this type of experimental scenario has a LONG history in the field with many case examples that could be cited where the mode of motion contrast is from immobile/quasi stationary to fast moving.

c. I would argue that Fig 6c does not show that the distribution of each diffusive state is accurately recovered by 4 of the top 5 methods. A more nuanced/detailed description is needed here. Since the participants of the top teams are listed as authors on this report, could they provide some insights into why their methods performed particularly well (or not)?

3. Line 487: “The information contained in an individual trajectory is typically sufficient to estimate CPs and diffusive properties. However, for some physical models, the knowledge of the model itself offers additional information that could be used to improve further CP detection and parameter estimation. This is the case for QTM and TCM, where changes in diffusion correspond to spatial constraints. For DIM, diffusion changes are associated with particle proximity; in addition, since particles in a dimer co-diffuse, one could use twice as much information to estimate K and α .”

a. Does the last sentence assume that both co-diffusing particles are detected simultaneously? That is a very unlikely case in STORM and PALM based single-molecule tracking due to the stochastic nature of single-molecule signals. Please rephrase if necessary.

4. Line 542: “We would further like to point out that optimizing methods to provide high scores for the metrics of the competition did not always translate into more meaningful insights about the underlying physical processes. For instance, teams M, H, and O showed significant biases across all experiments when predicting the K distribution but still achieved high rankings according to the metric in Eq. (3) (Supp. Fig. 6). Moreover, accurately predicting the number of true states did not provide a clear advantage with this metric, as most top teams overestimated the number of states but carefully adjusted their relative weights to minimize differences with the ground-truth distribution.”

a. I appreciate the authors bringing up this point. Would it be possible then to define and discuss other metrics that would provide insights into what type of trajectories are particularly challenging for the methods, which methods perform particularly well for a given scenario, which parameters should be trusted and which ones not in a given scenario. In many cases, a priori biological knowledge about a system may narrow down the types of motion that are expected. This manuscript should provide some clear take-home messages for researchers that would like to analyze single-molecule/single-particle trajectories reliably.

b. This paragraph should perhaps be moved to the discussion.

5. Line 584: “Second, ensemble predictions currently rely on post-processing of single-trajectory data. All participants in the Ensemble Task first extracted features from individual trajectories, followed by a post-processing step to estimate the parameters of the diffusive populations. Although this approach has yielded promising results in terms of the proposed metrics, it remains challenging to derive physical insights from such predictions due to the high number of predicted states and the instability in the variances of each mode. We anticipate that methods capable of directly predicting distributions from raw data may avoid the potential biases inherent in single-trajectory analyses. Furthermore, approaches that incorporate the entire set of trajectories contextually may offer improved performance.”

a. Given that high number of predicted states can lead to problems, the authors should caution readers here to not blindly trust the outputs of (still imperfect, I take it) analysis methods. Care should always be taken not to overfit the data with too many states that cannot be assigned to a biophysical even or process.

b. If the analysis suggests/predicts a high number of states, their identities should be validated through additional experiments utilizing purposeful biological, chemical, or physical perturbation. I suggest the manuscript provide context of this nature rather than solely presenting and ranking the results.

(Remarks on code availability)

Reviewer #2

(Remarks to the Author)

I find this work very useful and interesting. It can be suitable for publication in Nature. However, I have questions and comments.

Their list is presented below:

1) As far as I understand (line 1056), the trajectories had a maximum length of 200 frames (timesteps). Were there data gaps (NaN) in trajectories?

2) What number of segments were used? From 1 up to 10? What was the minimum segment length in result? I see line 985: "we set $t_{\min} = 3$.(integer or real?)". How to determine statistics with help of 3 values? I think three time steps are too small for any quantitative statistical analysis. Why did you select 3 timesteps? Or did I misunderstand something?

3) lines 413-416: "However, we also find less correlation between metric scores. This can be attributed to the fact that the predictions of K and α are generally obtained through different approaches, thus enhancing performance for one does not necessarily result in a corresponding improvement for the other."

This could mean something else. Different approaches have different accuracy. Or am I wrong?

4) Fig.2 (right panel): no ticklabels in the time axis? Am I right that each tick is 40 frames?

5) It seems to me that Fig.8 is not quite clear from its caption. What do the panels show? Can you explain in more accessible language?

6) lines 530-531: "However, even the top 5 teams had difficulty in reliably estimating the parameter distribution for some experiments."

I see that the teams could not cope with this task at all. Or is it not all so gloomy? I would like to see a more critical analysis of the results. What approaches failed?

7) Fig.7: how to explain the large spread of improvement (%) results on the left panel? What does it mean the improvement in 200% and more?

8) lines 147-149: "Each method has its own set of advantages and disadvantages, and its performance may depend on the specific problem under consideration. However, there is no universally accepted gold standard for determining which method to use to address each specific problem."

Can you write more specifically here? What do you mean? Here it looks like some kind of philosophical generalization.

9) lines 165-167: "However, the choice of FBM did not limit the generality of the Challenge since other models of diffusion and non-Gaussian behavior can be obtained by properly tuning the parameters of the simulations."

This is not true. You won't get, for example, the subordinated processes or resetting from FBM. I believe that the choice of FBM actually limited the generality of the Challenge. This is not bad. It is impossible to cover the entire diversity of experimental cases at once.

10) line 274: "... $3 = \text{directed } (\alpha \leq 1.9)$." Why 1.9 and not 1.95, for example? May be, it would be better to write $1 < \alpha \leq 1.9$.

(Remarks on code availability)

In my opinion, this code is good. I see that the results of the article are reproducible. The code is useful for a wide community. It contains the necessary comments to install and run the application. I can install and run this code.

Version 4:

Reviewer comments:

Reviewer #1

(Remarks to the Author)

The authors have adequately addressed all previous reviewer comments. I have no further comments and recommend publication of this manuscript.

- Andreas Gahlmann

(Remarks on code availability)

Reviewer #2

(Remarks to the Author)

I have reviewed the article again after revision. It is excellent and fully deserving of publication in Nature. The authors have done an outstanding job and achieved highly interesting results. It is essential for the scientific community to become widely acquainted with their findings. The work incorporates the latest algorithms and approaches for studying and analyzing motion changes in single-particle experiments. I thoroughly enjoyed reading this work and have no suggestions for criticism or improvement. The authors answered my questions well and made the necessary revisions to address the inaccuracies I identified in the article. Overall, the work is very strong.

(Remarks on code availability)

Point-by-point reply to reviewers of *Quantitative evaluation of methods to analyze motion changes in single-particle experiments* by G. Muñoz-Gil *et al.*

Reply to Reviewer 1

Comment R1.1

This manuscript describes a competition to quantitatively compare methods that analyze motion changes in single-molecule tracking experiments. This competition is a modified repeat of a previous competition that the authors organized in 2021, called the first AnDi Challenge. Such challenges are important in that they bring together members of diverse research communities (experimentalists, theoreticians, data analysts, and computer scientists). Three reasons were given to rationalize the running a second AnDi challenge: First, there is a need to evaluate methods that determine the switch between different diffusive behaviors within single-molecule trajectories. Second, it is necessary to assess the methods' crosstalk in detecting inherent anomalous diffusion due to motion constraints. Third, there is a need to determine whether the bottleneck of the analysis process is at the level of the analysis of the single trajectories or associated with their extraction from the experimental videos.

Reply to R1.1 We appreciate the reviewer's valuable feedback, which has greatly contributed to improving the clarity and scope of our article. We would like to stress that the competition we propose goes significantly beyond the aspects explored in the 1st AnDi Challenge. In particular, the 2nd AnDi Challenge aims to serve as a benchmark for methods capable of detecting changes in diffusion characteristics and features both a trajectory-based track and a movie-based track for result comparison.

Comment R1.2

The proposed second AnDi challenge will yield useful information for the field, if the following two criteria can be met: First, anomalous diffusion behaviors need to be clearly attributed to biological effects, such as molecular interactions or molecular confinement to certain subcellular regions. The simulated challenge data should aim to match the time and length scales of these biological effects and accurately represent experimental data (in terms of localization error or factors that contribute to localization errors, in particular).

Reply to R1.2 We agree with the importance of ensuring that the simulations utilized in the Challenge possess biological significance. For this reason, we have taken great care to construct simulations that align with the biophysical phenomena described in the literature. While the simulations are initially presented in generalized units (i.e., pixels and frames), they can be rescaled to meaningful temporal and spatial scales. This point has been now specified on page 7, lines 175–176, where we state:

Simulations are provided in generalized units (i.e., pixels and frames) that can be rescaled to meaningful temporal and spatial scales.

An illustrative example demonstrating this rescaling process can be found in the tutorial available at https://andichallenge.github.io/andi_datasets/tutorials/challenge_two_datasets.html. Similarly, dwell times in diffusive states and sizes of spatial confinement can be scaled in order to be correctly detected within the recording time and field of view. This approach ensures that the challenge remains biologically relevant and aligns with the specified objectives. However, at this stage, we are unable to provide quantitative information. The reason behind this limitation is to maintain the fairness of evaluating the methods submitted by participants. Sharing such information, even with the reviewers, could potentially grant an unfair advantage to those who might be participating in the challenge.

Furthermore, as emphasized in the introduction, while the Challenge primarily centers around data from biological systems, the application of regime-switching detection and trajectory segmentation extends far beyond the realm of living cells. Consequently, we firmly believe that the quantitative assessment of method performance in analyzing data with timescales that replicate dynamics in non-biological systems also holds significant value and relevance.

Comment R1.3

Second, it should be ruled out that observed anomalous diffusion behaviors can instead be attributed to how the data was acquired. This is particularly important when 3D diffusive motion is measured as a 2D (projected) trajectory. Consider, for example, a membrane protein diffusing freely in the membrane of a rod-shaped bacterial cell. If that membrane is pressed flat against the coverslip, then diffusion might appear normal ($\alpha=1$). On the other hand, if the same trajectory occurs on the side wall of the bacterial cell, then diffusion perpendicular to the long axis of the cell will appear highly confined (resulting in $\alpha<1$ in that dimension). A similar example could be made for directed motion along a cytoskeletal filament that is oriented along the z-axis of the microscope. In such a case, the 2D-projected motion would appear highly confined in both the x- and y- dimension ($\alpha<1$), while the actual motion is in fact superdiffusive ($\alpha>1$). The manuscript, in its current form, does not clearly disentangle these two effects. A clear connection between how the challenge data is simulated and biological effects resulting in anomalous diffusion is missing and the challenge data is only in the form of 2D data, which seems to restrict the scope of the proposed work to diffusion within flat membranes. The expected challenge results would be of higher significance and broader in scope, if 3D diffusion were included and the different contributions to experimentally observed anomalous diffusion could be clearly distinguished.

Reply to R1.3 The reviewer is correct that 3D diffusion is a relevant topic for understanding biological problems. During the challenge preparation, we extensively discussed the inclusion of different diffusion models. Naturally, the possibility of incorporating 3D geometries was thoroughly considered since it is relatively straightforward from a technical standpoint. However, our discussions led us to conclude that the inclusion of 3D examples should be avoided and might be even misleading for the following reasons:

- The number of studies on 2D diffusion in nearly flat membranes is significantly larger compared to 3D diffusion, primarily due to the simplicity of the technical approach and data interpretation.
- Live-cell single-molecule imaging provides indicators, such as changes in the diffraction pattern of fluorophores, that suggest the necessity of using 3D techniques to prevent misinterpretation of results, as exemplified by the reviewer.
- Inferring 3D diffusion from 2D projections is not a fair practice from an experimental/analytical perspective unless the isotropy of space and independence of motion along the three dimensions are known.
- Numerous experimental efforts have been made to track biological objects moving in 3D using alternative techniques to live-cell single-molecule imaging. However, these techniques yield different outputs and provide additional information to understand these processes. For cases like the one described by the reviewer, these alternative techniques should be preferred.
- The number of possible 3D geometries that can generate apparent deviations from Fickian diffusion is very large, making it unfeasible to consider all of them for the challenge. Selecting only a subset would be arbitrary and provide limited informative value.

After thorough consideration, we opted to focus the challenge on well-documented cases of diffusion changes in 2D. While the allure of 3D diffusion is undeniable, including it in the challenge would introduce

complexities and limitations that could compromise the clarity and scope of the proposed work. Therefore, our goal is to provide clear and meaningful results that align with the extensive literature on 2D diffusion.

In order to clarify our rationale for not including 3D models of diffusion, we have now inserted a discussion in the introduction that states (page 5, lines 116–122):

In several biological settings, motion takes place on non-flat surfaces or 3D space but only a 2D projection is recorded when using techniques such as live-cell single-molecule imaging. Unless the isotropy of space and independence of motion along the three dimensions are known, the inference of 3D diffusion properties from 2D projections can generate artifacts. Therefore, we opted to focus the challenge on well-documented cases of diffusion changes in 2D. To track biological objects moving in 3D, alternative experimental techniques that provide additional information should be preferred to properly characterize these processes.

Comment R1.4

The following specific comments will hopefully prove useful for improving the manuscript.

Specific Comments:

1. Page 2: “There are three heterogeneity classes that these methods aim to identify: (i) changes in the value of the diffusion coefficient D ; (ii) changes in the anomalous diffusion exponent (often classified as subdiffusion, diffusion, or superdiffusion); and (iii) changes in the phenomenological behavior associated with interactions with the environment (often classified as immobilization, confinement, (free) diffusion, and directed motion).”

o Points 1 and 3 make immediate sense from a biological perspective. Point 2 will be of limited interest to cell biologists, unless clear (biological) rationale for this type of motion can be given. That rationale should also include the relevant time and length scales for a given motion type.

Reply to R1.4 We appreciate the reviewer’s insightful comments. While it is true that changes in the diffusion coefficient and diffusion mode have been reported for quite some time now, the exploration of changes in the anomalous diffusion exponent is a relatively more recent development, particularly in the context of biological systems. Nonetheless, we have identified several relevant examples of such changes in various fields:

- Aging in biological systems (e.g., when the cytoskeleton is changing as a function of time [Lee et al, Sci Adv 7, eabe4334 (2021)] or when proteins form aggregates in neurodegenerative diseases).
- Lyso/endosome motion in eukaryotic cells [Han et al, eLife 9, e52224 (2020)].
- Change of viscoelastic properties in cells during life cycle [Odermatt et al, eLife 10, e64901 (2021)].
- Changes induced by blebbistatin treatment of amoeba cells [Revery et al, Sci Rep 5, 11690 (2015)].
- In viscoelastic solutions under pressure and/or concentration changes [Barlow et al, Proc R Soc Lond 327, 403 (1972); Caspers et al, J Chem Phys 158, 024901 (2023)].
- In actin gels, when the mesh size changes [Levin et al, J Chem Phys 154, 144901 (2021)].
- Variations in the behavior of large birds [Vilk et al, Phys Rev Res 4, 033055 (2022)].
- Financial data, e.g., volatility [Gatheral, Jaisson and Rosenbaum, Quantitative Finance, 18, 933-949 (2018)].

Consequently, we believe that including data involving changes of the anomalous diffusion exponent is essential to characterize methods with respect to this task. Furthermore, we also speculate that the less

intuitive nature of the anomalous diffusion exponent, compared to the diffusion coefficient, may have hindered its exploration in certain systems. In this regard, the introduction of new methods, along with a robust evaluation of their performance, as facilitated by the challenge, could provide a more comprehensive assessment of this behavior, even at the single-molecule level.

To further motivate this choice, we have added a sentence in the introduction (page 4, lines 82–86) that states:

While changes in the diffusion coefficient and in the phenomenological behavior have been widely reported, the exploration of changes in the anomalous diffusion exponent is a more recent development [17]. The introduction of new methods, along with a robust evaluation of their performance, could facilitate a more comprehensive assessment of this behavior.

Although it can be argued that some of the examples we mentioned of changes in the exponent α have not been detected using live-cell single-molecule imaging (which we utilize in Track 1), it is important to note that most of these instances were identified through trajectory analysis (as implemented in Track 2). Thus, we believe that the methods participating in the challenge hold great potential for addressing this task effectively.

Regarding the temporal and spatial scales, as mentioned earlier, our simulations are provided in generalized units (i.e., pixels and frames), enabling their rescaling to any meaningful temporal and spatial scales.

Last, as emphasized in the introduction, we would like to highlight that although the challenge primarily centers around data from biological systems, the application of regime-switching detection and trajectory segmentation extends far beyond the realm of living cells, see, e.g., the diverse systems mentioned above. Consequently, we believe that the quantitative assessment of method performance in detecting changes in the anomalous diffusion exponent, as well as analyzing data with timescales that replicate dynamics in non-biological systems, holds significant value and relevance.

Comment R1.5

2. Page 2: “In the following, we will refer to the anomalous exponent alpha as the characteristic feature of the generating motion.”

o It is clear that mathematically the anomalous exponent can be changed, . . . BUT what are the biological processes that would change alpha other than immobilization, confinement, and directed motion? Immobilization and directed motion are readily detectable in single-molecule trajectories without the employment of sophisticated analysis algorithms. Please provide some specific biological examples (with relevant time and length scales) where transient confinement could influence experimentally measured single-molecule trajectories in living cells. (Experimentally measured single-molecule trajectories can cover the 200 us – 2s timescales; trajectories typically contain less than 100 displacements (the majority contain 10-20), and localization precisions range from 20 to 200 nm depending on how fast emitters diffuse and the frame time with which images were acquired.)

Reply to R1.5 In the introduction, we highlight that the conventional analysis based on calculating the scaling exponent of the mean-squared displacement (MSD) can be influenced by factors such as confinement/immobilization or more intricate effects arising from the surrounding environment (correlations).

However, the anomalous exponent α , intended as the characteristic feature of the generating motion is not only a mathematical parameter. In our response to the previous comment, we provide a range of examples that demonstrate how biological factors can impact this anomalous diffusion exponent. This further supports the importance of considering these influences in our challenge.

Moreover, we respectfully disagree with the reviewer’s statement that “Immobilization and directed motion are readily detectable in single-molecule trajectories without the employment of sophisticated analysis algorithms”. Quantifying the exact durations of immobilization and directed motion is not a trivial task, particularly in the presence of noise, for short trajectories, or when motion changes take

place at the beginning or at the end of the recording. Nevertheless, this quantification is crucial for characterizing biological systems. This is evident from the extensive development of methods specifically designed over more than 30 years to address this challenge, some of which are documented in Table 1.

Comment R1.6

3. Page 2,3: “we will assess the methods’ performance on simulated datasets designed according to biologically inspired models of diffusion and interactions in typical experimental conditions. The participating methods will be further compared on real biological datasets obtained from particles moving in the cell nucleus, cytoplasm, and plasma membrane.”

o The conceptual connection between the simulated datasets and what would/will be observed in live-cell single-molecule tracking experiments is not clear

Reply to R1.6 We apologize for the lack of clarity that generated this comment. The text has been rephrased as (page 5, lines 113–115):

Datasets provided for the last phase of the competition will also include actual experiments for their comparative analysis with the challenge methods (these data will not be used for the ranking).

A more detailed explanation is provided in the section “Datasets and ground truth” (page 8, lines 182–188):

Datasets provided for the last phase of the competition will also include results from actual experiments that have reported the occurrence of heterogeneous diffusion but for which the ground truth is not established beyond any doubt. Therefore, these data will not contribute to the challenge scoring. Nevertheless, the predictions provided by different methods will be comparatively analyzed and discussed also with respect to the conclusions reached in the original publications. Together with the quantitative results obtained from simulations, these analyses will assess the applicability of the methods to real-world experimental data.

Comment R1.7

4. Page 3: “Motion of the FBM-type has been widely observed in biological experiments [7–9] and is often used to describe dynamics in viscoelastic environments [29].”

o Same critique as in Comment 2. I don’t think the reference given here provide a sufficient rationale of why it is worthwhile, from a biological perspective, to distinguish different FBM motion types.

o The cytoplasm and artificial dextran solutions have been shown to exhibit viscoelastic properties, when very large particles are used for tracking. What are biological mechanisms that could lead to anomalous diffusion for small proteins and protein complexes (≤ 200 kDa)?

Reply to R1.7 We believe that the response to the previous comment R1.5 also helps to address the current concern. Indeed, many of the examples mentioned earlier suggest fractional Brownian motion as the underlying motion and viscoelasticity as the cause of anomalous diffusion (e.g., Han et al, eLife 9, e52224 (2020), Revery et al, Sci Rep 5, 11690 (2015), Levin et al, J Chem Phys 154, 144901 (2021), Vilks et al, Phys Rev Res 4, 033055 (2022)). These examples serve to illustrate the relevance and biological significance of anomalous diffusion.

Furthermore, we acknowledge the reviewer’s point that the effect of viscoelasticity may be more pronounced in larger particles compared to smaller molecules. However, we believe that this does not undermine the biological significance or the simulations we produce for the challenge for several reasons. Firstly, particle tracking is a widely documented technique utilized to assess the heterogeneity of the environment, including viscoelastic effects. For example, micron-sized particles serve as probes for studies

of microrheology. Secondly, the tracker particles employed in these experiments possess a size that is comparable to that of viruses and intracellular cargos. This allows for a relevant representation of the dynamics and interactions that occur within biological environments. Last, it is important to note that the size of these particles is still diffraction-limited, thus they were imaged using the same setup employed for imaging individual molecules. This ensures consistency in the experimental methodology and facilitates a meaningful comparison between different systems and scales.

Comment R1.8

5. Page 3: “Multi-state model (MSM) — Particles diffusing according to a time-dependent multi-state (2 or more) model of diffusion, as observed for example in proteins undergoing transient changes of K and/or α , as induced by, e.g., allosteric changes or ligand binding [32, 33].”

o It makes sense that allosteric changes and ligand binding would change K (slightly), but why would it change α ? Please provide a biological example.

6. Page 3: “Dimerization model (DIM) — Particles diffusing according to a 2-state model of diffusion, with transient changes of K and/or α induced by encounters with other particles, observed for example in protein dimerization [34, 35].” o It makes sense that dimerization would change K , but why would it change α ? Please provide a biological example.

Reply to R1.8 We point out that we do not assume that allosteric changes, ligand binding, or dimerization automatically imply a change of α and we have rephrased the text to clarify this point (page 7, lines 156 and 160):

- *MSM — Particles diffusing according to a time-dependent multi-state (2 or more) model of diffusion undergoing transient changes of K and/or α . Examples of changes of K have been observed in proteins as induced by, e.g., allosteric changes or ligand binding [34–36].*
- *DIM — Particles diffusing according to a 2-state model of diffusion, with transient changes of K and/or α induced by encounters with other particles. Examples of changes of K have been observed in protein dimerization [37,38].*

However, for the sake of generality and considering that the role of anomalous diffusion might have been overlooked, we aim at offering a wide palette of diffusion changes on which the methods can be compared and quantified. Therefore, we believe that the possibility of changes in α should be contemplated in the challenge datasets.

Comment R1.9

7. Page 3: “Transient-confinement model (TCM) — Particles diffusing according to a space-dependent 2-state model of diffusion, observed for example in proteins being transiently confined in regions where diffusion properties might change, e.g., the confinement induced by clathrin-coated pits on the cell membrane [36]. In the limit of a high density of trapping regions, this model reproduces the picket-and-fence model used to describe the effect of the actin cytoskeleton on transmembrane proteins [37, 38].”

o The justification of the TCM is stronger overall, compared to the previous two models. Please also provide the relevant length and time scales of confinement in clathrin-coated pits. Are these readily measurable with single-molecule tracking methods?

o Same question for the picket-and-fence model: What are the relevant length and time scales of confinement and are these experimentally accessible using single-molecule tracking?

Reply to R1.9 Confinement caused by both biological mechanisms has been extensively studied using live-cell single-molecule imaging, for example in the references [39] and [40, 41] cited in the text. As shown

in the references, temporal (ms) and spatial scales (hundreds of nanometers) are fully accessible to the technique.

Comment R1.10

8. Page 3: “Quenched-trap model (QTM) — Particles diffusing according to a space-dependent 2-state model of diffusion, representing proteins being transiently immobilized at specific locations as induced by binding to immobile structures, such as cytoskeleton-induced molecular pinning [39].”

o What are the relevant length and time scales of cytoskeleton-induced molecular pinning and are these experimentally accessible using single-molecule tracking?

Reply to R1.10 As for the previous comment, live-cell single-molecule imaging has been largely used to detect trapping as, e.g., shown in Ref.[42]. Temporal (ms) and spatial scales (tens of nanometers) are fully accessible to the technique.

Comment R1.11

9. Page 3: “Datasets provided for the last phase of the competition will also include results from actual experiments corresponding to the motion of particles in living cells. These experiments have reported the occurrence of heterogeneous diffusion as a consequence of specific interactions, conformational changes, or underlying structures, but the ground truth is not established beyond any doubt. Therefore, these data will not contribute to the challenge scoring. Nevertheless, the predictions provided by different methods will be comparatively analyzed and discussed also with respect to the conclusions reached in the original publications.”

o Are these datasets intentionally withheld from the participants?

o If these are already known/published, then they could be used to address my previous comments.

Reply to R1.11 As also clarified above, these datasets are intentionally withheld from the participants to prevent any bias in the evaluation of the performance of the method.

Comment R1.12

10. Page 5: “Each FOV reports data from a variable number of particles diffusing on a 128 x 128 pixel² area.”

o Does that mean that only 2D diffusion is modeled?

o This would exclude many processes mentioned earlier, such as diffusion in the cytoplasm or within the nucleus.

o This raises a bigger question for me: Is anomalous diffusion observed, simply because 3D diffusive motion is measured as a 2D projected trajectory? In other words, is anomalous diffusion observed because of how the data were acquired and not because of biological reasons?

Reply to R1.12 As discussed above, we decided to focus the Challenge on 2D diffusion. This excludes diffusion in the cytoplasm or in the nucleus, which we believe should be better addressed using 3D tracking approach to prevent artifacts in data interpretation, as in the example described by the reviewer.

Comment R1.13

11. Page 5: “For Track 1 (Fig. 3(a)), the coordinates of particles in the same FOV are used to generate 200-frame videos as a series of 8-bit images in the multi-tiff format using Deeptrack 2.1 [5].”

o Are motion-blurring effects included in Deeptrack? The description in Appendix A suggests that it is not.

o Localization errors in single molecule tracking are due to finite fluorescent photon numbers (static localization errors) and motion-blur effects (dynamic localization errors). Therefore, localization errors for fast-moving emitters are worse than those of slowly moving emitters. These factors should be modeled

as realistically as possible.

12. Page 5: “Coordinates of simulated trajectories are corrupted with Gaussian noise corresponding to finite (subpixel) localization precision.”

o Same comment as Comment 11: Are motion-blurring effects included here?

Reply to R1.13 We agree with the reviewer that motion blur can introduce significant noise, particularly if the experimental design is suboptimal and the camera frame rate is slow compared to particle motion. While it is important to minimize such situations during experiment design, complete avoidance is sometimes not feasible. Therefore, we have currently incorporated the capability to generate data with motion blur noise both in the AnDi datasets library (for trajectories) and in DeepTrack 2.1 (for videos).

Similar to 3D diffusion, this point was also extensively discussed during the design of the Challenge. While its technical implementation is straightforward, incorporating motion blur into the competition data is not possible from the challenge perspective. This is because including motion blur in the videos and trajectories necessitates oversampling before subsequent downsampling (at the video recording rate). Consequently, defining the occurrence of changepoints and determining ground truth parameters for each segment becomes extremely challenging (and arbitrary to a certain degree).

Hence, while we have incorporated the capability to introduce noise generated by motion blur in the simulation software, we have made the decision to exclude this particular aspect from the challenge data. Although motion blur is recognized as a significant factor in localization errors, its inclusion would introduce complexities and potential limitations that could detract from the clarity and fairness of the proposed challenge.

To clarify the rationale of this choice, we have now added the following text (page 12, lines 272–277):

Besides localization precision, motion blur can introduce a significant contribution to noise, in particular if the camera frame rate is slow compared to the particle motion [42]. However, this aspect will not be included in the Challenge datasets since it would introduce complexities in the definition of the ground truth that could detract from the objectivity of the Challenge. Nevertheless, the simulation software incorporates the capability to introduce the effect of motion blur both in videos and trajectories.

Reply to Reviewer 2

Comment R2.1

This article focuses on the analysis of trajectories in single-particle experiments. It is based on methods following from properties of fractional Brownian motion, segmentation, hidden Markov model and so on. This research is of interest and could be worthy of publication in Nature. However, at present the manuscript needs some refinements and improvements. My comments are presented below.

Reply to R2.1 We thank the reviewer for endorsing our work. We have addressed all raised points as detailed below.

Comment R2.2

1. I propose to change the title to: "Quantitative evaluation of FBM-type methods to analyze motion changes in single-particle experiments". See my remark 2.
2. The fractional Brownian motion is one of many models useful for the description of motion changes in single-particle experiments. There are other models mentioned in [7-9]. I understand that it is difficult to include all the models together in the same paper and compare them in terms of quality in a particular experiment. Nevertheless, I propose to indicate that this work is the first attempt to implement a competition to characterize and rank the performance of FBM-type methods in analyzing the dynamic behavior of single molecules.

Reply to R2.2 We acknowledge the presence of various theoretical diffusion models that have been described and employed for interpreting individual trajectories. In fact, several of these models were incorporated into the 1st AnDi challenge, where a specific task was dedicated to their classification. However, for this 2nd Challenge, we made a deliberate decision to shift our focus towards the variation of diffusion parameters, specifically the generalized diffusion coefficient and the anomalous exponent α , rather than emphasizing the specific diffusion model itself.

In this context, fractional Brownian motion serves as a valuable tool that enables us to continuously tune these parameters. Additionally, by adjusting other parameters of the simulations, we can obtain other models such as the continuous-time random walk, the Lévy walk, and the annealed transient time motion. For these reasons, we prefer not to refer to fractional Brownian motion in the title.

We have introduced a sentence in the manuscript (page 5, lines 106–115) to clarify the rationale for our choice:

To rely on objective ground truth, we will assess the methods' performance on simulated datasets inspired by models of diffusion and interactions documented in biological systems. Datasets will describe particles undergoing fractional Brownian motion (FBM, [21]) with piecewise-constant parameters. In this context, FBM serves as a tool to enable the tuning of these parameters. Although other kinds of motion and even non-Gaussian behavior have been reported [8], this choice does not limit the generality of the Challenge since other models of diffusion and non-Gaussian behavior can be obtained by properly tuning the parameters of the simulations. Datasets provided for the last phase of the competition will also include actual experiments for their comparative analysis with the challenge methods (these data will not be used for the ranking).

Comment R2.3

3. In addition, it would be useful to mention alternative approaches (for example, 10.1103/PhysRevE.99.012101, 10.1039/C8CP06781C, 10.1039/c8cp06781c, 10.1016/j.chaos.2021.111606) close to the analysis of FBM.

Reply to R2.3 We thank the reviewer for suggesting these articles. While these articles propose interesting approaches to describe heterogeneous diffusion based on autoregressive fractionally integrated moving average models (10.1103/PhysRevE.99.012101, 10.1039/C8CP06781C) and for trajectory classification (10.1016/j.chaos.2021.111606), they do not describe methods for detecting changes of diffusion, which is the main objective of this work.

Comment R2.4

4. I have noticed in this paper that the classification of random trajectories is carried out only for many states, but the classification itself has a wider sense. This work considers Gaussian processes (although a fractional case). However, in many cases the random trajectories of single molecules can demonstrate a non-Gaussian behavior. This should be checked in advance. The reader should not get the impression that the authors offer a universal approach for all occasions. In this work, a particular problem is solved, although in a rather broad formulation.

Reply to R2.4 As discussed above, the Challenge does not focus on trajectory classification. The objective is the detection of changepoints and the characterization of the diffusion parameters in the identified segments. We believe it is reasonable to assume Gaussianity for these segments, whereas non-Gaussianity can be produced by the distribution of diffusivity or dwell times over the whole trajectory.

We have further clarified this point in the introduction (page 5, lines 110–113):

Although other kinds of motion and even non-Gaussian behavior have been reported [8], this choice does not limit the generality of the Challenge since other models of diffusion and non-Gaussian behavior can be obtained by properly tuning the parameters of the simulations.

Comment R2.5

5. In the references there is a mention of articles where all authors are shown without exception, despite a long list (for example, [15]), and there are articles where the authors are mentioned only the first of them [16,17,64]. This should be corrected.6. Table 1 is incomplete. For example,

- Ergodicity breaking in trajectory dynamics (10.1038/s41598-017-05911-y);
- Detection of trajectory-to-trajectory fluctuations (10.1088/1367-2630/abf204)

Reply to R2.5 We thank the reviewer for pointing out the issue with the number of authors, it has been fixed in the new version of the manuscript. With respect to Table 1, we acknowledge that there is a considerable volume of articles reporting methods for the detection of heterogeneous diffusion. Our utmost effort has been directed toward ensuring the inclusion of at least one representative article for each methodology. We have taken the reviewer’s suggestion into account and have now incorporated the recommended works into the table.

Comment R2.6

7. The classification of heterogeneity (p.2, left column, the first paragraph below) is very simplified like Table 1. I propose to reformulate the following your phrase: "There are three heterogeneity classes that these methods aim to identify:" to " We consider only three heterogeneity classes that these methods aim to identify:" It seems to me that it will be more accurate.

Reply to R2.6 The text has been rephrased as suggested by the reviewer.

Comment R2.7

8. While checking the Python software of the article, I ran into a data simulation problem. So slow and so long. I couldn’t wait for the command:

```
dataset = AD.create_dataset(T = 4, N_models = 2, exponents = [0.7, 0.9], models = [0, 2], save_trajectories = True, path = 'datasets/')
```

to complete. My computer has pretty decent hardware: Xeon E5-2689, RAM 16 Gb and SSD 256 Gb.

What are requirements for computers in your simulations? Can they be performed on a personal computer? Thus, the installation of your codes was OK, but my software verification in action is problematic.

Reply to R2.7 The reviewer is trying to execute a line of code that we use in our tutorial:

https://andichallenge.github.io/andi_datasets/tutorials/challenge_one_datasets.html

to describe how to generate the dataset of the 1st AnDi challenge. We have tested the code on several machines, and it runs smoothly on a regular laptop in a few seconds. The same code was also extensively used by the participants of the 1st AnDi Challenge without problems. It is thus difficult to recreate the issue experienced by the reviewer and understand its cause without further information. If the problem persists, we would appreciate the reviewer opening an issue in our GitHub repository:

https://github.com/AnDiChallenge/andi_datasets/issues.

Comment R2.8

9. I didn't fully understand what data are used in this paper. Only numerical simulations of random trajectories predefined properties? If yes, how can be this used for single-particle experiments or their analysis? It is not evident that your methods "guide researchers to identify optimal tools for analyzing their experiments". This approach is strongly limited in the framework of FBM. How do experimentalists know that their data are related to FBM?

Reply to R2.8 Similarly as we did in the 1st AnDi Challenge, we will use simulations to carry out the quantification of methods' performance. This is necessary since metrics evaluation requires knowledge of the ground truth, which is not available for experimental data. However, in addition to simulations of trajectories (Track 2), we will also use simulated videos (Track 1). Experimental data will be blindly integrated into the dataset of the final phase of the challenge. However, these data will not be used for scoring but only for comparative purposes. A more detailed explanation is provided in the section "Datasets and ground truth" (page 6):

Datasets provided for the last phase of the competition will also include results from actual experiments. These experiments have reported the occurrence of heterogeneous diffusion as a consequence of specific interactions, conformational changes, or underlying structures, but the ground truth is not established beyond any doubt. Therefore, these data will not contribute to the challenge scoring. Nevertheless, the predictions provided by different methods will be comparatively analyzed and discussed also with respect to the conclusions reached in the original publications. Together with the quantitative results obtained from simulations, these analyses will assess the applicability of the methods to real-world experimental data.

We would also like further highlight that our objective is to provide an objective assessment of existing and novel methods for the analysis of changes of diffusion. The performance of these methods over the datasets corresponding to different models and parameters will provide a guide for users and a benchmark for developers.

Along the line of our reply to a previous comment (R2.2), for the dataset simulations, fractional Brownian motion is a tool to continuously tune parameters such as the generalized diffusion coefficient and the anomalous diffusion exponent over time. By adjusting other parameters of the simulations, we can obtain other models such as the continuous-time random walk, the Lévy walk, and the annealed transient time motion. However, the objective of the challenge is to test methods over the detection of changepoints and the characterization of the diffusion parameters in the identified segments. Therefore, we believe that

the assumption of fractional Brownian motion over these segments does not limit the generality of the approach.

Comment R2.9

10. In the end of Introduction (as well as in Abstract) the authors write something like "typical experimental conditions". What are conditions? Could you, please, represent them concretely?

Reply to R2.9 We provide simulations in generalized units (i.e., pixels and frames) that can be rescaled to meaningful temporal and spatial scales corresponding to experimental conditions. This point has been now specified on page 7, lines 175–176, where we state:

Simulations are provided in generalized units (i.e., pixels and frames) that can be rescaled to meaningful temporal and spatial scales.

An illustrative example demonstrating this rescaling process can be found in our tutorial: https://andichallenge.github.io/andi_datasets/tutorials/challenge_two_datasets.html. Similarly, dwell times in diffusive states and sizes of spatial confinement can be scaled in order to be properly detected within the recording time and field of view. However, at this stage, we are unable to provide quantitative information. The reason behind this limitation is to maintain the fairness of evaluating the methods submitted by participants. Sharing such information, even with the reviewers, could potentially grant an unfair advantage to those who might be participating in the challenge.

Point-by-point reply to reviewers of *Quantitative evaluation of methods to analyze motion changes in single-particle experiments* by G. Muñoz-Gil *et al.*

Reply to Reviewer 1

Comment R1.1

Summary

While the authors have attempted to address my comments in their rebuttal, I do not believe the edits in the manuscript are sufficient given my concerns. My main concern is as follows: As the authors state themselves, “the Challenge primarily centers around data from biological systems”, and they make references to ligand binding and dimerization. For such molecular systems, anomalous diffusion is, to my knowledge, not established. This is in contrast to the field of microrheology, where researchers track tracer particles that are orders of magnitude larger than single proteins and protein complexes. In microrheology, anomalous diffusion is well documented and has been attributed to, e.g. the viscoelasticity of the cytoplasm, which becomes more pronounced at larger length scales. Unfortunately, there is MUCH confusion in the field about the differences between single (tracer) particle tracking (microrheology) and bona fide single-molecule tracking of small proteins and protein complexes.

Reply to R1.1 We thank the Reviewer for these comments that make it clear that we need to further clarify the scope of the competition we propose. For clarity, we have split the reply into different sections.

Objective of the Challenge and the role of anomalous diffusion

As we state in the title and the introduction, the main objective of the competition is to evaluate methods for the analysis of *changes occurring in the behavior of time traces associated with particles moving in biological environments*. These changes have been well documented across the biophysical literature for single molecules and organelles in living cells and have been associated with various kinds of physical and biochemical interactions. As we state in the manuscript (lines 50–57):

The abundance of experimental single-particle trajectories, encompassing molecules, protein complexes, vesicles, and organelles, has led to the development of numerous methods dedicated to the reliable detection of changes in their motion patterns (as summarized in Table 1). These changes serve as valuable indicators for the occurrence of interactions within the system. For instance, diffusing particles may exhibit variations in diffusion coefficients (due to processes like dimerization, ligand binding, or conformational changes) or shifts in their mode of motion (attributed to transient immobilization or confinement at specific scaffolding sites) (Fig. 1(a))[6].

Please notice that, at this point, we do not explicitly mention “anomalous diffusion”, even though interactions can be associated with the emergence of anomalous diffusion [1, 2, 3, 4, 5, 6, 7, 8]. We do this to keep our main focus on the heterogeneity of the behavior. To further stress this point and avoid confusion for the readers, we have included the following sentence (lines 81–85):

While we have retained the name of the 1st AnDi Challenge to build upon its already-established community, we would like to emphasize that the main focus of this 2nd AnDi Challenge is on revealing heterogeneity rather than anomalous diffusion. In the simulated datasets, anomalous diffusion will either emerge from heterogeneity itself or be intentionally introduced for evaluation purposes.

Anomalous diffusion in molecular systems is well established

In the cases mentioned so far there is no occurrence of correlation, as for the diffusion in viscoelastic media mentioned by the Reviewer. However, either the change of diffusion [9, 10, 11] or the presence of spatial constraints [12, 13, 14, 15, 16, 17] may produce (at least transiently) a deviation from linearity in the MSD, leading to the observation of anomalous diffusion. Technically speaking, we believe that the claim of anomalous diffusion is fully legitimate since the motion is either non-Fickian or non-Gaussian on the observation timescales.

The causes of anomalous diffusion in some of the examples described above are fundamentally different from those responsible for the behavior of tracer particles in a viscoelastic environment (microrheology), which are often described within the framework of FBM. Specifically, while FBM can be considered to produce an inherent anomalous diffusion, the examples from the biological experiments are related to mixing, heterogeneity, or disorder. To further clarify this point, we have modified the manuscript (lines 57–62):

These interactions can also result in deviations from standard Brownian motion, as characterized by Einstein’s free diffusion model, which includes a linear mean-squared displacement (MSD) and a Gaussian distribution of displacements [7]. This is the case, e.g., of spatiotemporal heterogeneities producing transient subdiffusion at specific timescales [8–19]. Other mechanisms can instead produce asymptotically-anomalous diffusion [2,20–22].

However, the biological literature also provides examples where the motion of macromolecular complexes is found compatible with FBM [18, 19, 20, 21, 22, 23, 24]. In addition, we would like to point out that other models can also produce inherent anomalous diffusion. Famous examples are the CTRW observed in random media [25, 26] and the Lévy walk observed in foraging animals [27]. Examples of motion compatible with models of anomalous diffusion have been reported in the literature, as now specified in the manuscript (lines 62–65):

Anomalous diffusion compatible with models such as fractional Brownian motion [23–28], continuous-time random walk [29,30], scaled Brownian motion [31], and Lévy walk [32] has been observed for telomers, macromolecular complexes, proteins, and organelles in living cells.

Therefore, we can safely argue that the occurrence of anomalous diffusion is widely established in molecular systems. The works cited above provide biological significance to the models of interaction that we propose for the Challenge.

Is diffusion always anomalous in living cells?

Focusing specifically on the cases of changes of diffusion due to dimerization and ligand binding, there are several publications reporting this evidence [28, 29, 30, 31, 32, 33, 34, 35, 36] that do not explicitly mention anomalous diffusion. Still, we believe that these examples provide further justification for the focus of the Challenge on the evaluation of methods for the analysis of changes of dynamic behavior: As previously mentioned, this objective is clearly stated in the title (which does not include the term “anomalous diffusion”):

Quantitative evaluation of methods to analyze motion changes in single-particle experiments

and in the manuscript (lines 73–74):

... there is a need to evaluate methods to determine the switch between different diffusive behaviors, as often observed in experiments.

and (lines 79–81):

These needs shaped the design of the 2nd AnDi Challenge, defining its scope with a focus on characterizing and ranking the performance of methods that analyze changes of dynamic behavior.

One could thus ask why we include examples of changes of dynamics that do not always correspond to previously reported experimental observations and “force” the occurrence of anomalous diffusion. We consider these cases for the sake of generality of the methods involved in the competition. In fact, we believe that it is important to include situations, such as $\alpha \neq 1$ or changes of α , that although might not have been observed so far in experiments can provide a thorough evaluation of the performance of the participating methods. To clarify this point, we have added the following text (lines 127–129):

The combination of parameter values and interaction models might produce situations that do not correspond to previously documented biological scenarios but will be valuable to test the methods’ performance in a wide range of conditions.

Similarly (lines 197–201):

While the interaction mechanisms producing the heterogeneous diffusion are inspired by biological scenarios, some of the combinations of diffusion parameters and models lead to situations that may not correspond to previously documented biological contexts. Nevertheless, this approach holds substantial value as it enables the comprehensive assessment of method performance across a broad spectrum of conditions.

However, we would also like to point out that changes in anomalous diffusion exponents have been recently observed [37, 38, 39, 40]. We have thus included the following text (lines 93–99):

While changes in the diffusion coefficient and in the phenomenological behavior have been widely reported, the exploration of changes in the anomalous diffusion exponent is a more recent development [43–46], which is attracting increasing interest also from the theoretical point of view [47–50]. The introduction of new methods for data analysis, as promoted by the Challenge, could push the performance for detecting subtle changes in these diffusion properties in systems where they have so far been overlooked.

Why FBM?

In order to be able to simulate normal and anomalous diffusion with (and without) changes of diffusion parameters, we rely on FBM. FBM allows reproducing situations experimentally corroborated (such as a single diffusion state with $\alpha = 1$ and changes of K with $\alpha = 1$), but provides also the flexibility needed to include other scenarios (such as varying α) that have been only recently observed to take place in cells [37, 38, 39, 40], or even scenarios that might not be realizable in biological systems (please refer to the previous subsection). We also note that FBM is currently under close scrutiny for theoretically describing heterogeneous and evolving environments [41, 42, 43, 44].

However, we do not mean to assert that individual molecules undergo FBM in all cases, nor that they experience the viscoelasticity of the environment. To make it explicit, we have added the following text (lines 122–125):

FBM-type motion has been widely observed in biological systems by means of microrheology, a technique that uses large tracer particles as probes to study the properties of the environment [55]. Anomalous diffusion compatible with FBM has also been reported for telomers and macromolecular complexes in living cells [20,23–28,56].

The questions that we want to answer through the Challenge are: If some particles were to move with FBM, would the methods be able to detect it? If particles performing anomalous FBM were to be suddenly confined or change diffusivity, would the methods be able to detect it?

The simulated datasets that we plan to provide for the Challenge do include situations that have been experimentally observed, but this information cannot be disclosed to participants to ensure the fairness of the Challenge and the usefulness of its outcome. At the same time, for the sake of generality, as discussed above, we will also include conditions that have not been experimentally observed (yet). In addition, as also pointed out in the previous reply, the less intuitive nature of the anomalous diffusion exponent, compared to the diffusion coefficient, may have hindered its exploration in certain systems. In this regard, the introduction of new methods, as facilitated by the Challenge, will provide the tools necessary for a more comprehensive exploration of exotic behaviors, such as the occurrence of changes of α .

In order to eliminate any possibility of confusion, when introducing the FBM, we have added the following text (lines 125-133):

Beyond this evidence, in the context of the Challenge, FBM serves as a tool to enable the tuning of diffusion parameters. The combination of parameter values and interaction models might produce situations that do not correspond to previously documented biological scenarios but will be valuable to test the methods' performance in a wide range of conditions. In biological experiments, other kinds of motion and even non-Gaussian behavior have been reported [21]. However, the choice of FBM does not limit the generality of the Challenge since other models of diffusion and non-Gaussian behavior can be obtained by properly tuning the parameters of the simulations.

We would like to further stress that, in the Introduction, we warn the reader about the different causes of anomalous diffusion (lines 75-76):

... it is necessary to assess the methods' crosstalk in detecting inherent anomalous diffusion from nonlinearity in the MSD due to motion constraints or heterogeneity.

This sentence expresses one of three needs that we identified from the discussion following the 1st AnDi Challenge. We believe that this goes exactly in the direction of clarifying the confusion mentioned by Reviewer 1. In fact, one of the issues we aim to address can be simplified through the following questions: “Are analysis methods taking part in the Challenge able to discern/discriminate between a trajectory moving according to a single diffusion state but that is inherently anomalous (e.g., due to viscoelasticity, as a tracer in the cytoplasm) and a trajectory diffusing “normally” but undergoing changes of D or switching between normal and confined diffusion (e.g., as some proteins on the cell membrane)?”.

Comment R1.2

It would undoubtedly be beneficial for readers if the authors clearly distinguished between single tracer particle tracking (microrheology) and actual single-molecule tracking and make it clear that the methods should not be applied, without clear biological justification, to the diffusion of molecules in cells.

Reply to R1.2 As discussed at the end of the previous comment, we have now specified the difference between microrheology and single-molecule imaging when introducing FBM (lines 122–125):

FBM-type motion has been widely observed in biological systems by means of microrheology, a technique that uses large tracer particles as probes to study the properties of the environment [55]. Anomalous diffusion compatible with FBM has also been reported for telomers and macromolecular complexes in living cells [20,23–28,56].

The data that we present for the Challenge include simulated videos and trajectories and, from an imaging technical perspective, might very well correspond to a diffraction-limited tracer or a single molecule. The methods that will take part in the Challenge are agnostic to the experimental technique and to the size of the particle. They should be able to analyze experiments with dynamics similar to those included in our simulations and provide the right answer, no matter whether the motion is strictly anomalous or just displaying changes that lead to a nonlinear MSD. Although it is not our main objective, one of

these methods applied to a microrheology video/trace should identify a single state model and report the corresponding parameters K , and α . Similarly, if applied to an individual molecule undergoing Brownian motion and experiencing switches in the diffusion coefficient, it should identify the corresponding changes and provide $\alpha = 1$.

Comment R1.3

I would also like to see a complete discussion of why anomalous effects may be appearing in tracking data. As highlighted in my initial peer-review comments and acknowledged by the authors in the rebuttal letter, anomalous diffusion may be an artifact due to how the data were acquired (e.g. 3D diffusive motion measured as a 2D (projected) trajectory).

Reply to R1.3 In our revised manuscript, we have expanded our explanation that now reads (lines 136–151):

The standard and straightforward approach in live-cell single-molecule imaging primarily captures information related to lateral motion. In cases involving flat membranes or isotropic systems, employing 2D imaging and tracking techniques suffices for obtaining accurate motion-related parameters. However, when dealing with motion on non-flat surfaces or within anisotropic 3D environments, relying solely on 2D projections can result in critical information being overlooked, potentially leading to the misinterpretation of diffusion coefficients or the appearance of apparent anomalous diffusion effects [57,58]. Consequently, drawing definitive conclusions under such circumstances should be avoided or approached with caution. To study motion occurring in 3D space, it is advisable to employ 3D tracking methods, such as off-focus imaging (i.e., the analysis of ring patterns in the defocused point spread function) [59], interference/holographic approaches [60], multifocus imaging [61], or point spread function engineering [62]. Although more challenging, these methods can measure also the motion along the axial dimension, facilitating a more accurate characterization. For the purposes of the Challenge, we have chosen to concentrate on studying changes in diffusion behavior occurring within a 2D context, driven by particle interactions of various types.

Comment R1.4

These critiques are also in line with Reviewer 2, who remarked that “It is not evident that your methods “guide” researchers to identify optimal tools for analyzing their experiments. . . . How do experimentalists know that their data are related to FBM?”

Reply to R1.4 We would like to point out that we never stated that “*the methods* will guide researchers to identify optimal tools” (which is a misquote from Reviewer 2). What we stated in the abstract is:

The competition will ... guide researchers to identify optimal tools for analyzing their experiments.

We strongly believe in this statement, since the results of the Challenge will provide quantitative information on how different methods perform on the analysis of changes of diffusion in a wide range of situations. In fact, as we did in the 1st AnDi Challenge, we will use simulations to carry out the quantification of the methods’ performance. This is necessary since quantitative metrics require knowledge of the ground truth, which is not available for experimental data. However, in addition to simulations of trajectories (Track 2), we will also use simulated videos (Track 1). Experimentalists will thus know which performance to expect with respect to the evaluation of specific parameters in a wide range of conditions. Obviously, as for any other case, the application of a method to a specific experiment requires further consideration. In the discussion, when presenting the results of the assessment, we will also suggest the readers make their

choice upon testing a few different methods using simulations based on parameters that can be estimated from their own data.

Regarding the question on FBM, as we already replied to Reviewer 2 in the first revision round and along the lines of our reply to a previous comment (R1.1), in our simulations FBM is a tool to continuously tune parameters such as the generalized diffusion coefficient and the anomalous diffusion exponent over time, see e.g. Ref. [44]. By adjusting other parameters of the simulations, we can obtain other models such as the continuous-time random walk, the Lévy walk, and the annealed transient time motion. However, the objective of the challenge is to test methods over the detection of change points and the characterization of the diffusion parameters in the identified segments. Therefore, we believe that the assumption of FBM over these segments does not limit the generality of the approach. However, if experimentalists need to perform a classification of the underlying model, they can rely on the methods that were already developed and ranked during the 1st AnDi Challenge.

Comment R1.5

If the necessary modifications are not introduced into the manuscript, my worry is that a myriad of inexperienced researchers will, upon reading it, simply apply the winning algorithm on their 2D single-molecule trajectories without sufficient considerations of why this algorithm may or may not be appropriate. I can appreciate that the authors want the competitors to address a general problem (formulated in generalized and later scalable length and time scales). I can also appreciate that that anomalous diffusion may play an important role in other fields (finance, bird movements, etc.) – although I am not an expert in those fields. If the authors want this manuscript to focus primarily on molecular and cellular biological systems (which it currently is), then they have the responsibility to clearly highlight the limitations of the challenge data and clearly delineate for which systems (i.e. microrheological motion, NOT single-molecule motion) the algorithms are appropriate.

Reply to R1.5 We understand and share the Reviewer’s concern regarding the potential misuse of the algorithms by inexperienced researchers. We agree that it is essential to emphasize the appropriate use and limitations of each algorithm. This is in fact a clear objective of the Challenge. We have made a series of modifications to the manuscript to avoid these risks and clarify the scope of the Challenge. For example, as specified in the reply to an earlier comment, we warn the readers about the different mechanisms that might produce anomalous diffusion (lines 75–76):

... it is necessary to assess the methods’ crosstalk in detecting inherent anomalous diffusion from nonlinearity in the MSD due to motion constraints or heterogeneity.

and about the risks of extracting information from 2D projections of 3D motion (lines 139–142):

... when dealing with motion on non-flat surfaces or within anisotropic 3D environments, relying solely on 2D projections can result in critical information being overlooked, potentially leading to the misinterpretation of diffusion coefficients or the appearance of apparent anomalous diffusion effects [57,58].

Along these lines, the Challenge will not only reveal a winner but quantitatively assess and discuss the performance of each method with respect to the analysis of video and trajectories to reveal the presence of changes in diffusion properties.

The results of the Challenge will be presented exactly as the Reviewer suggests. By comparing the methods’ performance in different situations, we will assess whether the methods are able to distinguish anomalous diffusion from changes of diffusion, therefore avoiding potential pitfalls in the interpretation of experimental data. If a method falsely identifies anomalous diffusion in a trajectory simulated with $\alpha = 1$ but with switches of the diffusion coefficient, it would be clear that care must be taken for its application. For this purpose, we plan to include a dedicated section in the manuscript that discusses the practical applications of the algorithms and their suitability for different types of data. However, these indications

cannot be provided a priori, since we do not know yet the results of the competition and therefore the performance and limitations of the methods.

As we detailed in the reply to comment R1.1, several publications have demonstrated the occurrence of anomalous diffusion for lipids, proteins, telomers, macromolecular complexes, and organelles in living cells and are now cited in the manuscript (lines 54–65):

For instance, diffusing particles may exhibit variations in diffusion coefficients (due to processes like dimerization, ligand binding, or conformational changes) or shifts in their mode of motion (attributed to transient immobilization or confinement at specific scaffolding sites) (Fig. 1(a)) [6]. These interactions can also result in deviations from standard Brownian motion, as characterized by Einstein’s free diffusion model, which includes a linear mean-squared displacement (MSD) and a Gaussian distribution of displacements [7]. This is the case, e.g., of spatiotemporal heterogeneities producing transient subdiffusion at specific timescales [8–19]. Other mechanisms can instead produce asymptotically-anomalous diffusion [2,20–22]. Anomalous diffusion compatible with models such as fractional Brownian motion [23–28], continuous-time random walk [29,30], scaled Brownian motion [31], and Lévy walk [32] has been observed for telomers, macromolecular complexes, proteins, and organelles in living cells.

We firmly believe that including the occurrence of inherent anomalous diffusion in addition to changes of diffusion coefficient and mode is of fundamental importance for the generality of the challenge and we do not see any reason why we should limit it a priori. The data that the methods will be applied to are technically the same (i.e., they have the same format), no matter if a tracer or a molecule is recorded (even though their meaning can be very different). In order to perform well, algorithms should provide reasonable results for a wide range of dynamic situations. Their quantitative evaluation will serve to identify which method performs best depending on the conditions.

Comment R1.6

Below are my responses to the authors’ rebuttal comments and changes made in the manuscript. Specific Comments:

1. Reply to R1.3 “... our discussions led us to conclude that the inclusion of 3D examples should be avoided and might be even misleading for the following reasons: a. The number of studies on 2D diffusion in nearly flat membranes is significantly larger compared to 3D diffusion, primarily due to the simplicity of the technical approach and data interpretation. Response: It is true 2D diffusion measurements are more numerous, perhaps due it being a technically simpler experimental approach. However, data interpretation is NOT simpler. In fact, it can be substantially complicated due to the potential of artifacts that lead to the appearance of anomalous diffusion. This needs to be made clear in the manuscript.

Reply to R1.6 As detailed in our previous reply, we are aware of the difficulties of data interpretation in 2D experiments if they correspond to 3D motion. The Reviewer seems to overlook that, as they quote, in the reply we explicitly refer to *studies on 2D diffusion in nearly flat membranes*. In these cases, the interpretation is undoubtedly easier with respect to the case of 3D motion projected in 2D.

We have further stressed this point in the new version of the manuscript (lines 139–142):

... when dealing with motion on non-flat surfaces or within anisotropic 3D environments, relying solely on 2D projections can result in critical information being overlooked, potentially leading to the misinterpretation of diffusion coefficients or the appearance of apparent anomalous diffusion effects [57,58].

Comment R1.7

b. Live-cell single-molecule imaging provides indicators, such as changes in the diffraction pattern of

fluorophores, that suggest the necessity of using 3D techniques to prevent misinterpretation of results, as exemplified by the reviewer.

Response: It needs to be made clear in the manuscript when researcher should not measure (projected) 2D trajectories. I am not quite sure what the authors mean by “changes in the diffraction pattern of fluorophores”. Please elaborate.

c. Numerous experimental efforts have been made to track biological objects moving in 3D using alternative techniques to live-cell single-molecule imaging. However, these techniques yield different outputs and provide additional information to understand these processes. For cases like the one described by the reviewer, these alternative techniques should be preferred.

Response: I am unclear which methods the authors refer to here. Please clarify. These methods are not mentioned in the manuscript either. If they are to be preferred to tracking approaches, they should be mentioned.

d. The number of possible 3D geometries that can generate apparent deviations from Fickian diffusion is very large, making it unfeasible to consider all of them for the challenge. Selecting only a subset would be arbitrary and provide limited informative value.

Response: True, but researchers still need to be made aware of this issue. Otherwise, any sophisticated data analysis could be meaningless, because of unconsidered artifacts in the data.

Reply to R1.7 Addressing the points raised by the Reviewer, we have now extended the text and included the proper references (lines 136–151, see also the response to comment R1.3.):

The standard and straightforward approach in live-cell single-molecule imaging primarily captures information related to lateral motion. In cases involving flat membranes or isotropic systems, employing 2D imaging and tracking techniques suffices for obtaining accurate motion-related parameters. However, when dealing with motion on non-flat surfaces or within anisotropic 3D environments, relying solely on 2D projections can result in critical information being overlooked, potentially leading to the misinterpretation of diffusion coefficients or the appearance of apparent anomalous diffusion effects [57,58]. Consequently, drawing definitive conclusions under such circumstances should be avoided or approached with caution. To study motion occurring in 3D space, it is advisable to employ 3D tracking methods, such as off-focus imaging (i.e., the analysis of ring patterns in the defocused point spread function) [59], interference/holographic approaches [60], multifocus imaging [61], or point spread function engineering [62]. Although more challenging, these methods can measure also the motion along the axial dimension, facilitating a more accurate characterization. For the purposes of the Challenge, we have chosen to concentrate on studying changes in diffusion behavior occurring within a 2D context, driven by particle interactions of various types.

Comment R1.8

To address Comment R1.3, the authors added the following text to the introduction: “In several biological settings, motion takes place on non-flat surfaces or 3D space but only a 2D projection is recorded when using techniques such as live-cell single-molecule imaging. Unless the isotropy of space and independence of motion along the three dimensions are known, the inference of 3D diffusion properties from 2D projections can generate artifacts. Therefore, we opted to focus the challenge on well-documented cases of diffusion changes in 2D. To track biological objects moving in 3D, alternative experimental techniques that provide additional information should be preferred to properly characterize these processes.”

Response: This added text is VERY brief and should be expanded on to address my concerns listed above. Please clarify what is meant by “well-documented cases of diffusion changes in 2D” and “alternative experimental techniques”.

Reply to R1.8 The text has been expanded to address the comments above, as reported in the reply to the previous comment.

Comment R1.9

2. In Response to Comment R1.4, the authors added the following text to the introduction: “While changes in the diffusion coefficient and in the phenomenological behavior have been widely reported, the exploration of changes in the anomalous diffusion exponent is a more recent development [17]. The introduction of new methods, along with a robust evaluation of their performance, could facilitate a more comprehensive assessment of this behavior.”

Response: ... which would enable what exactly? Please add one more sentence to complete the thought for the reader – perhaps by giving a tangible, concrete example based on the publications listed by the authors. The authors state in their rebuttal that “...we believe that the quantitative assessment of method performance in detecting changes in the anomalous diffusion exponent, as well as analyzing data with timescales that replicate dynamics in non-biological systems, holds significant value and relevance.” The revised text should convince the reader of this rather than just listing potential applications without much context.

Reply to R1.9 As discussed above (see comment R1.1), the Challenge dataset allows us to include examples of situations that are well documented in the literature (anomalous or Brownian diffusion with no changes, changes of diffusion coefficient, changes between diffusion and confinement, etc...) but is flexible enough to also include cases that might not have been experimentally observed so far or reported only recently. We believe that the Challenge, by pushing the development of methods able to perform in these circumstances, might thus trigger new discoveries or corroborate the existence of cases that have been only recently documented (i.e., the occurrence of changes of α [37, 38, 40]).

Following the Reviewer’s suggestion, we have now rephrased the text (lines 93–99):

While changes in the diffusion coefficient and in the phenomenological behavior have been widely reported, the exploration of changes in the anomalous diffusion exponent is a more recent development [43–46], which is attracting increasing interest also from the theoretical point of view [47–50]. The introduction of new methods for data analysis, as promoted by the Challenge, could push the performance for detecting subtle changes in these diffusion properties in systems where they have so far been overlooked.

Comment R1.10

3. In Response to Comment R1.7, the authors gave a list of references meant to establish Motion of the FBM-type in real biological systems. These references are however not focused on motion of small proteins and protein complexes (which is what my comment referred to), but they instead focus on the motion of endocytic organelles (Han et al, eLife 9, e52224 (2020)), multicellular spheroids (Revery et al, Sci Rep 5, 11690 (2015)), dynamics of large tracer particles in an actin network (Levin et al, J Chem Phys 154, 144901 (2021)), and rhodamine 6G molecules in water nanofilms of a few (one to eight) molecular layers on glass coverslips as well as large tracer particles in mammalian cell, and macroscopic objects (Vilk et al, Phys Rev Res 4, 033055 (2022)).

Response: None of the above examples cover the motion of small proteins and protein complexes. If no clear experimental evidence for anomalous diffusion of molecules in living cells can be referred to by the authors, then the manuscript should be revised to remove any overly broad and thus misleading statements and references claiming that anomalous diffusion behavior is “well documented” in “real biological systems” or in “biological environments” in general. Specifically, the language needs to be more precise overall and a distinction needs to be made, for the benefit of inexperienced readers, between single tracer particle tracking (microrheology) and actual single-molecule tracking of proteins and protein complexes.

The dynamics of these diffusers are NOT the same based on the experimental evidence available to date. In accordance with that fact, I appreciate that the authors changed the text in response to my comment R1.8. I do believe however that such edits need to be made where applicable, throughout in the manuscript.

Reply to R1.10 Addressing the Reviewer’s comments, we have now provided a thorough explanation of the relevance of anomalous diffusion of proteins and macromolecular complexes in living cells (see comment R1.1). We have further added a paragraph including the definition of microrheology and references to articles where anomalous diffusion was found compatible with FBM (lines 122–129):

FBM-type motion has been widely observed in biological systems by means of microrheology, a technique that uses large tracer particles as probes to study the properties of the environment [55]. Anomalous diffusion compatible with FBM has also been reported for telomers and macromolecular complexes in living cells [20,23–28,56]. Beyond this evidence, in the context of the Challenge, FBM serves as a tool to enable the tuning of diffusion parameters. The combination of parameter values and interaction models might produce situations that do not correspond to previously documented biological scenarios but will be valuable to test the methods’ performance in a wide range of conditions.

Regarding the language, we would like to point out that — following the works of K. Jaquaman and other experts in the field [45, 46] — we use the term “single-particle tracking” to refer to the actual “tracking” procedure, i.e., the data processing that leads to the extraction of trajectories, whereas we use the term “live-cell single-molecule imaging” for the experimental part. This is also exemplified in Fig. 1. Although it might be more specific to say “single-molecule tracking”, we believe that in this case, the difference is purely semantic since algorithms and methods that lead from the imaging data to the trajectory are essentially the same, as the algorithms for further trajectory analysis.

To the best of our knowledge, we have made the required edits where applicable ensuring clarity for the readers.

Comment R1.11

4. In Response to Comment R1.12, the authors state that “As discussed above, we decided to focus the Challenge on 2D diffusion. This excludes diffusion in the cytoplasm or in the nucleus, which we believe should be better addressed using 3D tracking approach to prevent artifacts in data interpretation, as in the example described by the reviewer.”

Response: I do believe that remains a weakness of the manuscript/proposed work. If the authors decide to stay with 2D diffusion only (due to it being the more abundant type of measurement), then a more complete discussion is needed to address whether anomalous diffusion is observed because of how the data were acquired and not because of biological reasons. Making a single fleeting mention to the potential of artifacts is not sufficient, in my opinion, as it would be easily overlooked by many readers. Specific use cases that are not appropriate for the tested methods should be specifically mentioned with sufficient context and rationale of why they are beyond the scope of this study. It should also be made clear that it is the future researcher’s responsibility to make sure that the assumption made in applying the methods that are included in the competition are satisfied for their respective application. This is necessary to guarantee the quality of research that will build on the proposed competition results.

Reply to R1.11 We agree that in principle it would be very interesting to extend the Challenge to 3D motion. Unfortunately, this would require to introduce a series of different geometries that would completely alter the focus of the competition, making it too broad.

We agree with the Reviewer that the readers should be made clearly aware of the ensuing limitations. Thus, we have amended the text to clarify the biological significance and better specified the possible drawbacks of applying 2D approaches to 3D motion (lines 136–151):

The standard and straightforward approach in live-cell single-molecule imaging primarily captures information related to lateral motion. In cases involving flat membranes or isotropic systems, employing 2D imaging and tracking techniques suffices for obtaining accurate motion-related parameters. However, when dealing with motion on non-flat surfaces or within anisotropic 3D environments, relying solely on 2D projections can result in critical information being overlooked, potentially leading to the misinterpretation of diffusion coefficients or the appearance of apparent anomalous diffusion effects [57,58]. Consequently, drawing definitive conclusions under such circumstances should be avoided or approached with caution. To study motion occurring in 3D space, it is advisable to employ 3D tracking methods, such as off-focus imaging (i.e., the analysis of ring patterns in the defocused point spread function) [59], interference/holographic approaches [60], multifocus imaging [61], or point spread function engineering [62]. Although more challenging, these methods can measure also the motion along the axial dimension, facilitating a more accurate characterization. For the purposes of the Challenge, we have chosen to concentrate on studying changes in diffusion behavior occurring within a 2D context, driven by particle interactions of various types.

In addition, in the discussion of the final manuscript, we will explicitly address the responsibility of researchers to validate and adapt the methods to their particular research contexts. We will also provide guidance on how researchers can approach this process effectively, ensuring that the quality and relevance of their research remain high.

Reply to Reviewer 2

Comment R2.1

The results of AnDi Challenge 1, which aimed to develop state-of-the-art methods for analysing diffusion and comparing them with each other, were published in the journal Nature Commun, 12, 6253 (<https://doi.org/10.1038/s41467-021-26320-w>). This submission is devoted to AnDi Challenge 2. The analysis of methods performance is also successful as a previous one. The datasets used in the challenge include different experiments, but it is fixed to a specific model and a given set of parameters. The performance of existing and novel methods is estimated using special metrics. I believe that the software library created in this work will be useful for a wide circle of readers.

This work has been carried out thoroughly. It can identify two classes of methods: (i) ensemble methods and (ii) single-trajectory methods. Since the method for each specific problem is chosen from researcher's preference, a variety of simulations facilitate this choice. The combination of experimental data analysis with simulations makes this work meaningful.

I am satisfied with the answers of the authors to my referee report. The authors have significantly improved their presentation. I recommend this manuscript for publication in Nature Communications.

Reply to R2.1 We thank the reviewer for endorsing our work.

References

- [1] M. J. Saxton, “A biological interpretation of transient anomalous subdiffusion. i. qualitative model,” *Biophysical journal*, vol. 92, no. 4, pp. 1178–1191, 2007.
- [2] A. Furlan, M. Gonzalez-Pisfil, A. Leray, D. Champelovier, M. Henry, C. Le Nézet, O. Bensaude, M. Lefranc, T. Wohland, B. Vandenbunder, *et al.*, “Hexim1 diffusion in the nucleus is regulated by its interactions with both 7sk and p-tefb,” *Biophysical Journal*, vol. 117, no. 9, pp. 1615–1625, 2019.
- [3] M. Woringer and X. Darzacq, “Protein motion in the nucleus: from anomalous diffusion to weak interactions,” *Biochemical Society Transactions*, vol. 46, no. 4, pp. 945–956, 2018.
- [4] D. Krapf, “Mechanisms underlying anomalous diffusion in the plasma membrane,” *Current topics in membranes*, vol. 75, pp. 167–207, 2015.
- [5] F. Höfling and T. Franosch, “Anomalous transport in the crowded world of biological cells,” *Reports on Progress in Physics*, vol. 76, no. 4, p. 046602, 2013.
- [6] R. Metzler, J.-H. Jeon, A. G. Cherstvy, and E. Barkai, “Anomalous diffusion models and their properties: non-stationarity, non-ergodicity, and ageing at the centenary of single particle tracking,” *Physical Chemistry Chemical Physics*, vol. 16, no. 44, pp. 24128–24164, 2014.
- [7] C. Manzo and M. F. Garcia-Parajo, “A review of progress in single particle tracking: from methods to biophysical insights,” *Reports on Progress in Physics*, vol. 78, no. 12, p. 124601, 2015.
- [8] J. A. Torreno-Pina, C. Manzo, and M. F. Garcia-Parajo, “Uncovering homo-and hetero-interactions on the cell membrane using single particle tracking approaches,” *Journal of Physics D: Applied Physics*, vol. 49, no. 10, p. 104002, 2016.
- [9] P. R. Smith, I. E. Morrison, K. M. Wilson, N. Fernandez, and R. J. Cherry, “Anomalous diffusion of major histocompatibility complex class i molecules on hela cells determined by single particle tracking,” *Biophysical Journal*, vol. 76, no. 6, pp. 3331–3344, 1999.
- [10] M. Weiss, H. Hashimoto, and T. Nilsson, “Anomalous protein diffusion in living cells as seen by fluorescence correlation spectroscopy,” *Biophysical journal*, vol. 84, no. 6, pp. 4043–4052, 2003.
- [11] C. Manzo, J. A. Torreno-Pina, P. Massignan, G. J. Lapeyre Jr, M. Lewenstein, and M. F. G. Parajo, “Weak ergodicity breaking of receptor motion in living cells stemming from random diffusivity,” *Physical Review X*, vol. 5, no. 1, p. 011021, 2015.
- [12] K. Ritchie, X.-Y. Shan, J. Kondo, K. Iwasawa, T. Fujiwara, and A. Kusumi, “Detection of non-brownian diffusion in the cell membrane in single molecule tracking,” *Biophysical journal*, vol. 88, no. 3, pp. 2266–2277, 2005.
- [13] F. Daumas, N. Destainville, C. Millot, A. Lopez, D. Dean, and L. Salomé, “Confined diffusion without fences of a G-protein-coupled receptor as revealed by single particle tracking,” *Biophysical Journal*, vol. 84, no. 1, pp. 356–366, 2003.
- [14] D. S. Banks and C. Fradin, “Anomalous diffusion of proteins due to molecular crowding,” *Biophysical journal*, vol. 89, no. 5, pp. 2960–2971, 2005.
- [15] K. M. Spillane, J. Ortega-Arroyo, G. de Wit, C. Eggeling, H. Ewers, M. I. Wallace, and P. Kukura, “High-speed single-particle tracking of gm1 in model membranes reveals anomalous diffusion due to interleaflet coupling and molecular pinning,” *Nano letters*, vol. 14, no. 9, pp. 5390–5397, 2014.

- [16] A. Mosqueira, P. A. Camino, and F. J. Barrantes, “Antibody-induced crosslinking and cholesterol-sensitive, anomalous diffusion of nicotinic acetylcholine receptors,” *Journal of Neurochemistry*, vol. 152, no. 6, pp. 663–674, 2020.
- [17] Y.-J. Chai, C.-Y. Cheng, Y.-H. Liao, C.-H. Lin, and C.-L. Hsieh, “Heterogeneous nanoscopic lipid diffusion in the live cell membrane and its dependency on cholesterol,” *Biophysical Journal*, vol. 121, no. 16, pp. 3146–3161, 2022.
- [18] I. Golding and E. C. Cox, “Physical nature of bacterial cytoplasm,” *Physical review letters*, vol. 96, no. 9, p. 098102, 2006.
- [19] M. Magdziarz, A. Weron, K. Burnecki, and J. Klafter, “Fractional brownian motion versus the continuous-time random walk: A simple test for subdiffusive dynamics,” *Physical review letters*, vol. 103, no. 18, p. 180602, 2009.
- [20] E. Kepten, I. Bronshtein, and Y. Garini, “Ergodicity convergence test suggests telomere motion obeys fractional dynamics,” *Physical Review E*, vol. 83, no. 4, p. 041919, 2011.
- [21] S. A. Tabei, S. Burov, H. Y. Kim, A. Kuznetsov, T. Huynh, J. Jureller, L. H. Philipson, A. R. Dinner, and N. F. Scherer, “Intracellular transport of insulin granules is a subordinated random walk,” *Proceedings of the National Academy of Sciences*, vol. 110, no. 13, pp. 4911–4916, 2013.
- [22] I. Bronshtein, E. Kepten, I. Kanter, S. Berezin, M. Lindner, A. B. Redwood, S. Mai, S. Gonzalo, R. Foisner, Y. Shav-Tal, *et al.*, “Loss of lamin a function increases chromatin dynamics in the nuclear interior,” *Nature communications*, vol. 6, no. 1, p. 8044, 2015.
- [23] S. C. Weber, A. J. Spakowitz, and J. A. Theriot, “Bacterial chromosomal loci move subdiffusively through a viscoelastic cytoplasm,” *Physical review letters*, vol. 104, no. 23, p. 238102, 2010.
- [24] T. J. Lampo, S. Stylianidou, M. P. Backlund, P. A. Wiggins, and A. J. Spakowitz, “Cytoplasmic rna-protein particles exhibit non-gaussian subdiffusive behavior,” *Biophysical journal*, vol. 112, no. 3, pp. 532–542, 2017.
- [25] J.-P. Bouchaud and A. Georges, “Anomalous diffusion in disordered media: statistical mechanisms, models and physical applications,” *Physics reports*, vol. 195, no. 4-5, pp. 127–293, 1990.
- [26] H. Scher, M. F. Shlesinger, and J. T. Bendler, “Time-scale invariance in transport and relaxation,” *Physics Today*, vol. 44, no. 1, pp. 26–34, 1991.
- [27] J. Klafter and G. Zumofen, “Lévy statistics in a hamiltonian system,” *Physical Review E*, vol. 49, no. 6, p. 4873, 1994.
- [28] D. Mainali and E. A. Smith, “The effect of ligand affinity on integrins’ lateral diffusion in cultured cells,” *European Biophysics Journal*, vol. 42, pp. 281–290, 2013.
- [29] M. Yanagawa, M. Hiroshima, Y. Togashi, M. Abe, T. Yamashita, Y. Shichida, M. Murata, M. Ueda, and Y. Sako, “Single-molecule diffusion-based estimation of ligand effects on g protein-coupled receptors,” *Science Signaling*, vol. 11, no. 548, p. eaao1917, 2018.
- [30] B. da Rocha-Azevedo, S. Lee, A. Dasgupta, A. R. Vega, L. R. de Oliveira, T. Kim, M. Kittisopikul, Z. A. Malik, and K. Jaqaman, “Heterogeneity in vegf receptor-2 mobility and organization on the endothelial cell surface leads to diverse models of activation by vegf,” *Cell reports*, vol. 32, no. 13, 2020.

- [31] A. M. Achimovich, T. Yan, and A. Gahlmann, “Dimerization of ilid optogenetic proteins observed using 3d single-molecule tracking in live e. coli,” *Biophysical Journal*, vol. 122, no. 16, pp. 3254–3267, 2023.
- [32] S. T. Low-Nam, K. A. Lidke, P. J. Cutler, R. C. Roovers, P. M. van Bergen en Henegouwen, B. S. Wilson, and D. S. Lidke, “ErbB1 dimerization is promoted by domain co-confinement and stabilized by ligand binding,” *Nature structural & molecular biology*, vol. 18, no. 11, pp. 1244–1249, 2011.
- [33] C. C. Valley, D. J. Arndt-Jovin, N. Karedla, M. P. Steinkamp, A. I. Chizhik, W. S. Hlavacek, B. S. Wilson, K. A. Lidke, and D. S. Lidke, “Enhanced dimerization drives ligand-independent activity of mutant epidermal growth factor receptor in lung cancer,” *Molecular Biology of the Cell*, vol. 26, no. 22, pp. 4087–4099, 2015.
- [34] A. Tabor, S. Weisenburger, A. Banerjee, N. Purkayastha, J. M. Kaindl, H. Hübner, L. Wei, T. W. Grömer, J. Kornhuber, N. Tschammer, *et al.*, “Visualization and ligand-induced modulation of dopamine receptor dimerization at the single molecule level,” *Scientific Reports*, vol. 6, no. 1, pp. 1–16, 2016.
- [35] T. Sungkaworn, M.-L. Jobin, K. Burnecki, A. Weron, M. J. Lohse, and D. Calebiro, “Single-molecule imaging reveals receptor–g protein interactions at cell surface hot spots,” *Nature*, vol. 550, no. 7677, pp. 543–547, 2017.
- [36] J. Grimes, Z. Koszegi, Y. Lanoiselee, T. Miljus, S. L. O’Brien, T. M. Stepniewski, B. Medel-Lacruz, M. Baidya, M. Makarova, D. M. Owen, *et al.*, “Single-molecule analysis of receptor-beta-arrestin interactions in living cells,” *bioRxiv*, pp. 2022–11, 2022.
- [37] J. Janczura, M. Balcerek, K. Burnecki, A. Sabri, M. Weiss, and D. Krapf, “Identifying heterogeneous diffusion states in the cytoplasm by a hidden markov model,” *New Journal of Physics*, vol. 23, no. 5, p. 053018, 2021.
- [38] G. Lee, G. Leech, M. J. Rust, M. Das, R. J. McGorty, J. L. Ross, and R. M. Robertson-Anderson, “Myosin-driven actin-microtubule networks exhibit self-organized contractile dynamics,” *Science Advances*, vol. 7, no. 6, p. eabe4334, 2021.
- [39] B. Requena, S. Masó, J. Bertran, M. Lewenstein, C. Manzo, and G. Muñoz-Gil, “Inferring pointwise diffusion properties of single trajectories with deep learning,” *arXiv preprint arXiv:2302.00410*, 2023.
- [40] D. Han, N. Korabel, R. Chen, M. Johnston, A. Gavrilova, V. J. Allan, S. Fedotov, and T. A. Waigh, “Deciphering anomalous heterogeneous intracellular transport with neural networks,” *Elife*, vol. 9, p. e52224, 2020.
- [41] W. Wang, F. Seno, I. M. Sokolov, A. V. Chechkin, and R. Metzler, “Unexpected crossovers in correlated random-diffusivity processes,” *New Journal of Physics*, vol. 22, no. 8, p. 083041, 2020.
- [42] M. Balcerek, K. Burnecki, S. Thapa, A. Wyłomańska, and A. Chechkin, “Fractional brownian motion with random hurst exponent: Accelerating diffusion and persistence transitions,” *Chaos: An Interdisciplinary Journal of Nonlinear Science*, vol. 32, no. 9, 2022.
- [43] W. Wang, M. Balcerek, K. Burnecki, A. V. Chechkin, S. Janušonis, J. Ślęzak, T. Vojta, A. Wyłomańska, and R. Metzler, “Memory-multi-fractional brownian motion with continuous correlations,” *Physical Review Research*, vol. 5, no. 3, p. L032025, 2023.
- [44] J. Ślęzak and R. Metzler, “Minimal model of diffusion with time changing hurst exponent,” *Journal of Physics A: Mathematical and Theoretical*, vol. 56, no. 35, p. 35LT01, 2023.

- [45] K. Jaqaman, D. Loerke, M. Mettlen, H. Kuwata, S. Grinstein, S. L. Schmid, and G. Danuser, “Robust single-particle tracking in live-cell time-lapse sequences,” *Nature methods*, vol. 5, no. 8, pp. 695–702, 2008.
- [46] N. Chenouard, I. Smal, F. De Chaumont, M. Maška, I. F. Sbalzarini, Y. Gong, J. Cardinale, C. Carthel, S. Coraluppi, M. Winter, *et al.*, “Objective comparison of particle tracking methods,” *Nature methods*, vol. 11, no. 3, pp. 281–289, 2014.

Point-by-point response to reviewers of *Quantitative evaluation of methods to analyze motion changes in single-particle experiments*

Muñoz-Gil et al.

Response to Reviewer 1

Comment 1

Summary

This manuscript has been accepted as a registered report. It describes a competition to quantitatively compare methods that analyze motion changes in single-molecule and single-particle tracking experiments. The competition is a modified repeat of a previous competition that the authors organized in 2021, called the first AnDi Challenge. Three reasons are given for running a second AnDi challenge: First, there is a need to evaluate methods that determine the switch between different diffusive behaviors within single-molecule trajectories. Second, it is necessary to assess the methods' crosstalk in detecting inherent anomalous diffusion due to motion constraints. Third, there is a need to determine whether the bottleneck of the analysis process is at the level of the analysis of the single trajectories or associated with their extraction from the experimental videos. The authors are now providing the analysis and discussion of the Challenge results.

The results presented are interesting and intriguing, but I am afraid I am not grasping the overall conclusion that could be drawn based on these results. My main critique is that the results are shown in terms of error metrics, which (while they are clearly defined in the manuscript) do not provide an intuitive understanding of how good (or bad) the performance of individual tracking methods were. I get the vague sense that none of them were perfect in all tasks. What methods should experimentalist then turn to (or not) to analyze single-molecule/single-cell trajectory data (of a specific type, which may be known a priori). Perhaps the authors could summarize the results into a few, very clear take-home messages. Second, please discuss the results more clearly and in more detail in reference to the three points mentioned above (that provided the rationale for this second AnDi challenge). I am interested in learning what key insights was gained from the results of this challenge.

The critiques below focus mostly on the results and discussion section, as the other part of the manuscript have already been reviewed.

Reply to Comment 1

We thank the reviewer for this clear and constructive summary. To address the three original rationales of the second AnDi Challenge, we have explicitly linked key results to their corresponding motivation throughout both the Results and Discussion sections.

Additionally, to provide an intuitive overview of our findings, we have added a new subsection—*Results Summary and Take-home Messages*—at the end of the Results. This section distills the performance of top methods into a few actionable insights:

Results Summary and Take-home Messages

Robust changepoint detection: Top single-trajectory methods (e.g., based on UNet3+) consistently achieve over 95% accuracy in identifying segment boundaries, with only minor false-positive rates across all scenarios.

Distinguishing confinement, immobilization, and anomalous diffusion: Leading algorithms accurately classify segments arising from geometric constraints or anomalous dynamics. Only very short segments and exponents close to zero remain challenging, indicating minimal crosstalk between distinct diffusion mechanisms.

Trajectory extraction is a bottleneck: Video-Track performance lags the Trajectory Track by 10–30%, highlighting that linking and localization errors—not downstream analysis—drive most of the accuracy loss.

Parameter estimation benefits from physical priors: Incorporating known physical models may yield significant gains in changepoint detection, but separate estimation pipelines for K and α result in only modest improvements in parameter accuracy.

Dedicated ensemble approaches are needed: Ensemble Task submissions rely on Gaussian mixture model or k-means clustering of per-trajectory outputs, which fragments broad parameter distributions (e.g., EXP 7–8). Ensemble approaches, either bypassing single-trajectory clustering or using more sophisticated grouping techniques, hold potential for uncovering population-scale insights.

Comment 2

Specific Comments:

1. Line 469: “In this case, only the top method (team I, light blue) was able to produce a reliable classification of the diffusion type of the segments. The difficulty in inferring the correct type of mechanism producing interaction underscores the challenges in accurately analyzing these kind of data, which can have significant implications for the biological interpretation of the results. However, despite the misclassification, all predictions provided a reliable estimation of the residence time within both trapped and unconstrained states (inset of Fig. 6a), thus allowing precise quantification of the relevant biophysical parameters of the experiments.” a. The first conclusion seems to indicate the substantial care should be taken when drawing biological conclusions because tracking analysis results may be incorrect. By contrast, the conclusion just two sentences later states that biophysical parameters can be precisely quantified. I am therefore not sure what I should take away from these seemingly differing conclusion. Please define more clearly what is meant by “biological interpretation” and “relevant biophysical parameters” to communicate a more cohesive message.

Reply to Comment 2

We thank the reviewer for this comment and agree that the phrasing of the original sentence was misleading. We aimed to emphasize that, even when state classification is not perfectly accurate, the

participating algorithms yield highly precise estimates of key biophysical parameters. In this context, *relevant biophysical parameters* refers specifically to the dwell times in each diffusive state. Accurate dwell-time measurements are critical for interpreting many molecular processes—allowing, for example, quantification of binding kinetics, confinement lifetimes, or transition rates—and thereby link the observed motion directly to underlying biological functions.

Accordingly, we have replaced the original line with:

Although perfect classification of diffusive states remains challenging, the algorithms nonetheless provide precise estimates of critical biophysical parameters, namely, the average dwell times in both trapped and unconstrained states (inset of Fig. 6a). The measure of these parameters is essential for quantifying binding kinetics, confinement lifetimes, and transition rates that directly inform biological interpretation.

Comment 3

Line 482: “Four of the top 5 methods classify the segment rather precisely, which allows them to accurately recover the distribution of each diffusive state, as shown in Fig. 6b–d.”

a. I do not know how to interpret “rather precisely”. Such language is too vague. Determining a segments length/duration is often of great biological significance. Can the methods do this well or not?

b. Perhaps not surprisingly, Fig. 6b shows that it is rather straightforward to distinguish a moving particle from an immobile one and assign trajectory segments accordingly. I believe it should be pointed out that this type of experimental scenario has a LONG history in the field with many case examples that could be cited where the mode of motion contrast is from immobile/quasi stationary to fast moving.

c. I would argue that Fig 6c does not show that the distribution of each diffusive state is accurately recovered by 4 of the top 5 methods. A more nuanced/detailed description is needed here. Since the participants of the top teams are listed as authors on this report, could they provide some insights into why their methods performed particularly well (or not)?

Reply to Comment 3

We thank the reviewer for this constructive feedback. Following the reviewer’s comments, we have now clarified the description of Fig. 6. Figure 6 evaluates method performance on three visually similar but mechanistically distinct scenarios, where free diffusion is interrupted by restricted mobility generated by different mechanisms. We aimed to test whether algorithms can discern these subtle differences:

- **EXP 3:** transient trapping events interrupting diffusion,
- **EXP 4:** confined anomalous diffusion within small compartments,
- **EXP 5:** pronounced subdiffusion without spatial constraints ($\alpha \approx 0.2$).

Despite the challenge, the top methods perform remarkably well. For example, none misclassify the restricted motion in EXP 4 or EXP 5 as pure immobilization.

In revision, we have addressed the reviewer’s points:

- Removed the vague phrase “rather precisely” and replaced it with explicit accuracy metrics.
- Cited the extensive literature on immobile–mobile segmentation in both the Introduction and Supplementary Table 1.
- Explained why EXP 4 (Fig. 6c) is the hardest case: short segments yield few boundary reflections, making it difficult to distinguish true confined from unconstrained anti-persistent subdiffusion.

The updated paragraph now reads:

Figure 6b–d highlights the performance of the top five methods across EXP 3–5. Teams I, C, and R each correctly classify over 95% of segments, closely matching the true distribution of diffusive states. Team E tends to over-label segments as diffusive, while Team O occasionally confuses confined segments for diffusive ones and vice versa. Team R, despite its high overall accuracy, also makes occasional misclassifications of diffusive segments as immobile or confined. Importantly, for EXP 4 (small-radius confinement) and EXP 5 (dimerization-induced subdiffusion), misclassification as immobile is negligible for Teams I, C, and R. Detecting confinement in EXP 4 is particularly challenging since short dwell times in confined areas yield few boundary reflections, inducing confusion with unconstrained anti-persistent subdiffusion of EXP 5. The ability of Teams I, C, and R to resolve these subtle cases underscores the high sensitivity and robustness of their methods.

Comment 4

3. Line 487: “The information contained in an individual trajectory is typically sufficient to estimate CPs and diffusive properties. However, for some physical models, the knowledge of the model itself offers additional information that could be used to improve further CP detection and parameter estimation. This is the case for QTM and TCM, where changes in diffusion correspond to spatial constraints. For DIM, diffusion changes are associated with particle proximity; in addition, since particles in a dimer co-diffuse, one could use twice as much information to estimate K and α .” a. Does the last sentence assume that both co-diffusing particles are detected simultaneously? That is a very unlikely case in STORM and PALM based single-molecule tracking due to the stochastic nature of single-molecule signals. Please rephrase if necessary.

Reply to Comment 4

We agree that under typical experimental conditions, especially in SMLM-based methods with blinking and sparse activation, simultaneous detection of both members of a dimer is unlikely. Our original comment referred to idealized tracking conditions as implemented in the Trajectory Track of the Challenge, where continuous, long trajectories are available. In such single-particle tracking (SPT) experiments without blinking (e.g., using stable fluorophores), co-diffusing pairs can indeed be observed, although resolution limits and spot overlap can complicate separation. These issues can be mitigated

by dual-color imaging approaches.

Accordingly, we have clarified the text and replaced the original sentence with:

For DIM, diffusion changes are associated with particle proximity; in addition, since particles in a dimer co-diffuse, one could in principle use twice as much information to estimate K and α , although in typical experimental conditions it may be very challenging to track two co-diffusing particles.

Comment 5

4. Line 542: “We would further like to point out that optimizing methods to provide high scores for the metrics of the competition did not always translate into more meaningful insights about the underlying physical processes. For instance, teams M, H, and O showed significant biases across all experiments when predicting the K distribution but still achieved high rankings according to the metric in Eq. (3) (Supp. Fig. 6). Moreover, accurately predicting the number of true states did not provide a clear advantage with this metric, as most top teams overestimated the number of states but carefully adjusted their relative weights to minimize differences with the ground-truth distribution.” a. I appreciate the authors bringing up this point. Would it be possible then to define and discuss other metrics that would provide insights into what type of trajectories are particularly challenging for the methods, which methods perform particularly well for a given scenario, which parameters should be trusted and which ones not in a given scenario. In many cases, a priori biological knowledge about a system may narrow down the types of motion that are expected. This manuscript should provide some clear take-home messages for researchers that would like to analyze single-molecule/single-particle trajectories reliably. b. This paragraph should perhaps be moved to the discussion.

Reply to Comment 5

We thank the reviewer for this suggestion. A main limitation that we found was that many participants focused on minimizing the W_1 distance—matching the predicted and ground-truth distributions—without any post-processing to limit the number of states or improve physical interpretability (for example, filtering overlapping modes or very low-population segments). Moreover, as noted in the manuscript, no team developed dedicated ensemble-level algorithms; instead, existing statistical tools (e.g., Gaussian mixture model or k-means) were applied to single-trajectory features.

During our post-challenge analysis, we explored alternative metrics that combined distributional fidelity with correct state counting. However, merging these criteria introduced artificial biases and failed to produce a more informative ranking. Consequently, we retained W_1 as our primary evaluation metric and have expanded the Discussion to detail its limitations, highlighting that these distribution-matching results are most meaningful when interpreted alongside a priori biological knowledge.

The updated Discussion text is included at the end of the next comment, which closely relates to this one.

Comment 6

5. Line 584: “Second, ensemble predictions currently rely on post-processing of single-trajectory data. All participants in the Ensemble Task first extracted features from individual trajectories, followed by a post-processing step to estimate the parameters of the diffusive populations. Although this approach has yielded promising results in terms of the proposed metrics, it remains challenging to derive physical insights from such predictions due to the high number of predicted states and the instability in the variances of each mode. We anticipate that methods capable of directly predicting distributions from raw data may avoid the potential biases inherent in single-trajectory analyses. Furthermore, approaches that incorporate the entire set of trajectories contextually may offer improved performance.” a. Given that high number of predicted states can lead to problems, the authors should caution readers here to not blindly trust the outputs of (still imperfect, I take it) analysis methods. Care should always be taken not to overfit the data with too many states that cannot be assigned to a biophysical event or process. b. If the analysis suggests/predicts a high number of states, their identities should be validated through additional experiments utilizing purposeful biological, chemical, or physical perturbation. I suggest the manuscript provide context of this nature rather than solely presenting and ranking the results.

Reply to Comment 6

We thank the reviewer for raising this important point. We fully agree that readers must be warned against uncritical acceptance of analytical outputs and the dangers of overfitting. Accordingly, we have expanded the Discussion to provide clear recommendations for additional validation whenever an analysis yields an unexpectedly large number of states.

To clarify, we have added the following text:

Second, in the Ensemble Task, all participants relied on post-processing of single-trajectory outputs. Features were first extracted from individual trajectories and then a separate step was used to infer the parameters of the diffusive populations. No team developed dedicated ensemble-level algorithms or used established ensemble frameworks.

Although this single-trajectory-based approach produced high Challenge rankings, it offered limited biophysical insight due to the proliferation of predicted states and the instability of each mode’s variance. Minimizing the Wasserstein-1 (W_1) distance aligns predicted and ground-truth distributions, but W_1 offers no penalty for over-splitting into numerous states or for unstable variance estimates, nor does it encourage physically interpretable solutions (e.g., filtering overlapping modes or very low-population segments). This warns us that outputs should not be blindly trusted when applied to real experiments. Care should always be taken not to overfit the data with too many states that cannot be assigned to a biophysical process. Whenever an analysis yields a large number of states, their identities should be validated through control experiments. In practice, a priori biological knowledge often narrows the expected state count, providing essential context for interpreting algorithmic results.

Looking ahead, methods capable of inferring population distributions directly from the

raw ensemble of trajectories, thereby bypassing single-trajectory feature extraction and clustering, may deliver deeper physical insights. Moreover, approaches that treat the full set of trajectories contextually, rather than in isolation, are likely to enhance both performance and interpretability.

Response to Reviewer 2

Comment 1

I find this work very useful and interesting. It can be suitable for publication in Nature. However, I have questions and comments. Their list is presented below:

Reply to Comment 1

We thank the reviewer for their positive assessment and are pleased that they find the work useful and interesting. We address all of their questions and comments in detail below.

Comment 2

1) As far as I understand (line 1056), the trajectories had a maximum length of 200 frames (timesteps). Were there data gaps (NaN) in trajectories?

Reply to Comment 2

We thank the reviewer for this question. Although many linking algorithms permit gaps in trajectories, we chose to focus exclusively on segmentation performance and therefore did not introduce missing-frame complications that could confound interpretation. We have added the following sentence to the Extended Methods to make this explicit:

All simulated trajectories were generated without missing frames: no gaps were introduced, yielding continuous tracks to isolate segmentation performance from linking or gap-filling complexities.

Comment 3

2) What number of segments were used? From 1 up to 10? What was the minimum segment length in result? I see line 985: "we set $t_{\min} = 3$.(integer or real?)". How to determine statistics with help of 3 values? I think three time steps are too small for any quantitative statistical analysis. Why did you select 3 timesteps? Or did I misunderstand something?

Reply to Comment 3

We thank the reviewer for raising this important point. As detailed in the Extended Methods, the number of segments per trajectory was not fixed in advance but was determined by the biophysical parameters of each simulation model. The full distribution of changepoint counts and segment lengths appears in Supplementary Figure 1a–b.

To ensure that every segment retained minimal statistical information, we imposed a lower bound of $t_{\min} = 3$ (integer) frames. Although quantitative inference from only three points is indeed challenging, this threshold was chosen intentionally to probe the limits of segmentation methods. Moreover, many segments of the same diffusive state recur within each trajectory, allowing algorithms to aggregate

evidence across repeated segments. Previous work (cited in our manuscript) has successfully addressed similarly brief segments, and several top Challenge performers achieved high accuracy even for these short intervals (see Fig. 5c–e).

To clarify, we have added the following sentence to the Extended Methods:

For the Challenge, we set $t_{\min} = 3$ frames to test the sensitivity and robustness of the segmentation methods under minimal data conditions.

We have also corrected the punctuation to present $t_{\min} = 3$ as an integer threshold.

Comment 4

3) lines 413-416: "However, we also find less correlation between metric scores. This can be attributed to the fact that the predictions of K and α are generally obtained through different approaches, thus enhancing performance for one does not necessarily result in a corresponding improvement for the other." This could mean something else. Different approaches have different accuracy. Or am I wrong?

Reply to Comment 4

We thank the reviewer for this observation. We intended to convey that each team’s pipeline begins with a changepoint detection. Thus, it is reasonable to expect that any improvement in that step raises the Jaccard score (JSC) and lowers the root-mean-square error (RMSE) for changepoint localization, producing the strong correlation seen in Fig. 3c. By contrast, the diffusion coefficient K and anomalous exponent α are typically estimated by separate algorithms, so gains in one do not automatically improve the other, resulting in a weaker correlation between their error metrics.

To clarify, we have rephrased the sentence as follows:

Across methods, enhanced CP detection, reflected by higher JSC and lower RMSE, yields a tight correlation between these metrics (Fig. 3c). A similar but weaker trend appears for K and α errors (Fig. 3d–e), because their estimation often relies on distinct algorithms, decoupling improvements in one from the other.

Comment 5

4) Fig.2 (right panel): no ticklabels in the time axis? Am I right that each tick is 40 frames?

Reply to Comment 5

We thank the reviewer for pointing this out. We have added tick labels to the right panel of Fig. 2.

Comment 6

5) It seems to me that Fig.8 is not quite clear from its caption. What do the panels show? Can you explain in more accessible language?

Reply to Comment 6

We have clarified and streamlined the caption as follows:

Ensemble Task predictions for the Trajectory Track. Panels (a–d) show the predicted distributions of the diffusion coefficient K and panels (e–h) the anomalous exponent α for Experiments 4, 7, 8, and 9, respectively. Distributions were computed from the estimated means and variances (see Scoring and evaluation – Ensemble Task). Only the results of the top five teams are displayed, with color indicating rank (blue is the highest, red the lowest). Black curves denote the ground-truth distributions.

Comment 7

6) lines 530-531: "However, even the top 5 teams had difficulty in reliably estimating the parameter distribution for some experiments." I see that the teams could not cope with this task at all. Or is it not all so gloomy? I would like to see a more critical analysis of the results. What approaches failed?

Reply to Comment 7

We thank the reviewer for this comment. While the top teams generally produced accurate predictions in many experiments (e.g., Fig. 8a,b), they struggled most with the wide dynamic range of K in EXP 7 and EXP 8 (Fig. 8b,c). In these cases, segmentation-based pipelines followed by standard clustering (Gaussian mixture model or k-means) tended to fragment the K values into too many narrow modes, failing to capture the true broad distribution.

To clarify this point, we have inserted the following sentence into the Results section:

This is particularly evident for the K distribution in EXP 7 and EXP 8 (Fig. 8b,c). Since the methods rely on estimates of K per segment and then apply Gaussian mixture models or k-means, they generally tend to over-fragment wide K ranges, misrepresenting the overall distribution.

Comment 8

7) Fig.7: how to explain the large spread of improvement (%) results on the left panel? What does it mean the improvement in 200% and more?

Reply to Comment 8

We define the relative improvement for a given metric m as

$$\Delta m(\%) = \frac{m_{\text{DIM}} - m_{\text{MSM}}}{m_{\text{MSM}}} \times 100\%,$$

where m_{DIM} and m_{MSM} are the metric values under the DIM and MSM models, respectively. For example, $\Delta m = 200\%$ implies $m_{\text{DIM}} = 3 m_{\text{MSM}}$, i.e. the DIM prediction is three times better than MSM.

To clarify this in the manuscript, we have added:

To quantify model-based gains, we computed the relative improvement

$$\Delta m(\%) = \frac{m_{\text{DIM}} - m_{\text{MSM}}}{m_{\text{MSM}}} \times 100\%$$

for each subtask metric (JSC, MSLE, and MAE). Figure 7 reports these improvements for all methods, with the overall average shown as a dashed line.

We attribute the broad spread of JSC gains to the extra information provided by dimerization, which substantially enhances changepoint detection.

Comment 9

8) lines 147-149: "Each method has its own set of advantages and disadvantages, and its performance may depend on the specific problem under consideration. However, there is no universally accepted gold standard for determining which method to use to address each specific problem." Can you write more specifically here? What do you mean? Here it looks like some kind of philosophical generalization.

Reply to Comment 9

This statement points to a concrete gap: although many algorithms exist for detecting changepoints in diffusive trajectories (ranging from threshold-based sliding-window methods to Bayesian inference and machine-learning approaches) there is currently no common dataset or agreed-upon set of performance metrics that allows head-to-head comparison of their accuracy, speed, and robustness. Establishing such a unified benchmark suite with standardized simulation, ground-truth annotations, and evaluation criteria, was the primary objective of the AnDi Challenge.

We note that this passage belongs to a section that was finalized and accepted during Stage 1 review and therefore cannot be altered at this point. We trust the clarification above helps justify the original wording.

Comment 10

9) lines 165-167: "However, the choice of FBM did not limit the generality of the Challenge since other models of diffusion and non-Gaussian behavior can be obtained by properly tuning the parameters of the simulations." This is not true. You won't get, for example, the subordinated processes or resetting from FBM. I believe that the choice of FBM actually limited the generality of the Challenge. This is not bad. It is impossible to cover the entire diversity of experimental cases at once.

Reply to Comment 10

We thank the reviewer for highlighting this important point. It is indeed correct that standard FBM cannot generate processes such as subordinated dynamics or stochastic resetting. Our intention was to convey that, by coupling FBM with the four interaction models we introduce (multi-state switching, transient confinement, quenched trapping, and dimerization), we can reproduce a broad array of diffusive and non-Gaussian behaviors within a single framework. While this approach does not capture every conceivable diffusion mechanism, it systematically spans heterogeneity, confinement, trapping, and binding-like events, which are central to many biological systems.

We note that this passage belongs to a section finalized and accepted during Stage 1 review and therefore cannot be altered at this point. We trust the clarification above helps justify the original wording.

Comment 11

10) line 274: "... 3 = directed ($\alpha \leq 1.9$)." Why 1.9 and not 1.95, for example? May be, it would be better to write $1 < \alpha \leq 1.9$.

Reply to Comment 11

We thank the reviewer for this suggestion. We have corrected the typographical error and clarified the state definitions as follows:

0 = immobile, 1 = confined, 2 = free (unconstrained, $0.05 \leq \alpha < 1.9$), 3 = directed ($1.9 \leq \alpha < 2.0$)

We chose the cut-off of $\alpha = 1.9$ (rather than, say, 1.95) to provide a clear separation between normal/subdiffusive motion and genuinely ballistic (directed) behavior, while allowing for statistical fluctuations in estimated exponents. In our simulations, trajectories with true directed motion consistently yielded α values above 1.9, whereas mild over-diffusive noise rarely exceeded this threshold. This choice minimizes misclassification due to sampling variability and ensures that only robust superdiffusive segments are labeled as directed.

Comment 12

Remarks on code availability:

In my opinion, this code is good. I see that the results of the article are reproducible. The code is useful for a wide community. It contains the necessary comments to install and run the application. I can install and run this code.

Reply to Comment 12

We thank the reviewer for these positive remarks. We are glad to hear that the installation and execution were straightforward and hope the code proves valuable to the broader research community.

This article focuses on the analysis of trajectories in single-particle experiments. It is based on methods following from properties of fractional Brownian motion, segmentation, hidden Markov model and so on. This research is of interest and could be worthy of publication in Nature. However, at present the manuscript needs some refinements and improvements. My comments are presented below.

1. I propose to change the title to: "Quantitative evaluation of **FBM-type** methods to analyze motion changes in single-particle experiments". See my remark 2.

2. The fractional Brownian motion is one of many models useful for the description of motion changes in single-particle experiments. There are other models mentioned in [7-9]. I understand that it is difficult to include all the models together in the same paper and compare them in terms of quality in a particular experiment. Nevertheless, I propose to indicate that this work is the first attempt to implement a competition to characterize and rank the performance of **FBM-type** methods in analyzing the dynamic behavior of single molecules.

3. In addition, it would be useful to mention alternative approaches (for example, [10.1103/PhysRevE.99.012101](https://arxiv.org/abs/10.1103/PhysRevE.99.012101), [10.1039/C8CP06781C](https://arxiv.org/abs/10.1039/C8CP06781C), [10.1039/c8cp06781c](https://arxiv.org/abs/10.1039/c8cp06781c), [10.1016/j.chaos.2021.111606](https://arxiv.org/abs/10.1016/j.chaos.2021.111606)) close to the analysis of FBM.

4. I have noticed in this paper that the classification of random trajectories is carried out only for many states, but the classification itself has a wider sense. This work considers Gaussian processes (although a fractional case). However, in many cases the random trajectories of single molecules can demonstrate a non-Gaussian behavior. This should be checked in advance. The reader should not get the impression that the authors offer a universal approach for all occasions. In this work, a particular problem is solved, although in a rather broad formulation.

5. In the references there is a mention of articles where all authors are shown without exception, despite a long list (for example, [15]), and there are articles where the authors are mentioned only the first of them [16,17,64]. This should be corrected.

6. Table 1 is incomplete. For example,

- Ergodicity breaking in trajectory dynamics ([10.1038/s41598-017-05911-y](https://arxiv.org/abs/10.1038/s41598-017-05911-y));

- Detection of trajectory-to-trajectory fluctuations ([10.1088/1367-2630/abf204](https://arxiv.org/abs/10.1088/1367-2630/abf204))

7. The classification of heterogeneity (p.2, left column, the first paragraph below) is very simplified like Table 1. I propose to reformulate the following your phrase: "There are three heterogeneity classes that

these methods aim to identify:" to " **We consider only** three heterogeneity classes that these methods aim to identify:" It seems to me that it will be more accurate.

8. While checking the Python software of the article, I ran into a data simulation problem. So slow and so long. I couldn't wait for the command:

```
dataset = AD.create_dataset(T = 4, N_models = 2, exponents = [0.7, 0.9], models = [0, 2],  
                           save_trajectories = True, path = 'datasets/')
```

to complete. My computer has pretty decent hardware: Xeon E5-2689, RAM 16 Gb and SSD 256 Gb. What are requirements for computers in your simulations? Can they be performed on a personal computer? Thus, the installation of your codes was OK, but my software verification in action is problematic.

9. I didn't fully understand what data are used in this paper. Only numerical simulations of random trajectories predefined properties? If yes, how can be this used for single-particle experiments or their analysis? It is not evident that your methods "guide researchers to identify optimal tools for analyzing their experiments". This approach is strongly limited in the framework of FBM. How do experimentalists know that their data are related to FBM?

10. In the end of Introduction (as well as in Abstract) the authors write something like "typical experimental conditions". What are conditions? Could you, please, represent them concretely?